# Towards Foundation Models for Cryo-ET Subtomogram Analysis

## Abstract

Cryo-electron tomography (cryo-ET) enables in situ visualization of macromolecular structures, where subtomogram analysis tasks such as classification, alignment, and averaging are critical for structural determination. However, effective analysis is hindered by scarce annotations, severe noise, and poor generalization. To address these challenges, we take the first step towards foundation models for cryo-ET subtomograms. First, we introduce CryoEngine, a large-scale synthetic data generator that produces over 904k subtomograms from 452 particle classes for pretraining. Second, we design an Adaptive Phase Tokenization-enhanced Vision Transformer (APT-ViT), which incorporates adaptive phase tokenization as an equivariance-enhancing module that improves robustness to both geometric and semantic variations. Third, we introduce a Noise-Resilient Contrastive Learning (NRCL) strategy to stabilize representation learning under severe noise conditions. Evaluations across 27 synthetic and real datasets demonstrate state-of-the-art (SOTA) performance on all three major subtomogram tasks and strong generalization to unseen datasets, advancing scalable and robust subtomogram analysis in cryo-ET.

## 1 Introduction

Cryo-electron tomography (cryo-ET) is a powerful imaging technique that enables high-resolution visualization of macromolecular structures in their native cellular contexts, and thus plays a critical role in advancing structural and cellular biology (Doerr, 2017; Ni et al., 2021). A typical cryo-ET analysis pipeline begins with particle picking (Moebel et al., 2021; Liu et al., 2024), where regions of interest are identified from 3D tomograms, followed by subtomogram-level analysises that extract structural and functional insights from these volumes (Kim et al., 2023). This paper focuses on subtomogram-level analysis, which involves three major tasks: (1) classification, to separate macromolecules into structurally or functionally distinct categories (Zeng et al., 2021a); (2) alignment, to register subtomograms by estimating rotations and translations into a common frame (Jiang et al., 2025); (3) averaging, to integrate aligned subtomograms for recovering high-resolution structures while suppressing noise and missing wedge artifacts (X & M, 2020). These tasks are essential for resolving molecular structures at high resolution (Chen et al., 2019; Hou et al., 2023) and for elucidating biological processes such as bacterial effector secretion (Chang et al., 2014; 2017) and mammalian neural function (Davies et al., 2011; Guo et al., 2018).

However, cryo-ET subtomogram analysis remains highly challenging. The difficulties can be attributed to four main factors: (1) the scarcity of high-quality annotated datasets, which limits effective model training; (2) the extremely low signal-to-noise ratios of subtomograms (approximately 0.01–0.1), further complicated by cytoplasmic background and low electron doses (Danev et al., 2010); (3) the random orientations and displacements of macromolecular structures, which introduce substantial geometric variability (Jiang et al., 2025); and (4) structural heterogeneity, as diverse complexes exhibit vastly different shapes, unlike the relatively consistent structures in medical imaging.

Beyond these general difficulties, each of the three subtomogram analysis tasks faces additional specific challenges. For classification, the primary challenge lies in distinguishing subtle structural variations under extremely low signal-to-noise conditions. Traditional template-matching approaches require predefined references and suffer from bias toward known structures (Zhan et al., 2025; Castaño-Díez et al., 2012). While recent deep learning methods (Zeng et al., 2021a) and unsupervised approaches (Zeng et al., 2023) have shown improvements, they remain highly sensitive to noise and

require carefully designed training strategies. For alignment and averaging, the key challenge stems from their interdependence: weak geometric feature extraction causes alignment errors that directly degrade averaging quality. Traditional iterative approaches (Xu et al., 2012; Chen et al., 2013) suffer from sensitivity to initialization and convergence to local optima. Recent data-driven methods using deep CNNs (X & M, 2020; Jiang et al., 2025) have improved both speed and accuracy, but still struggle with large prediction errors in noisy environments. While specialized model like BOE-ViT (Jiang et al., 2025) have introduced equivariant designs for geometric handling, these approaches remain task-specific and have not demonstrated joint optimization across subtomogram tasks.

Foundation models have been highly effective in advancing biomedical imaging (Zhou et al., 2023; Wu et al., 2023; Chen et al., 2024) and structural biology (Zhou et al., 2025; Shen et al., 2024; Yan et al., 2024). They offer several advantages for subtomogram analysis: (1) learning generalizable representations from large-scale pretraining, thereby reducing dependence on scarce annotated datasets; (2) demonstrating robustness to noise and distribution shifts across domains; and (3) enabling multi-task and transfer learning that can exploit interdependencies among classification, alignment, and averaging. However, foundation models in cryo-ET remain underexplored, primarily due to the lack of large-scale annotated datasets for pretraining, as well as the need for advanced architectures to capture complex 3D geometric transformations and robust noise-handling strategies.

To address these challenges, we present the first foundation model for cryo-ET subtomogram analysis, comprising three key components to tackle the identified limitations. First, to overcome the lack of large-scale annotated datasets, we develop **CryoEngine**, a biophysically informed synthetic data engine that generates diverse samples across structural and postural variations, producing 904k subtomograms from 452 particle classes to provide the unprecedented scale for pretraining. Second, to handle complex 3D geometric transformations, we propose **Adaptive Phase Tokenization-enhanced Vision Transformer** (APT-ViT), which integrates learnable phase selection with spherical steerable convolutions to improve equivariance to both translations and rotations in the SE(3) group, thereby enhancing performance beyond standard ViTs, which is critical for cryo-ET tasks (Jiang et al., 2025). Third, to develop robust noise-handling mechanism, we introduce a **Noise-Resilient Contrastive Learning** (NRCL) strategy, which ensures robust representation in latent space under the severe noise conditions characteristic of cryo-ET data. Extensive experiments on classification, alignment, and averaging tasks across 27 synthetic and real cryo-ET datasets demonstrate that our approach achieves SOTA performance with strong generalization and multi-task capabilities, while remaining robust on challenging complex scenes including filamentous and asymmetric homotrimeric structures.

Our main contributions are as follows:

- We develop **CryoEngine**, a cryo-ET subtomogram simulation data generation framework grounded in biophysical principles at both the imaging and structural levels. It produces a large-scale dataset of 904k volumes from 452 distinct particle classes, enabling diverse category and pose coverage necessary for foundation model pretraining.

- We present the **first foundation model** for cryo-ET subtomogram analysis. As the backbone, we introduce **APT-ViT**, which extends polyphase decomposition to the SE(3) group and incorporates adaptive phase tokenization with a learnable selection network to enhance shift equivariance, while spherical steerable convolutions provide rotation equivariance. In addition, we propose a novel **NRCL** strategy, which leverages noise-aware sampling to stabilize representation learning under high-noise conditions.

- Our framework achieves **SOTA** performance on classification, alignment, and averaging across 27 out-of-distribution synthetic and real cryo-ET datasets, highlighting its strong generalization on challenging structures as a foundation model for subtomogram analysis.

## 2 METHODOLOGY

**Overview.** We present the first foundation model for cryo-ET subtomogram analysis, built upon three key components: a biophysically informed data synthesis engine (CryoEngine, Sec. 2.1), an enhanced-equivariance backbone (APT-ViT, Sec. 2.2), and a noise-robust training strategy (NRCL, Sec. 2.3). These modules respectively enable large-scale pretraining, strengthen the ViT backbone with an equivariant design, and improve resilience to severe noise. The overall pipeline is illustrated in Fig. 1, and the pretrained encoder can serve as a foundation model for diverse downstream subtomogram tasks.

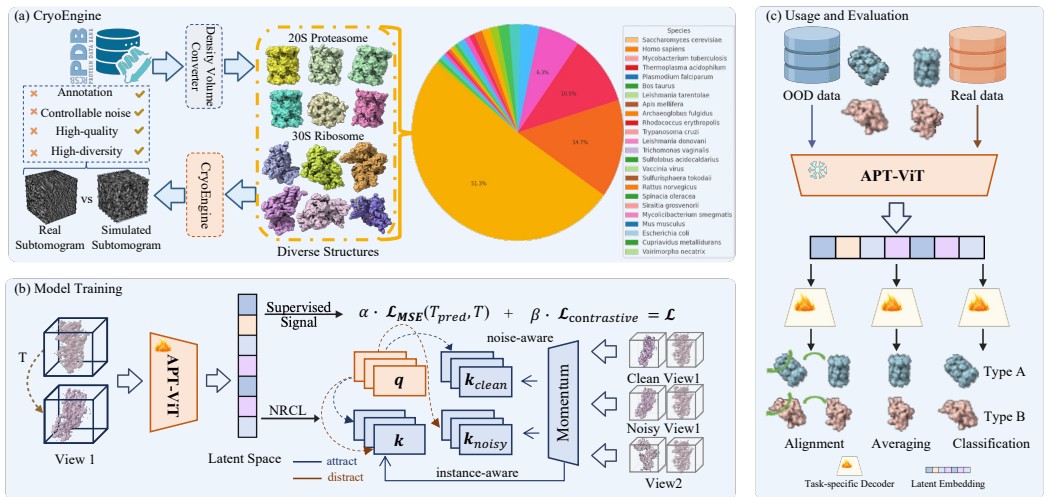

Figure 1: Foundation model overview. (a) CryoEngine. Atomic structures from the RCSB Protein Data Bank are retrieved and converted to density volumes, processed via CryoEngine, and synthesized subtomograms with ground truth and multi-SNR versions. The composition and visualization of the structural data are shown in the diagram. (b) Model training. The input is a subtomogram pair, one derived from the other via a rigid SE(3) transform $T$. Following a MoCo-style paradigm (Pham et al., 2023), the APT-ViT backbone (Sec. 2.2) encodes them into a latent embedding $z$, which is passed into two branches: one supervised by $T$, the other optimized with NRCL (Sec. 2.3). The total loss combines the two branches. (c) Usage and evaluation. The trained encoder serves as a frozen foundation model for downstream tasks, evaluated on both OOD and real data in Sec. 3.

## 2.1 CRYOENGINE

To address the scarcity, low SNR, structural heterogeneity and lack of ground truth of real training data, we develop CryoEngine, a synthetic data engine that systematically simulates large batches of diverse subtomograms. Our engine employs a multi-stage pipeline by replicating each stage of cryo-ET imaging. The engine integrates structural fidelity, pose diversity, and imaging realism in a unified framework tailored for representation learning. As illustrated in Fig. 2, atomic structures are first converted into density volumes and then spatially and rotationally distributed using a density- and orientation-controlled strategy that maximizes utilization while ensuring each extracted subvolume contains a single, isolated particle with uniformly sampled pose coverage across SO(3) for alignment training. The simulated particles are projected into tilt series with realistic microscope geometry and reconstructed via weighted back-projection. From the reconstructed volumes, $32^3$-voxel subtomograms were extracted with position-orientation ground truth metadata preservation and followed by calibrated noise generation. Full simulation specifications are provided in Appendix C.2.

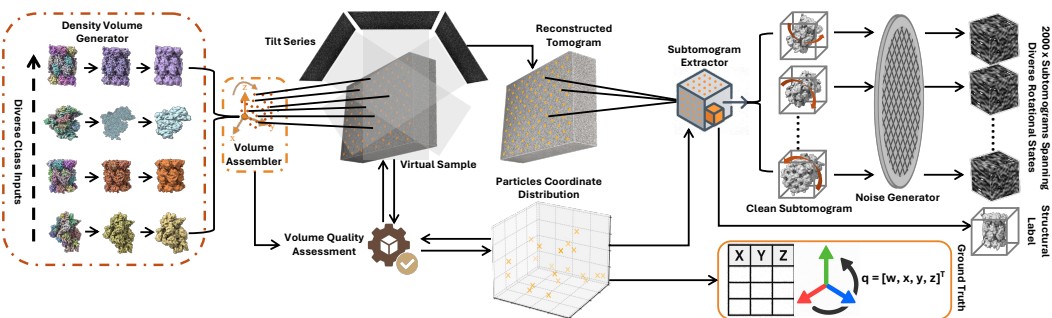

Figure 2: CryoEngine architecture. Diverse atomic structures are converted into density maps, embedded into a virtual sample, projected into a tilt series, and reconstructed into a tomogram. $32^3$ subtomograms are then extracted, augmented with calibrated noise, and paired with ground-truth.

Built on CryoEngine, our synthetic dataset contains two well-characterized complexes, the 20S proteasome core particle and the 30S ribosomal subunit, yielding 452 distinct particle classes in total, and each class yields 2k subtomograms spanning dense and diverse rotational states, replicated across different SNR levels. The dataset captures a wide range of biophysical variability, from large, symmetric assemblies to compact, asymmetric folds, and reflects the compositional diversity observed in situ. Details and visualizations about the 904k subtomograms can be found in Appendix C.2.

## 2.2 APT-ViT

**Overall Architecture.** This work introduces APT-ViT, which serves as the backbone of the foundation model for cryo-ET subtomogram analysis, as illustrated in Fig. 3. The core innovation of APT-ViT lies in its novel *adaptive phase tokenization* mechanism that enhances SE(3) equivariance properties of ViTs. The architecture integrates APT into a standard ViT backbone with task-specific output heads, with the complete APT workflow provided in Algorithm 1.

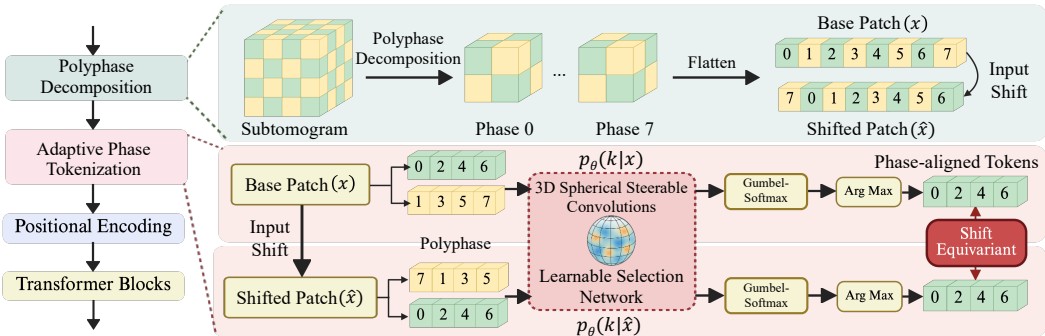

Figure 3: Overview of APT-ViT architecture. Input subtomograms undergo polyphase decomposition to generate multiple phase components with different spatial offsets. A learnable selection network based on 3D spherical steerable convolutions evaluates all phase components and produces selection probabilities to select the optimal phase-aligned tokens. The selected tokens are then processed by transformer blocks to produce refined embeddings for downstream subtomogram analysis tasks.

**Polyphase Decomposition in 3D Space.** Inspired by recent work on shift-equivariant ViTs for 2D images (Ding & Smith, 2023; Rojas-Gomez et al., 2024) and equivariant CNN techniques (Chaman & Dokmanic, 2021; Rojas-Gomez et al., 2022), we reformulate volume tokenization using polyphase decomposition and extend it to 3D space to enable adaptive phase selection for equivariant token generation. For an input volume $\mathbf{X} \in \mathbb{R}^{B \times C \times D \times H \times W}$ and a patch size $\mathbf{s} = (s_D, s_H, s_W)$, the polyphase decomposition operator $\Psi$ generates a set of phase components:

$$\Psi(\mathbf{X})_{(p,q,r)} = \left\{ \mathbf{X}_{:,:,i \cdot s_D + p, j \cdot s_H + q, k \cdot s_W + r} \mid i, j, k \in \mathbb{Z}_{\geq 0} \right\} \quad (1)$$

where $(p, q, r)$ is the phase offset, with $p \in \{0, \ldots, s_D - 1\}$, and similarly for $q$ and $r$. This operation partitions the input into $s_D \times s_H \times s_W$ sets of non-overlapping patches, each corresponding to a different spatial offset.

**Adaptive Phase Tokenization.** Instead of using a fixed phase, which is sensitive to input shifts, our APT module learns to select the optimal phase component that provides a consistent reference frame under SE(3) transformations. The output of the module is the optimally selected phase:

$$\text{APT}(\mathbf{X}) = \Psi(\mathbf{X})_{(p^*, q^*, r^*)} \quad (2)$$

The optimal phase $(p^*, q^*, r^*)$ is determined by maximizing a selection probability $p_\theta(k|\mathbf{X})$, which is modeled by a learnable selection network $f_\theta$:

$$p_\theta(k = (p, q, r)|\mathbf{X}) \triangleq \frac{\exp[f_\theta(\Psi(\mathbf{X})_{(p,q,r)})]}{\sum_{(p',q',r')} \exp[f_\theta(\Psi(\mathbf{X})_{(p',q',r')})]} \quad (3)$$

To ensure the selection process itself is equivariant, we design $f_\theta$ using rotation-equivariant spherical steerable convolutions (Weiler et al., 2018a), followed by global average pooling:

$$f_\theta(\Psi(\mathbf{X})_{(p,q,r)}) = \frac{1}{|V|} \sum_{v \in V} \tilde{f}_\theta(\Psi(\mathbf{X})_{(p,q,r)})[v] \quad (4)$$

---

**Algorithm 1:** Adaptive Phase Tokenization

---

**Input:** Subtomogram volume $\mathbf{X} \in \mathbb{R}^{B \times C \times D \times H \times W}$, patch size $\mathbf{s} = (s_D, s_H, s_W)$, APT parameters $\theta$, temperature $t$, mode flag (training/inference)

**Output:** Optimally phased tokens $\mathbf{Z} \in \mathbb{R}^{B \times C \times \lfloor D/s_D \rfloor \times \lfloor H/s_H \rfloor \times \lfloor W/s_W \rfloor}$

1 Decompose $\mathbf{X}$ into polyphase components $\{\Psi(\mathbf{X})_{(p,q,r)}\}_{p,q,r=0}^{s_D-1, s_H-1, s_W-1}$;

2 **for** $(p,q,r) \in \{0, ..., s_D - 1\} \times \{0, ..., s_H - 1\} \times \{0, ..., s_W - 1\}$ **do**

3      Compute maximum spherical harmonic degree $J_{\max}(r) \leftarrow \lfloor \frac{\pi r}{\Delta x} \rfloor$;

4      Construct steerable filter $\tilde{f}_\theta(\mathbf{x}) \leftarrow \sum_{J=0}^{J_{\max}} \sum_{m=-J}^{J} R_J(||\mathbf{x}||) \cdot Y_J^m \left( \frac{\mathbf{x}}{||\mathbf{x}||} \right) \cdot w_{J,m}$;

5      Extract features $h_{(p,q,r)} \leftarrow \tilde{f}_\theta(\Psi(\mathbf{X})_{(p,q,r)})$;

6      Apply global pooling $\ell_{(p,q,r)} \leftarrow \frac{1}{|V|} \sum_{v \in V} h_{(p,q,r)}[v]$;

7 Compute selection probabilities $p_{(p,q,r)} \leftarrow \frac{\exp(\ell_{(p,q,r)})}{\sum_{(p',q',r')} \exp(\ell_{(p',q',r')})}$;

8 **if** *training* **then**

     // Training mode

9      Sample relaxed probabilities $\tilde{p}_{(p,q,r)} \sim \text{Gumbel-Softmax}(p(p,q,r), t)$;

10      Select optimal phase offset $(p^*, q^*, r^*) \leftarrow \arg\max (p,q,r) \tilde{p}_{(p,q,r)}$;

11 **else**

     // Inference mode

12      Deterministically select phase offset $(p^*, q^*, r^*) \leftarrow \arg\max (p,q,r) p(p,q,r)$;

13 Extract tokens from selected polyphase component $\mathbf{Z} \leftarrow \Psi(\mathbf{X})_{(p^*,q^*,r^*)}$;

---

where $\tilde{f}_\theta$ is a spherical steerable convolutional filter and $V$ is the set of spatial locations. The filter is defined as:

$$\tilde{f}_\theta(\mathbf{x}) = \sum_{J=0}^{J_{\max}} \sum_{m=-J}^{J} R_J(||\mathbf{x}||) \cdot Y_J^m \left( \frac{\mathbf{x}}{||\mathbf{x}||} \right) \cdot w_{J,m} \quad (5)$$

Here, $Y_J^m$ are the spherical harmonics, $R_J$ are learnable radial functions, and $w_{J,m}$ are learnable weights. This construction guarantees that the feature extraction is rotation-equivariant.

While the selection network outputs probabilities over phase components, the arg max for choosing the optimal phase is non-differentiable. We employ Gumbel-Softmax (Jang et al., 2016) for differentiable categorical sampling:

$$\tilde{p}_{(p,q,r)} \sim \text{Gumbel-Softmax}(p_\theta(k = (p,q,r)|\mathbf{X}), t) \quad (6)$$

where $t$ is the temperature parameter. During forward pass, we apply arg max to obtain discrete phase indices:

$$(p^*, q^*, r^*) = \arg\max_{(p,q,r)} \tilde{p}_{(p,q,r)} \quad (7)$$

This enables gradient flow via Gumbel-Softmax during backpropagation while maintaining discrete selection. The selected polyphase component $\Psi(\mathbf{X})_{(p^*,q^*,r^*)}$ forms the token representation $\mathbf{Z}$, which passes through the patch embedding layer and shift-equivariant positional encoding (Jiang et al., 2025) before entering the ViT blocks. Theoretical analysis of the equivariance properties of the core APT mechanism is provided in Appendix B.

### 2.3 Noise-resilient Contrastive Learning Strategy

In this work, we propose NRCL, a contrastive learning framework tailored for cryo-ET subtomograms. A key feature of our approach is a *noise-aware sampling* strategy that constructs positive and negative pairs. This strategy follows the contrastive principle of pulling similar embeddings closer while pushing dissimilar ones apart in the latent space. Building on this design, we further introduce a contrastive loss that enhances robustness against severe noise, which can be formulated as:

$$\mathcal{L}_{\text{contrastive}} = \mathcal{L}_{\text{instance}} + \mathcal{L}_{\text{noise}} \quad (8)$$

We define the inputs as pairs of subtomograms $\mathbf{I} = (X_1, X)$ where $X_1 = TX$ represents a transformed version of the original subtomogram $X \in \mathbb{R}^{B \times C \times D \times H \times W}$, with $T$ denoting spatial transformations

applied to the 3D volume. The whole training paradigm follows the MoCo v3-style, with one online base encoder and one momentum encoder which delays the update for a robust reference. For the workflow of the whole strategy, please see Appendix D.

**Instance Discrimination.** In instance discrimination, similar to the common practice in contrastive learning where positive samples are constructed using different augmentations of the same instance(Pham et al., 2023; Grill et al., 2020; Chen et al., 2020), we include $\mathbf{I}^+ = (X_2, X)$ as a positive sample where $X_2 = T'X$. The negative samples $\mathbf{I}^-$ are the other samples in the same mini-batch.

Inspired by RINCE(Chuang et al., 2022), we introduce a symmetric exponential loss to alleviate the effect of inappropriate negative samples in the mini-batch of size B and stabilize the training. Let $f$ denotes the encoder, $\tau$ denotes the temperature and $c$ denotes the parameter used to control the weight of positive and negative samples, the equation is:

$$\mathcal{L}_{\text{sym}} = -\frac{e^{c \cdot s^+}}{c} + \frac{1}{c} \log(e^{c \cdot s^+} + e^{c \cdot s^-}), \text{ where } s^{\pm} = \frac{z^{\top} z^{\pm}}{\tau}, z = f(\mathbf{I}), z^+ = f(\mathbf{I}^+), \text{and } z^- = f(\mathbf{I}^-) \quad (9)$$

To further enhance the instance discrimination capability and robustness to noise, we adopt sinkhorn wasserstein distance as an additional term. The Sinkhorn-Wasserstein distance is defined with a cost matrix $\mathbf{C}$ based on squared Euclidean pairwise distances and uniform marginal distributions.

$$\mathcal{L}_{\text{wass}} = \text{Sinkhorn}_{\varepsilon}(z, z^+) = \min_{\mathbf{P}} \langle \mathbf{C}^p, \mathbf{P} \rangle, \text{ where } \sum_j \mathbf{P}_{ij} = \frac{1}{B}, \sum_i \mathbf{P}_{ij} = \frac{1}{B} \quad (10)$$

Let $\lambda_W$ denotes the weight of $\mathcal{L}_{\text{wass}}$, the complete equation for instance discrimination is:

$$\mathcal{L}_{\text{instance}} = \mathcal{L}_{\text{sym}} + \lambda_W \cdot \mathcal{L}_{\text{wass}} \quad (11)$$

**Noise-aware Sampling.** As in Fig. 1b, the noise-aware sampling would define the negative sample as an extremely noisy version of the input, while the positive sample be a clean version. Since CryoEngine is able to generate subtomograms at controllable noise levels, the negative and positive sample pairs could be readily obtained by changing the parameter for the noise generator in Fig. 2. This makes it feasible to define positive and negative pairs not only through the commonly practiced instance discrimination, but also by explicitly leveraging noise perturbations. The intuition of NRCL is to drag the latent embedding of the input closer to that of the clean ground truth subtomograms while pushing it away from the noisy ones. Since each input corresponds to one clean positive and one noisy negative sample under full control, so balancing is not required. Thus, we only adopt the classic InfoNCE loss. Note that $z^+$ and $z^-$ are computed from positive and negative samples generated via noise-aware sampling here.

$$\mathcal{L}_{\text{noise}} = \mathcal{L}_{\text{InfoNCE}} = -\log \frac{\exp(z^{\top} z^+/\tau)}{\exp(z^{\top} z^+/\tau) + \exp(z^{\top} z^-/\tau)} \quad (12)$$

## 3 EXPERIMENTS

### 3.1 EXPERIMENTAL SETUP

**CryoEngine-generated Data for Pre-training.** As stated in Sec. 2.1, the dataset for pretraining is constructed by CryoEngine. A total of 904k pairs of subtomograms in 452 different PDB IDs are used for the pretrain process. The subtomograms have a resolution of 10Å, each with a size of 32×32×32. The selected structures are mainly proteasomes and ribosomes, which are among the most commonly analyzed categories in cryo-ET imaging by biologists. We emphasize that all test datasets used in downstream analyses are out-of-distribution (OOD) and highly distinct from the pretraining families generated by CryoEngine. All the training-based baseline models undergo the same finetune procedure as our method in each downstream task.

**Pre-training Details.** We explore our APT-ViT-based model performance pre-trained with the noise-resilient contrastive learning(NRCL) strategy stated in Sec. 2. The encoder is an APT-ViT composed of two embedding modules equipped with the APT design to process each subtomogram independently in the input pair, a simple module for embedding fusion, and four transformer blocks with embedding dimension 120. Training is conducted for 200 epochs using a batch size of 2048. The learning rate follows the square-root scaling rule. The weight decay is fixed at $1 \times 10^{-4}$. For the contrastive learning framework, we use a temperature parameter of 0.1 and a momentum coefficient of 0.99 for the momentum encoder.

Table 1: Classification accuracy↑ (%) for overall datasets across different SNR levels. All encoders of ours and the baselines are kept **frozen**. Best results are bold, second best underlined.

| Pre-train Data | Method | Model | SNR 0.1 | SNR 0.05 | SNR 0.03 | SNR 0.01 |
|---|---|---|---|---|---|---|
| ImageNet | Supervised | ConvNeXt v1 | 32.52 | 27.85 | 26.07 | 22.08 |
| | | ConvNeXt v2 | 20.37 | 20.00 | 20.03 | 20.10 |
| | | ViT-B | 26.64 | 23.14 | 21.90 | 20.52 |
| | | PVT v2 | 24.35 | 21.05 | 20.92 | 20.00 |
| | | SwinViT-S | 51.76 | 33.12 | 27.22 | 22.00 |
| | | SwinViT-B | 51.22 | 35.18 | 27.98 | 20.06 |
| | Self-supervised | Moco v3 | 41.00 | 29.84 | 25.28 | 21.24 |
| | | MAE | 44.70 | 31.14 | 27.82 | 23.76 |
| LVD-142M | Self-supervised | DINO v2 | 32.41 | 25.27 | 23.34 | 21.22 |
| CryoEngine (**Ours**) | Supervised | ViT-B | 31.38 | 23.75 | 24.15 | 19.85 |
| | MAE | ViT-B | 56.93 | 41.13 | 30.65 | 22.40 |
| | NRCL(**Ours**) | APT-ViT (**Ours**) | **67.42** | **53.13** | **40.10** | **27.50** |

## 3.2 CLASSIFICATION

**Finetune Implementation.** We attach a lightweight classification head, which is a 3-layer MLP, to the frozen encoder for the classification task. The test dataset is a 5-class cryo-ET benchmark dataset (Zeng et al., 2021a). For each test dataset corresponding to a specific SNR level, we fine-tune the head for 40 epochs using a combination of SNR 100 samples and 10% of the samples from the target low-SNR data, which is split into training, validation, and test sets using a 1:1:8 ratio. More experiment settings and introduction of datasets and baselines are provided in Appendix E.

**Results.** To evaluate downstream classification, we compare against a diverse set of classical and SOTA foundation models, covering both supervised and self-supervised paradigms, and including ViT-based as well as non-ViT architectures. As shown in Table 1, our method consistently outperforms all baselines across SNR levels, with particularly large gains at SNR 0.05 and 0.03 (around 50%), which are typical conditions in cryo-ET. Moreover, we provide additional experimental results across five datasets with varying SNRs in Tables 7-11 in Appendix E.3. These results highlight the complementary roles of CryoEngine, APT-ViT and NRCL in driving the overall improvement.

## 3.3 ALIGNMENT

**Finetune Implementation.** The alignment fine-tuning phase employs reduced augmentation ranges. This refined augmentation scope allows the model to specialize on subtle misalignments that are more representative of real-world scenarios where large transformations are less common. For each fine-tuning epoch, we apply 5 augmentations per sample with a batch size 4. The transformation head is a 3-layer MLP. The dataset used for alignment testing is the same as the one used for classification. For detailed implementation and introduction of baselines, see Appendix F.

**Results.** We benchmark against traditional algorithms, CNN-based models, ViT-based methods, and equivariant networks. To validate the effectiveness of our equivariant design, we include three representative point-cloud–based models—SE(3)-Transformer (Fuchs et al., 2020a), ConDor (Sajnani et al., 2022), and Equi-Pose (Li et al., 2021)—as well as the equivariance-enhanced ViT baseline, BOE-ViT (Jiang et al., 2025). As shown in Table 2, APT-ViT achieves the lowest alignment errors across all SNRs, with strong robustness to noise. Moreover, additional results across diverse macromolecular structures with varying SNRs are provided in Appendix F.3 (Tables 13-17). At the extremely low SNR of 0.01, our method substantially reduces both rotation and translation errors compared to traditional and equivariant baselines, and achieves over 30% lower translation error than BOE-ViT, underscoring the superiority of the APT-ViT design.

**Generalization to Challenging Structures.** To assess transfer beyond the globular complexes used during pre-training, we evaluate our model on filamentous and asymmetric homotrimeric structures. The results in Table 19 in Appendix F.5 show strong zero-shot transfer to these complex scenes and challenging sturctures in subtomogram alignment, indicating robust generalization beyond the pretraining distribution.

(a) Visualization of the aligned particles.

(b) 2D visualization of subtomograms.

Figure 4: Visualization of alignment results, showing improved structural consistency in both 3D particles (a) and 2D subtomogram slices (b).

Table 2: Subtomogram alignment accuracy at different SNR levels. Values are mean↓ ± std of rotation and translation errors. Best results are bold, second best underlined.

| Method | SNR 0.1 | | SNR 0.05 | | SNR 0.03 | | SNR 0.01 | |
| --- | --- | --- | --- | --- | --- | --- | --- | --- |
| | Rotation | Translation | Rotation | Translation | Rotation | Translation | Rotation | Translation |
| H-T align | 1.22±1.07 | 4.76±4.56 | 1.93±0.98 | 7.26±4.77 | 2.22±0.77 | 8.86±4.72 | 2.38±0.57 | 11.33±5.02 |
| F-A align | 1.34±1.13 | 5.39±4.90 | 1.95±0.98 | 7.54±4.94 | 2.22±0.77 | 8.99±4.81 | 2.38±0.57 | 11.32±4.92 |
| GumNet-MP | 1.30±0.79 | 4.93±3.36 | 1.44±0.79 | 5.46±3.88 | 1.53±0.78 | 5.96±3.34 | 1.67±0.77 | 7.28±3.38 |
| GumNet-AP | 1.09±0.73 | 4.20±2.96 | 1.30±0.77 | 5.00±3.15 | 1.45±0.77 | 5.70±3.25 | 1.65±0.78 | 7.18±3.35 |
| GumNet-SC | 1.16±0.77 | 4.41±3.23 | 1.36±0.79 | 5.13±3.34 | 1.48±0.78 | 5.75±3.34 | 1.67±0.77 | 7.24±3.46 |
| GumNet | 0.62±0.69 | 2.41±2.61 | 0.80±0.77 | 3.20±2.78 | 1.13±0.75 | 4.09±2.75 | 1.50±0.78 | 6.78±4.22 |
| JimNet | 0.51±0.62 | 2.12±2.47 | 0.80±0.73 | 3.20±3.02 | 1.02±0.75 | 4.12±3.12 | 1.58±0.77 | 6.78±3.44 |
| SE(3)-Transformer | 1.77±0.47 | 5.12±0.67 | 1.53±0.47 | 4.11±0.58 | 1.68±0.35 | 5.25±0.63 | 1.82±0.50 | 4.31±0.65 |
| ConDor | 6.73±1.63 | 6.61±1.39 | 6.57±1.59 | 6.47±1.46 | 6.79±1.39 | 6.67±1.57 | 6.88±1.32 | 6.78±1.61 |
| Equi-Pose | 4.40±2.13 | 3.96±2.10 | 6.00±2.25 | 4.36±2.12 | 6.84±2.20 | 4.56±2.41 | 5.74±2.37 | 5.04±2.49 |
| BOE-ViT | 0.33±0.15 | 2.58±0.93 | 0.34±0.15 | 2.45±0.87 | 0.34±0.15 | 2.50±0.89 | 0.34±0.15 | 2.54±0.91 |
| **Ours** | **0.25±0.08** | **2.00±0.80** | **0.25±0.08** | **1.95±0.85** | **0.25±0.08** | **2.00±0.80** | **0.25±0.09** | **2.02±0.82** |

### 3.4 AVERAGING

**Finetune Implementation.** To avoid structural bias, we evaluate averaging using an independent half-reconstruction strategy: each half-dataset bootstraps its own reference frame, and the model aligns all subtomograms within that half accordingly. This setup follows the foundation model paradigm—pretraining provides equivariant features and transformation predictors from self-alignment, while finetuning adapts them to inter-particle alignment for consensus averaging. The two half-maps are only rigidly registered to establish a common coordinate frame for resolution estimation, ensuring unbiased evaluation. We benchmarked averaging on the aboved simulation datasets and four real datasets against five task-specific baselines, with details provided in Appendix F.

**Results.** As shown in Table 3, our method achieves SOTA resolutions across all datasets. Visualizations of iterative subtomogram averaging on 4 real datasets are provided in Fig. 12 in Appendix G.3. These results demonstrate that the foundation model is generalizable and transferable to real-world data.

### 3.5 MECHANISM ANALYSIS AND ABLATION STUDY

**Separability of the Latent Embeddings.** To further eliminate the possible effect of classification head and evaluate the effectiveness of our method, we show the comparison of KNN-accuracy between our model and the baselines after pretraining. Table 21 in Appendix H.2 clearly shows that

Table 3: Subtomogram averaging results across different SNR levels. Each cell reports the achieved resolution↓(nm). Best results are bold, second best underlined.

| Method | SNR 0.1 | SNR 0.05 | SNR 0.03 | SNR 0.01 | 80S | TMV | Aldolase | Insulin |
|---|---|---|---|---|---|---|---|---|
| H-T align | 2.89 | 3.79 | 4.92 | 4.41 | 3.05 | 2.23 | 2.34 | 1.90 |
| F-A align | 2.78 | 4.36 | 3.81 | 4.53 | 2.77 | 2.52 | 3.13 | 2.18 |
| Gum-Net | 2.78 | 2.95 | 4.01 | 4.22 | 2.73 | 2.16 | 1.97 | 1.77 |
| BOE-ViT | 2.57 | 2.42 | 3.20 | 3.49 | 2.42 | 1.98 | 1.45 | 1.71 |
| **Ours** | **2.56** | **2.40** | **2.95** | **3.20** | **1.21** | **1.98** | **0.97** | **1.14** |

Table 4: Impact of APT-ViT and NRCL on classification. Each cell shows accuracy↑(%) at different SNR. We study the effect of NRCL, APT-ViT, and NRCL loss terms. If not specified, the encoder is kept **frozen** during finetune. Best results are bold, second best underlined.

| Metric | Architecture | SNR 0.1 | SNR 0.05 | SNR 0.03 | SNR 0.01 | Average |
|---|---|---|---|---|---|---|
| BYOL | APT-ViT | 44.65 | 34.45 | 30.03 | 22.33 | 32.87 |
| No Pre-train | ViT-B | 20.00 | 20.00 | 20.00 | 20.00 | 20.00 |
| No Pre-train | APT-ViT | 52.60 | 29.24 | 21.22 | 20.00 | 30.77 |
| NRCL w/o $\mathcal{L}_{sym}$ | APT-ViT | 53.53 | 41.84 | 32.65 | 23.81 | 37.96 |
| NRCL w/o $\mathcal{L}_{InfoNCE}$ | APT-ViT | 63.43 | 52.56 | **41.72** | **27.78** | 46.38 |
| NRCL(**Ours**) | APT-ViT | **67.42** | **53.13** | 40.10 | 27.50 | **47.04** |

Table 5: Impact of APT-ViT components on alignment. Each cell reports the mean↓ and standard deviation of the rotation error and translation error.

| Model | Method | SNR 0.1 | | SNR 0.05 | | SNR 0.03 | | SNR 0.01 | |
|---|---|---|---|---|---|---|---|---|---|
| | | Rotation | Translation | Rotation | Translation | Rotation | Translation | Rotation | Translation |
| ViT-based | w/o APT | 0.66±0.30 | 6.14±1.78 | 0.66±0.30 | 6.14±1.76 | 0.65±0.30 | 6.15±1.76 | 0.66±0.30 | 6.20±1.78 |
| ViT-based | w/o APS | 0.34±0.15 | 2.63±0.87 | 0.34±0.15 | 2.62±0.87 | 0.35±0.15 | 2.62±0.88 | 0.35±0.15 | 2.62±0.91 |
| ViT-based | w/o Steerable | 0.25±0.09 | 2.95±1.37 | 0.26±0.10 | 3.09±1.42 | 0.26±0.09 | 3.22±1.49 | 0.26±0.09 | 3.42±1.56 |
| **ViT-based** | **Ours** | **0.25±0.08** | **2.00±0.80** | **0.25±0.08** | **1.95±0.85** | **0.25±0.08** | **2.00±0.80** | **0.25±0.09** | **2.02±0.82** |

our model outperforms the baselines, indicating that our method effectively enables the model to learn meaningful representations under low-SNR conditions. We also provide the results of linear probing on the latent space in Appendix H.2, indicating that the discriminative power comes from the learned latent embeddings themselves, rather than from the classifier head. It is further validated in Appendix I, where Grad-CAM visualizations (Selvaraju et al., 2017) show that the model consistently attends to structurally meaningful regions in subtomograms.

**Impact of NRCL & APT-ViT on Classification.** We conducted ablation studies by comparing different training strategies, architectures, and variations of contrastive loss function in NRCL. The results in Table 4 demonstrate that our design in Sec. 2 works synergistically to achieve optimal performance on average. Detailed implementation and additional results are provided in Appendix H.3 and Appendix H.5.

**Impact of Rotation Representation for Alignment.** To better explore geometric representations for alignment tasks, we investigated common rotation parameterizations including Euler angles, $\mathbb{R}^6$ with Gram-Schmidt orthonormalization (Zhou et al., 2019), and $\mathbb{R}^9$ with singular value decomposition (Levinson et al., 2020) as representations in the SO(3) group, which is detailed in Appendix B.4. Our experimental results in Table 20 in Appendix H.1 demonstrate that for cryo-ET subtomogram alignment, the Euler angle representation achieves superior performance.

**Impact of APT-ViT Components on Alignment.** As shown in Table 5, the ablation results demonstrate that each APT component is critical for equivariance. Removing APT causes severe translation degradation, confirming standard ViT's failure under spatial transformations. The adaptive phase selection is essential for translation equivariance, while spherical steerable convolutions are crucial for robust spatial alignment, as their removal significantly impairs translation accuracy.

**Impact of Box Size on Alignment and Classification** The input box size trades off spatial context against efficiency and noise sensitivity. We therefore transfer from $32^3$ to $64^3$ inputs for both classification and alignment by reusing $32^3$ pretrained weights and applying identical fine-tuning,

rather than pretraining at $64^3$ since 3D pretraining scales cubically. As shown in Table 12 in Appendix E.4, we report classification recall (%) across SNR levels under varying box sizes and resolutions. For alignment, Table 18 in Appendix F.4 shows that our method consistently outperforms BOE-ViT across all SNR levels and at both box sizes, while maintaining strong rotation accuracy at $64^3$ subtomograms.

**Parameter Exploration.** Detailed results under different hyperparameter settings are in Appendix H.4.

## 4 CONCLUSION

In this work, we present the first foundation model for cryo-ET subtomogram analysis, integrating CryoEngine, APT-ViT, and NRCL. Together, these components enable large-scale data generation, enhanced equivariance, and robust representation learning under noise. Our model achieves SOTA performance across classification, alignment, and averaging tasks on 27 out-of-distribution datasets, demonstrating multi-task capability and strong generalization in challenging structures. By bridging the gap in cryo-ET pretraining, this work provides meaningful insights for building cryo-ET foundation models, advancing downstream analysis, and facilitating real-world structural discovery.

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

## A    RELATED WORK

**Foundation Models in Life Science.** Pretraining and foundation models have been instrumental in advancing biomedical imaging and structure biology. In medical imaging and pathology, supervised pretraining on labeled datasets such as RadImageNet (Mei et al., 2022) and KimiaNet (Riasatian et al., 2021), together with large-scale self-supervised or cross-modal approaches like RETFound (Zhou et al., 2023), UniRad (Wu et al., 2023), and UNI (Chen et al., 2024), has enabled broad transfer across tasks including disease classification, lesion segmentation, and retrieval. In structural biology, foundation models are beginning to emerge: CryoFM (Zhou et al., 2025) and DRACO (Shen et al., 2024) show how flow-based or denoising-reconstruction pretraining can generalize across cryo-EM densities and support downstream micrograph analysis, while CryoIEF (Yan et al., 2024) builds on this paradigm to separate particles from different structures and cluster them by pose. However, no foundation model has yet been developed for cryo-ET subtomograms, largely due to the scarcity of annotated data and the absence of tailored architectures. To address this gap, we propose the first foundation model for cryo-ET subtomograms, offering more efficient and robust tools for life sciences research.

**Cryo-ET Subtomogram Analysis.**  Cryo-ET enables in situ 3D imaging of macromolecules, facilitating structural studies under native conditions (Turk & Baumeister, 2020; Zhang et al., 2023). A subtomogram is a small 3D volume extracted around a macromolecule from a tomogram, and its analysis is critical for structural recovery in cryo-ET (Chen et al., 2019; Schur, 2019). Subtomogram analysis involves *geometric* tasks like alignment, which require transformation-sensitive features (Bauer et al., 2009; Střelák et al., 2020; Jiang et al., 2023), and *semantic* tasks such as classification and clustering, which favor invariant representations under high noise (Che et al., 2017; Granberry et al., 2023). CNN-based classifiers (Che et al., 2017) have been extended to open-set (Zeng et al., 2021b), few-shot (Li et al., 2020), and domain-adaptive settings (Bandyopadhyay et al., 2021), while unsupervised clustering aims to discover structures without labels (Chen et al., 2014; Zeng et al., 2023; Martinez-Sanchez et al., 2020). Currently, most approaches are task-specific models with limited generalization capability. Our work represents the first foundation model for cryo-ET subtomograms, leveraging large-scale pretraining to enhance transferability across diverse analysis tasks.

**Group Equivariant Neural Networks.** Previous studies show that CNNs lack shift equivariance due to pooling disrupting translational symmetry (Azulay & Weiss, 2019; Krizhevsky et al., 2012; He et al., 2016), motivating the development of G-CNNs to restore equivariance by extending translation to broader symmetry groups (Cohen & Welling, 2016) and to continuous, more complex transformations (Weiler et al., 2018b; Kondor & Trivedi, 2018; Tu et al., 2024). In parallel, Vision Transformers (ViTs) emerged as strong CNN alternatives for visual recognition, leveraging self-attention to model global dependencies (Touvron et al., 2021; Islam, 2022; Khan et al., 2022; Parvaiz et al., 2023). Recent shift-equivariant ViTs with adaptive tokenization and positional encodings (Rojas-Gomez et al., 2024; Chu et al., 2021) have extended SE(3)-equivariant Transformers originally applied to point clouds (Wang & Jörnsten, 2024; Fuchs et al., 2020b; Gao et al., 2024; Chatzipantazis et al., 2022; Li et al., 2021; Thomas et al., 2018) to image tasks by addressing challenges in equivariant patch design. However, existing equivariant models are primarily designed for 2D images or point clouds, and do not directly address the unique challenges of 3D subtomograms. Therefore, specialized tokenization strategies are needed to enhance ViTs with equivariance to both translations and rotations.

**Self-Supervised Learning.** Self-supervised learning (SSL) has emerged as a powerful strategy for training on large-scale unlabeled datasets (Chen et al., 2021; Gui et al., 2024; Balestriero et al., 2023). Contrastive learning approaches (Chen et al., 2020; Caron et al., 2020; He et al., 2020; Grill et al., 2020; Chen et al., 2021; Oquab et al., 2024) leverage encoder networks to learn robust representations by contrasting positive and negative pairs from augmented views. Recent works have explored learning robust representations under heavy noise conditions (Chuang et al., 2022; Zhou et al., 2024; Jiang et al., 2024). In structural biology, SSL has enhanced cryo-EM micrograph denoising and classification (Zhou et al., 2025; Shen et al., 2024), and enabled effective subtomogram representation learning in cryo-ET for tasks under limited supervision (Campanella et al., 2025; Gupta et al., 2022; Stojanovska et al., 2024). However, self-supervised learning methods remain underexplored in the cryo-ET domain, and standard contrastive approaches often suffer from feature collapse under the extreme noise conditions intrinsic to subtomograms. Therefore, designing noise-resilient contrastive learning strategies is essential to stabilize representation learning in cryo-ET.

# B    Theoretical Analysis

## B.1    Preliminaries

This section provides the necessary mathematical background for analyzing equivariance within our framework. These preliminaries lay the foundation for understanding the model design, particularly its behavior under geometric transformations such as rotations and translations. We also include a discussion of the equivariance properties of conventional ViTs to contrast with our proposed approach.

**Notation.**    Let $\mathbf{X} \in \mathbb{R}^{B \times C \times D \times H \times W}$ denote a batch of subtomogram volumes, where $B$ is batch size, $C$ is number of channels, $D, H, W$ are spatial dimensions (depth, height, width). Let the patch size be $\mathbf{s} = (s_D, s_H, s_W)$, where $s_D$, $s_H$, and $s_W$ are the patch sizes along each spatial axis.

**Definition 1** (Group Actions and $G$-sets). *Let $G$ be a group acting on sets $X$ and $Y$ via*

$$\alpha : G \times X \to X, \qquad \beta : G \times Y \to Y. \tag{13}$$

*The sets $X$ and $Y$ are called $G$-sets under these actions.*

**Definition 2** ($G$-Equivariance). *A map $f : X \to Y$ is $G$-equivariant if, for all $g \in G$ and $x \in X$,*

$$f\big(\alpha(g,x)\big) = \beta\big(g,\, f(x)\big). \tag{14}$$

**Definition 3** ($G$-Invariance). *If $\beta$ is the identity on $Y$, then $f$ is $G$-invariant, i.e.,*

$$f\big(\alpha(g,x)\big) = f(x), \qquad \forall\, g \in G,\ x \in X. \tag{15}$$

**Definition 4** (Polyphase Decomposition). *The polyphase decomposition of $\mathbf{X}$ is defined as*

$$\Psi(\mathbf{X})_{(p,q,r)} = \big\{ \mathbf{X}_{:,:,\, i\, s_D + p,\, j\, s_H + q,\, k\, s_W + r} \ \big| \ i, j, k \in \mathbb{Z}_{\geq 0} \big\}, \tag{16}$$

*where*

$$p \in \{0, \dots, s_D - 1\}, \quad q \in \{0, \dots, s_H - 1\}, \quad r \in \{0, \dots, s_W - 1\}. \tag{17}$$

**Equivariance of ViT Modules.**    The Vision Transformer (ViT) architecture comprises a patch embedding layer, positional encoding, transformer blocks, and MLP layers. As noted by Ding et al.(Ding et al., 2023), the patch embedding layer lacks both shift and rotation equivariance due to the downsampling operation, which disrupts spatial continuity. Furthermore, commonly used positional encoding schemes—whether absolute (Dosovitskiy et al., 2021) or relative (Liu et al., 2021; 2022a)—are inherently non-equivariant to translations or rotations. While normalization, global self-attention, and MLP layers preserve shift equivariance, achieving rotation equivariance requires deliberate architectural modifications (Assaad et al., 2022; Kundu & Kondor, 2024).

## B.2    Translation Equivariance of APT-ViT module

To ensure reliable 3D subtomogram alignment, it is crucial that token representations remain consistent under spatial translations. APT achieves this by adaptively selecting phase components in a translation-equivariant manner. The following lemmas and theorem formally establish this property, showing that a translated input yields a correspondingly translated output token.

**Lemma 1** (Polyphase Decomposition Equivariance). *Let $\mathcal{P}$ be the polyphase anchoring operator and $T_{\mathbf{g}}$ denote a translation operator that shifts $\mathbf{X}$ spatially by $\mathbf{g} = (g_D, g_H, g_W)$. Then, there exists a translation $\mathbf{g}' = (g'_D, g'_H, g'_W)$ such that*

$$\mathcal{P}\big(T_{\mathbf{g}}\mathbf{X}\big) = T_{\mathbf{g}'}\big(\mathcal{P}(\mathbf{X})\big), \tag{18}$$

*where $T_{\mathbf{g}}$ is translation.*

*Proof.* By definition, the polyphase decomposition divides $\mathbf{X}$ into components:

$$\mathbf{X}^{(p,q,r)} = \left\{ \mathbf{X}_{:,:,i \cdot s_D + p, \, j \cdot s_H + q, \, k \cdot s_W + r} \mid i, j, k \in \mathbb{Z}_{\geq 0} \right\}. \tag{19}$$

For each component, compute the norm: $N^{(p,q,r)} = \|\mathbf{X}^{(p,q,r)}\|_p$.

When applying translation $T_{\mathbf{g}}$:

$$\mathcal{P}(T_{\mathbf{g}}\mathbf{X}) = T_{\Delta\hat{k}|T_{\mathbf{g}}\mathbf{X}} \cdot T_{\mathbf{g}} \cdot \mathbf{X}, \tag{20}$$

where $T_{\Delta\hat{k}|T_{\mathbf{g}}\mathbf{X}}$ is the anchoring shift determined by

$$(\hat{p}, \hat{q}, \hat{r}) = \arg\max_{(p,q,r)} N^{(p,q,r)}_{T_{\mathbf{g}}\mathbf{X}}. \tag{21}$$

Due to circular padding, the relative ordering of norms is preserved up to a cyclic permutation under translation. Thus, there exists a translation $\mathbf{g}'$ such that

$$\mathcal{P}(T_{\mathbf{g}}\mathbf{X}) = T_{\mathbf{g}'} \cdot \mathcal{P}(\mathbf{X}). \tag{22}$$

$\square$

**Lemma 2** (Equivariance of Adaptive Phase Selection (APS)). *Let $f_\theta$ be a shift-equivariant feature extractor. Then, the selection probabilities satisfy (Rojas-Gomez et al., 2022).*

$$p_\theta(k = \pi(k) \mid T_{\mathbf{g}}\mathbf{X}) = p_\theta(k = k \mid \mathbf{X}), \tag{23}$$

*Where $\pi$ is the permutation induced by translation.*

*Proof.* Let $\hat{\mathbf{X}} \triangleq T_{\mathbf{g}}\mathbf{X}$. By definition,

$$p_\theta(k = \pi(k) \mid T_{\mathbf{g}}\mathbf{X}) = \frac{\exp\left[f_\theta(\Psi(T_{\mathbf{g}}\mathbf{X})_{\pi(k)})\right]}{\sum_j \exp\left[f_\theta(\Psi(T_{\mathbf{g}}\mathbf{X})_j)\right]}. \tag{24}$$

By shift-equivariance of $f_\theta$ and Lemma 1,

$$f_\theta(\Psi(T_{\mathbf{g}}\mathbf{X})_{\pi(k)}) = f_\theta(\Psi(\mathbf{X})_k). \tag{25}$$

Therefore,

$$p_\theta(k = \pi(k) \mid T_{\mathbf{g}}\mathbf{X}) = p_\theta(k = k \mid \mathbf{X}). \tag{26}$$

$\square$

**Theorem 3** (APT Translation Equivariance). *Let APT denote the Adaptive Phase Selection. Then,*

$$APT(T_{\mathbf{g}}\mathbf{X}) = T_{\mathbf{g}'} APT(\mathbf{X}), \tag{27}$$

*for some translation $\mathbf{g}'$ determined by $\mathbf{g}$ and the anchoring procedure.*

*Proof.* From Lemma 2, the optimal selection for the translated input follows the shift-permutation equivariance:

$$(\hat{p}^*, \hat{q}^*, \hat{r}^*) = \arg\max_{(p,q,r)} p_\theta(k = (p,q,r) \mid T_{\mathbf{g}}\mathbf{X}) = \pi^{-1}(p^*, q^*, r^*), \tag{28}$$

where $(p^*, q^*, r^*)$ is the optimal index for $\mathbf{X}$. Therefore,

$$
\begin{aligned}
\text{APT}(T_{\mathbf{g}}\mathbf{X}) &= \Psi(T_{\mathbf{g}}\mathbf{X})_{(\hat{p}^*, \hat{q}^*, \hat{r}^*)} \\
&= T_{\mathbf{g}'}\Psi(\mathbf{X})_{(p^*, q^*, r^*)} \\
&= T_{\mathbf{g}'}\text{APT}(\mathbf{X}),
\end{aligned}
\tag{29}
$$

where $T_{\mathbf{g}'}$ is the translation corresponding to the permutation $\pi$. $\qquad\square$

### B.3 Rotation Equivariance of APT-ViT module

To enable reliable orientation alignment in 3D space, it is essential that the selected token remains consistent under global rotations. Our APT module achieves this by leveraging spherical steerable convolutions, which guarantee rotation-equivariant feature extraction. The following lemma and theorem formally establish that APT preserves rotation equivariance under $SO(3)$ transformations.

**Lemma 4** (Steerable Convolution Equivariance). *Let $R \in SO(3)$ be a rotation operator. The steerable convolution satisfies (Weiler et al., 2018a).*

$$
\tilde{f}_\theta(Rx) = \sum_{J=0}^{J_{\max}} \sum_{m'=-J}^{J} R_J(\|x\|) \cdot Y_J^{m'}\left(\frac{x}{\|x\|}\right) \cdot \sum_{m=-J}^{J} D_{m',m}^{(J)}(R) w_{J,m},
\tag{30}
$$

*where $R_J(\|x\|)$ is radial function, $Y_J^{m'}(\cdot)$ is spherical harmonics, $D_{m',m}^{(J)}(R)$ is Wigner D-matrix for rotation $R$, $w_{J,m}$ is learnable weights.*

*Proof.* By the transformation property of spherical harmonics,

$$
Y_J^m(R\hat{x}) = \sum_{m'=-J}^{J} D_{m',m}^{(J)}(R) Y_J^{m'}(\hat{x}),
\tag{31}
$$

and the radial function is rotation-invariant: $R_J(\|Rx\|) = R_J(\|x\|)$. $\qquad\square$

**Theorem 5** (APT Rotation Equivariance). *Let $R \in SO(3)$ be a rotation operator. Then,*

$$
APT(R\mathbf{X}) = R\,APT(\mathbf{X}).
\tag{32}
$$

*Proof.* The feature extraction function with steerable convolutions and global pooling,

$$
f_\theta(\Psi(\mathbf{X})_{(p,q,r)}) = \frac{1}{V} \sum_{v \in V} \tilde{f}_\theta(\Psi(\mathbf{X})_{(p,q,r)})[v],
\tag{33}
$$

is rotation-equivariant by Lemma 4. Thus, the selection probabilities satisfy

$$
p_\theta(k = (p, q, r) \mid R\mathbf{X}) = p_\theta(k = (p, q, r) \mid \mathbf{X}).
\tag{34}
$$

Therefore, the same polyphase component is selected, and

$$
\begin{aligned}
\text{APT}(R\mathbf{X}) &= \Psi(R\mathbf{X})_{(p^*, q^*, r^*)} \\
&= R\Psi(\mathbf{X})_{(p^*, q^*, r^*)} \\
&= R\,\text{APT}(\mathbf{X}).
\end{aligned}
\tag{35}
$$

$\qquad\square$

B.4   Rotation Representations in $SO(3)$

In the Hitchhiker's Guide to $SO(3)$ (Geist et al., 2024), a variety of rotation representations are systematically categorized based on their mathematical properties and implications for learning. Inspired by this framework, we explore three representative parameterizations for 3D subtomogram alignment: Euler angles, $\mathbb{R}^6$ with Gram-Schmidt orthonormalization (GSO), and $\mathbb{R}^9$ with singular value decomposition (SVD). While each representation offers distinct trade-offs in terms of continuity, redundancy, and interpretability, we find that Euler angles surprisingly perform best for our alignment task. Detailed results are provided in Appendix H.1, and an overview of their properties is summarized in Table 6.

Table 6: Comparison of selected $SO(3)$ rotation representations used in 3D subtomogram alignment.

| Representation | Notation | Dim | $g(R)$ continuous | Uses Angles | Double Cover |
|---|---|---|---|---|---|
| Euler angles | Euler | 3 | ✗ | ✓ | ✓ |
| $\mathbb{R}^6$ + Gram-Schmidt orthonormalization | $\mathbb{R}^6 + GSO$ | 6 | ✓ | ✗ | ✗ |
| $\mathbb{R}^9$ + Singular Value Decomposition | $\mathbb{R}^9 + SVD$ | 9 | ✓ | ✗ | ✗ |

**Euler Angles.**   One classical method for representing 3D rotations is to use three angular parameters $(\alpha, \beta, \gamma) \in [-\pi, \pi)^3$, commonly known as Euler angles. A rotation matrix can then be constructed by composing a series of axis-specific rotations:

$$R(\alpha, \beta, \gamma) = R_3(\gamma)\, R_2(\beta)\, R_1(\alpha), \tag{36}$$

where $R_i$ applies rotation about the $i$-th axis. Despite its intuitive formulation, this representation introduces issues in practical learning systems. The angle domain exhibits discontinuities due to periodic boundaries, and different sets of angles may correspond to the same rotation. For instance, both $[0, \pi/2, 0]$ and $[-\pi/2, \pi/2, -\pi/2]$ produce identical transformations in $SO(3)$. These ambiguities and the lack of continuity in the mapping from rotation matrices to angles make Euler angles suboptimal for learning tasks, as documented in prior works (Huynh, 2009; Zhou et al., 2019; Brégier, 2021; Pepe et al., 2022).

**$\mathbb{R}^6$ + Gram-Schmidt Orthonormalization (GSO).**   An alternative parameterization leverages two unconstrained 3D vectors $(v_1, v_2) \in \mathbb{R}^{3\times2}$ to implicitly define a rotation. Using the Gram-Schmidt process, these vectors are orthonormalized to recover a valid rotation matrix in $SO(3)$ (Zhou et al., 2019). The procedure involves normalizing $v_1$ to obtain a unit vector $\mathbf{v}_1$, removing its projection from $v_2$ to form $\mathbf{v}_2^\perp$, normalizing that to get $\mathbf{v}_2$, and finally constructing $\mathbf{v}_3$ as the cross product $\mathbf{v}_1 \times \mathbf{v}_2$. The resulting matrix

$$R = [\mathbf{v}_1, \mathbf{v}_2, \mathbf{v}_3] \tag{37}$$

is guaranteed to be orthonormal and lie within $SO(3)$. This formulation is continuous and overcomes angular discontinuities, and generalizes naturally to higher-dimensional rotation groups (Macdonald, 2010).

**$\mathbb{R}^9$ + Singular Value Decomposition (SVD).**   A third formulation represents a rotation via an unconstrained $3 \times 3$ real matrix $M \in \mathbb{R}^{3\times3}$, which is projected onto the space of valid rotations using singular value decomposition (SVD). Decomposing $M$ as $M = U\Sigma V^\top$ with $U, V \in \mathbb{R}^{3\times3}$ orthogonal and $\Sigma$ diagonal, the projection onto $SO(3)$ is defined as

$$f(M) = U \cdot \mathrm{diag}(1, 1, \det(UV^\top)) \cdot V^\top. \tag{38}$$

This ensures that the resulting matrix has unit determinant and is the closest rotation (in Frobenius norm) to $M$ (Levinson et al., 2020). The forward and inverse mappings are well-defined and continuous, making this representation attractive for learning.

## C  CRYOENGINE

### C.1  WORKFLOW

**Structure Modeling.** We curated a library that includes a total of 240 distinct 20S proteasome structures and 212 distinct 30S ribosome structures, capturing a wide range of compositional heterogeneity. All structures are sourced from experimentally validated atomic models archived in the RCSB Protein Data Bank (PDB). Each atomic model is embedded in a cubic grid with a fixed voxel size of 10 Å, the edges of which are extended by a solvent margin that scales with the van der Waals envelopes of its outermost atoms. Electron scattering is approximated by placing a three-dimensional isotropic Gaussian at every atomic center. The kernel width is element-specific, so heavier atoms contribute broader, higher-amplitude densities. Summing these Gaussians yields a continuous Coulombic potential that is subsequently low-pass filtered with a Gaussian whose standard deviation corresponds to half the target resolution (~30 Å), thereby matching the spatial bandwidth of cryo-ET reconstructions. The map is then linearly normalised to unit maximum and voxels below 0.5% of the peak are suppressed, producing a clean, moderate-resolution electron-density volume in MRC format that serves as ground-truth input for all downstream simulation stages.

**Placement Strategy.** Particle centers are generated by Poisson-disk sampling within the interior of a $500 \times 500 \times 200$-voxel simulation box. To guarantee that every $32^3$-voxel subtomogram contains a single macromolecule, we adopt a Poisson-disk sampling strategy in the 3D space. Each accepted center becomes a hard exclusion sphere of radius $R_{ex} = \frac{1}{2}L_{box} + \Delta$, where $L_{box}$=32 and $\Delta$ is a three-voxel safety margin that absorbs later tilt series padding and centring jitter. Subsequent candidates falling inside any existing sphere or crossing the volume boundary are rejected, otherwise they are added to the accepted set and the process continues until the target occupancy is reached. The resulting point pattern is thus collision-free by construction, ensuring that no two particles overlap and that each subtomogram box is entirely filled by a single density. For each accepted center we draw an orientation from a uniform distribution over $SO(3)$ using the Shoemake quaternion algorithm, thereby furnishing dense, unbiased pose coverage for alignment training. The 3D coordinates and quaternions are stored as ground-truth metadata for every instance, yielding a collection of large virtual samples in which each protein appears at a known position and orientation.

**Tilt Series Simulation.** For every virtual samples we simulate the cryo-ET image acquisition process by generating a tilt series of 2D projections from each 3D volume. The 3D density is first re-oriented so that the nominal tilt axis coincides with the detector $y$-axis, after which a sequence of line-integral projections is computed at evenly spaced angles in $2°$ steps. CryoEngine uses a $-60°$ to $+60°$ tilt range by default to reproduce cryo-ET missing wedge, whereas for the pretraining set we used $-90°$ to $+90°$ to decouple noise modeling from anisotropic information loss. At each tilt angle, the volume is rotated by the requested angle via cubic B-spline interpolation, and the rotated density is then integrated along the electron-beam ($z$) direction with an oversampling factor of two to suppress aliasing; the oversampled image is finally decimated to the detector pixel size (10 Å). Each projection is further perturbed by a random sub-pixel in-plane shift to mimic stage drift in practice. CryoEngine supports tilt series noise injection, whereas for the benchmark subtomograms we inject calibrated Gaussian noise later at the subtomogram level to ensure an exact voxel-wise SNR for comparative experiments by preventing any SNR degradation during the reconstruction process. The result of this stage is a set of raw tilt series images for each volume, aligned input for the downstream reconstruction module.

**Alignment and Reconstruction.** The collection of simulated tilt images is then processed to correct the imposed translations. Each projection is Fourier transformed and phase correlated with a reference initialised by the $0°$ view and updated iteratively as the average of the currently aligned images; the correlation peak is fitted quadratically to estimate translation vectors with fractional-pixel precision. These shifts are applied cumulatively so that every projection shares a common coordinate frame, and the resulting parameters $(\Delta x_i, \Delta y_i)$ are stored together with their tilt angles $\theta_i$. A global refinement then determines a single tilt axis orientation and vertical offset by minimising the mean-squared reprojection error, producing a geometrically consistent stack.

Then we employ filtered weighted back-projection for 3-D reconstruction. Each aligned projection is multiplied in Fourier space by a Hann-tapered ramp filter and rescaled by $w(\theta) = |\cos\theta|$ to compensate for the non-uniform angular sampling. The filtered images are then back-projected along

their respective beam directions into a $500 \times 500 \times 200$ voxel volume, with trilinear interpolation used to accumulate contributions on the Cartesian grid.

**Subtomogram Extraction.** From each reconstructed volume, we extract a smaller cubic subtomogram around the location of the macromolecule. We used a box size of $32^3$ voxels for each subtomogram. The center of this crop is based on the known coordinates of the inserted structure. To make the dataset more realistic, we add a minor random offset to the cropping center along each axis. This simulates the practical scenario of particle picking in which the centering of a particle in a subtomogram is not perfect. Before cropping, a coordinate-based quality check confirms that the entire $32^3$ cube stays inside the tomogram and that no other recorded particle centre lies within 17 voxels of the proposed centre, candidates failing either test are skipped. After extraction, each subtomogram is a $32 \times 32 \times 32$ voxel volume that ideally contains the particle roughly centered amidst some surrounding context. We carry forward the ground-truth metadata for each subtomogram, including its class label, the exact orientation that was applied, and the precise position offset within the subtomogram. Additionally, CryoEngine also outputs a binary mask for each subtomogram that labels the voxels belonging to the particle versus background.

**Noise Augmentation.** To emulate the severe photon-limited conditions of cryo-ET, each clean $32^3$ subtomogram is replicated at four calibrated noise levels in addition to a nearly noise-free reference. Let $v_{sig} = \text{Var}(S)$ denote the voxel-wise variance of the signal component of a clean subtomogram. For a desired signal-to-noise ratio $\text{SNR}_{tar}$ (defined as $v_{sig}/\sigma^2$), we draw zero-mean Gaussian noise $N \sim \mathcal{N}(0, \sigma^2)$ with $\sigma^2 = v_{sig}/\text{SNR}_{tar}$ and add it voxel-wise: $V_{noisy} = S + N$. Target values $\text{SNR}_{tar} \in \{100, 0.10, 0.05, 0.03, 0.01\}$ produce five versions of every subtomogram that span the full experimental range from almost deterministic to extremely noise-dominated. Because the same $v_{sig}$ is used to calibrate $\sigma^2$ for each individual volume, the synthetically generated noise preserves relative contrast differences across structures while enforcing the intended global SNR. These augmented versions are used to evaluate and pretrain our models under different noise regimes, ensuring that our learned representations are robust to the severity of cryo-ET noise.

## C.2 Dataset Composition

We simulated over 452 distinct structural classes, with 2,000 subtomograms per class. To illustrate our synthetic dataset, we visualized 84 examples of the 30S ribosome (Fig. 5, Fig. 6, and Fig. 7) and 56 examples of the 20S proteasome (Fig. 8 and Fig. 9). For each selected class, we show three elements: (1) the input atomic model used to initiate simulation; (2) the structure density map after element-specific Gaussian scattering and low-pass filtering; (3) different slices of a synthetic noise-free subtomogram. These visualizations demonstrate the compositional heterogeneity and the high-fidelity resemblance to experimental subtomograms of our synthetic datasets. Additionally, we generated subtomograms at various calibrated noise levels.

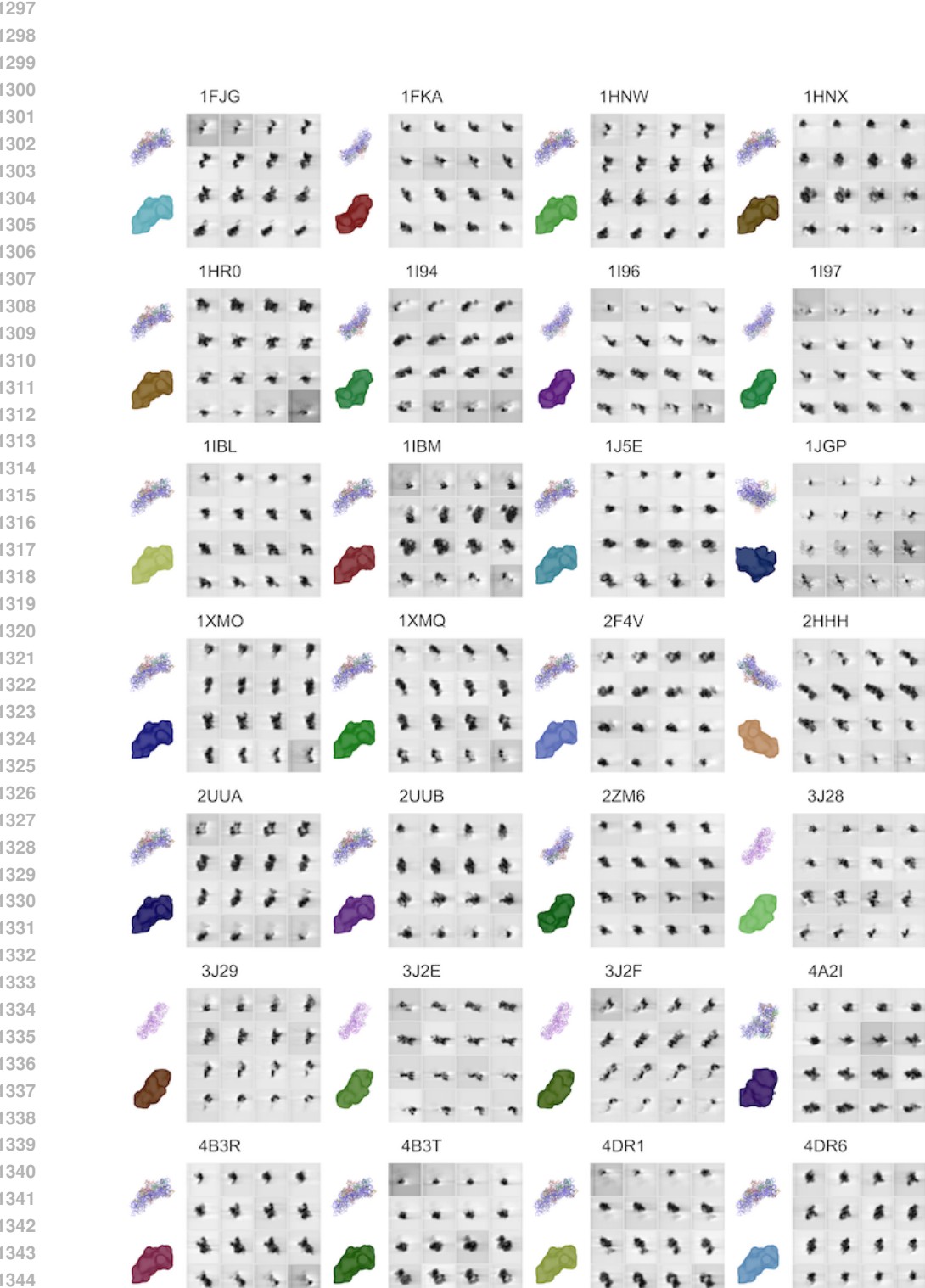

Figure 5: Diverse 30S ribosomes and corresponding noise-free subtomograms.

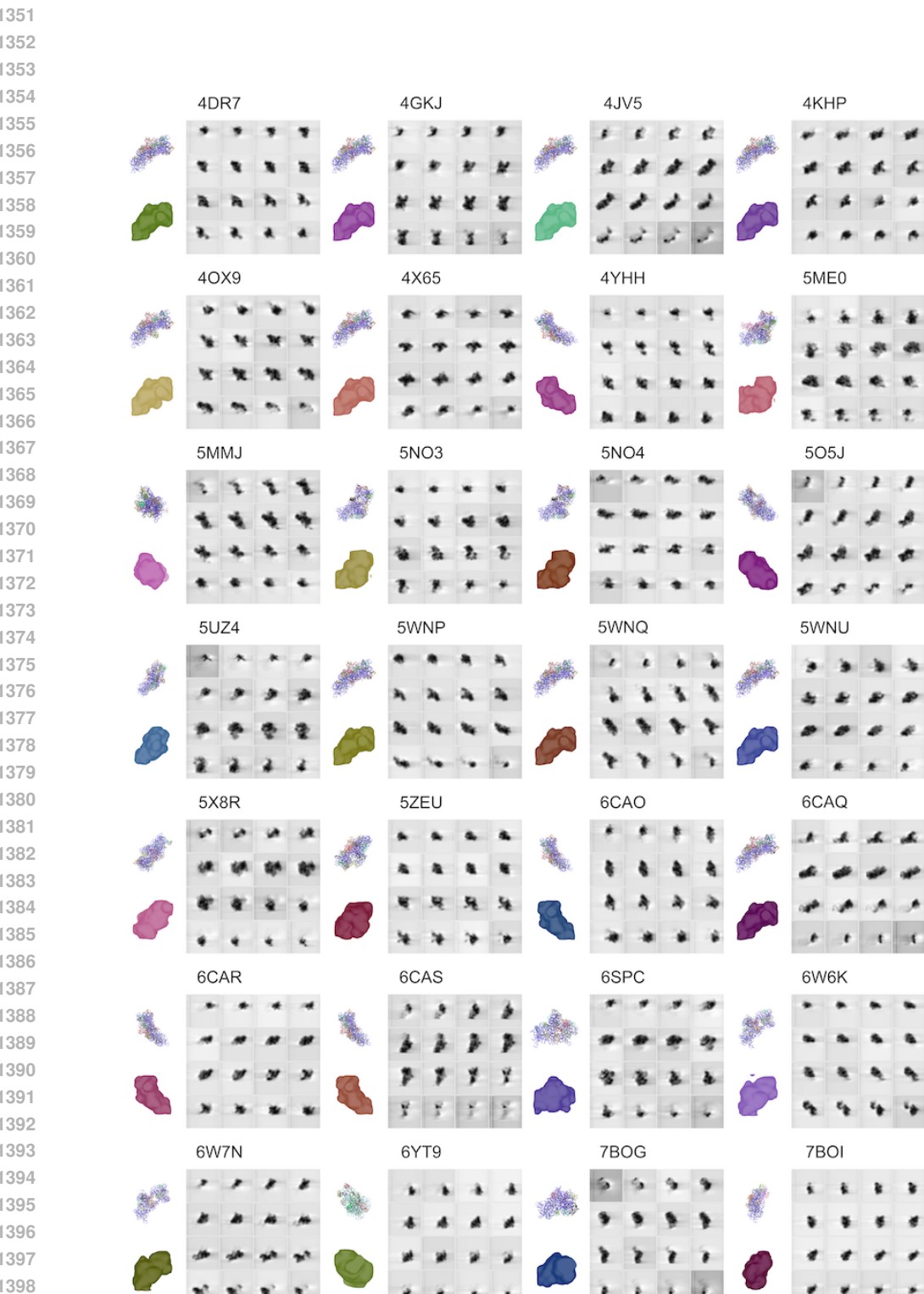

Figure 6: Diverse 30S ribosomes and corresponding noise-free subtomograms.

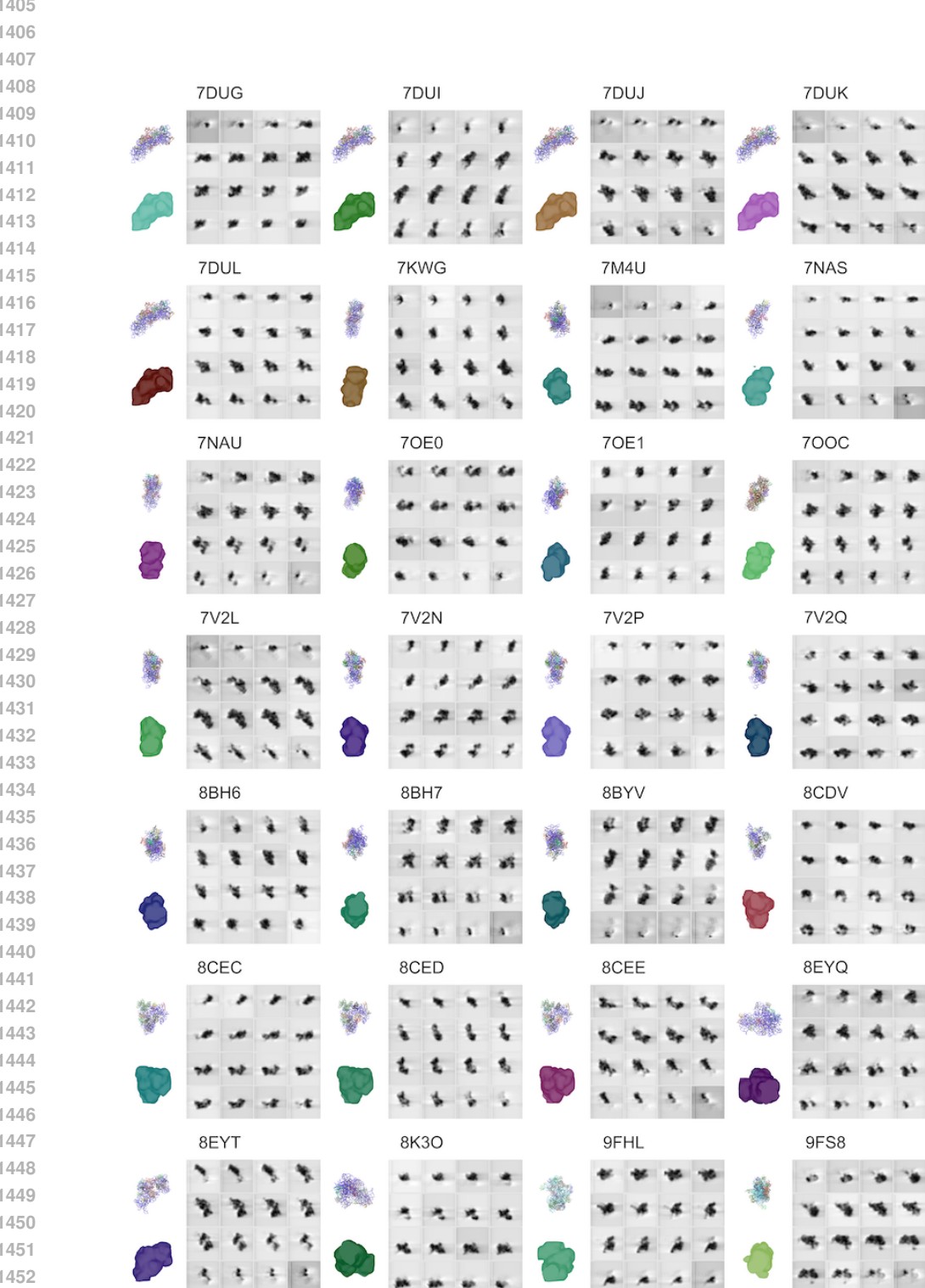

Figure 7: Diverse 30S ribosomes and corresponding noise-free subtomograms.

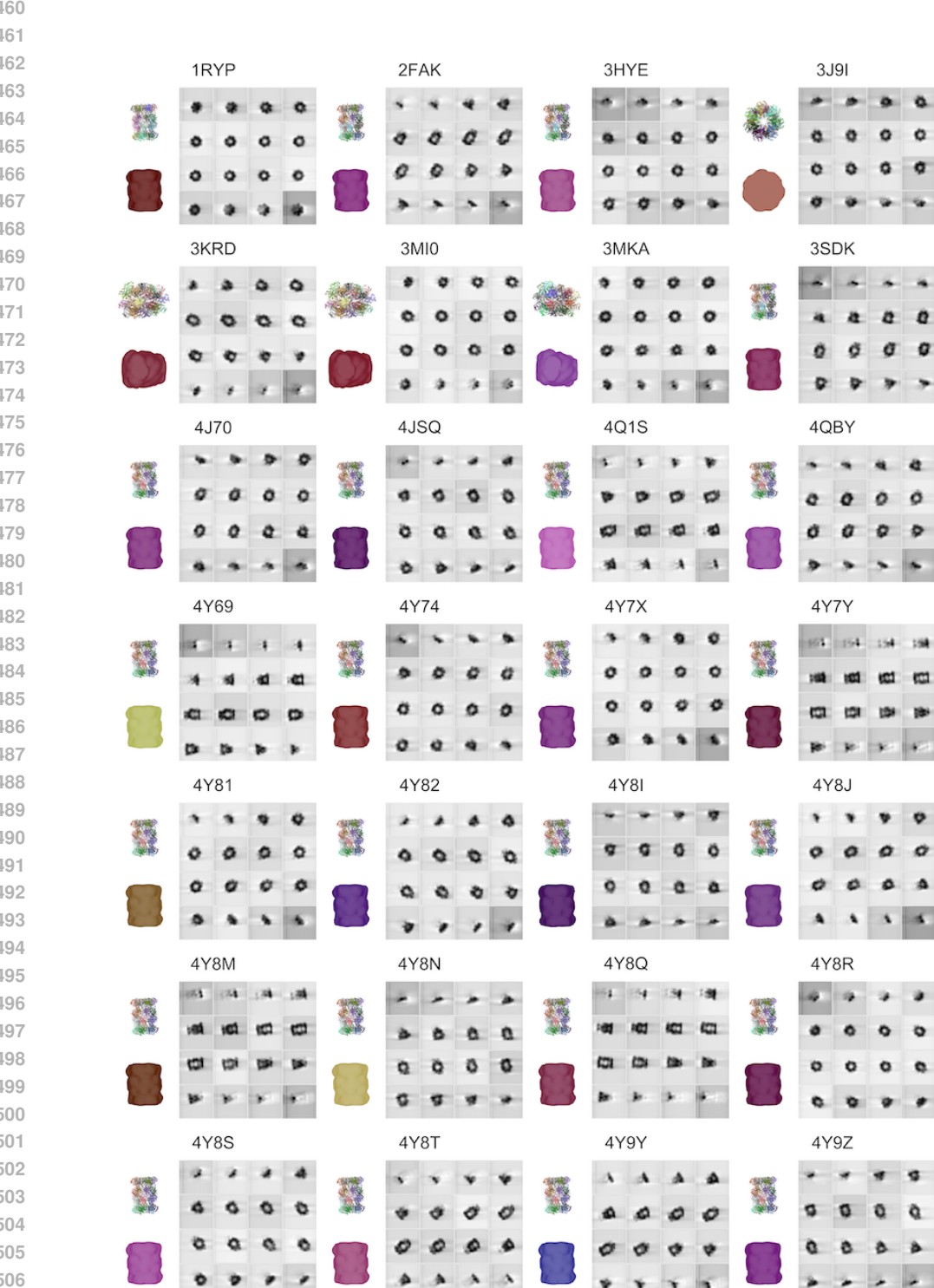

Figure 8: Diverse 20S proteasomes and corresponding noise-free subtomograms.

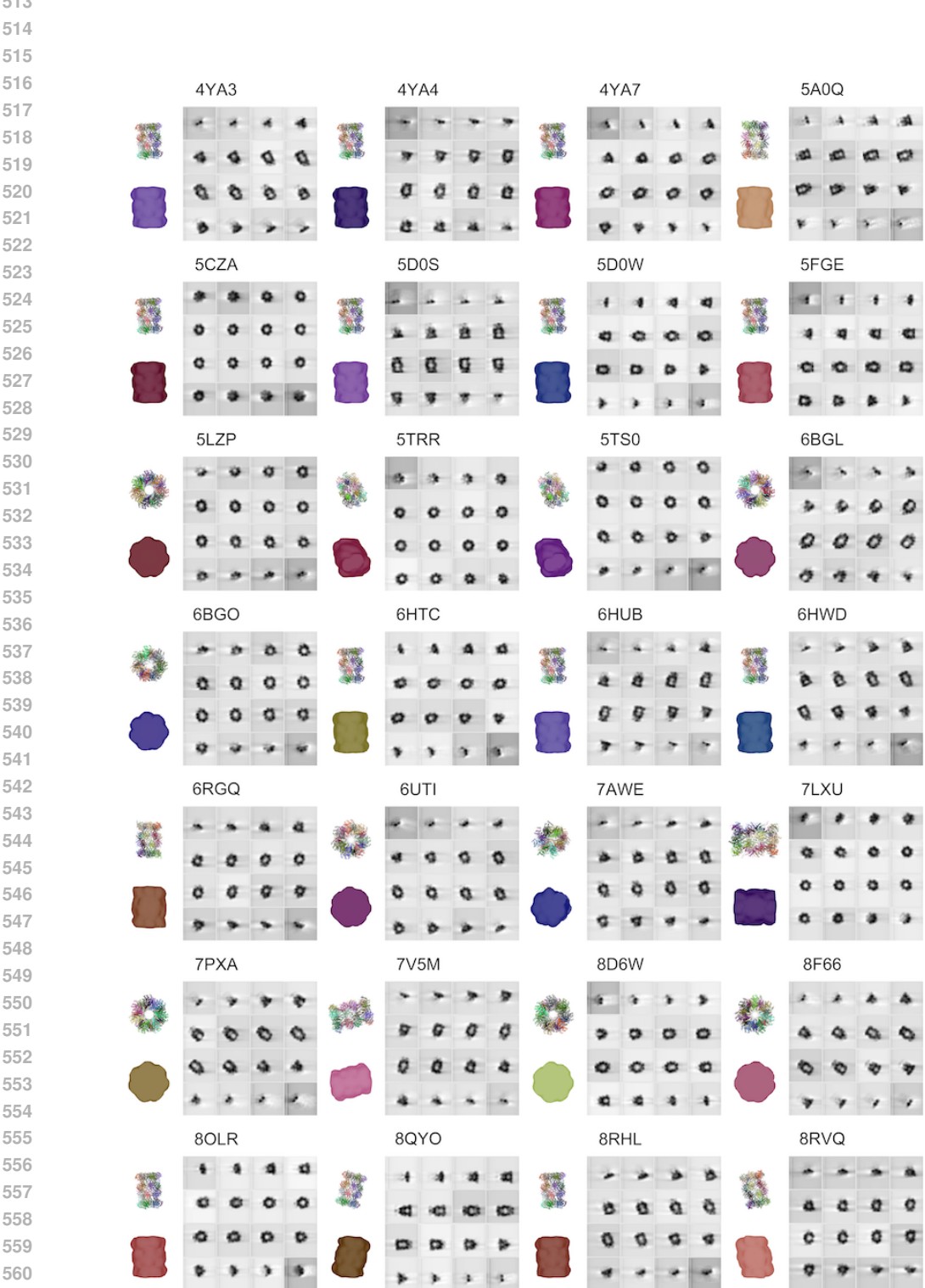

Figure 9: Diverse 20S proteasomes and corresponding noise-free subtomograms.

# D  Noise-resilient Contrastive Learning Strategy

Algorithm 2 summarizes the workflow of noise-resilient contrastive learning (NRCL). The procedure begins by generating two augmented views of each subtomogram, together with corresponding clean and noise-amplified variants. A base encoder and a momentum encoder then extract latent representations from these inputs. The learning objective combines three parts: (1) an instance-level contrastive loss that enforces consistency between different augmented views, (2) an additional term (e.g., Wasserstein-based) that encourages base and momentum representations to remain aligned at the distribution level, and (3) a noise-aware contrast that treats clean versions as positives and noise-amplified versions as negatives. Together, these objectives yield representations that are geometrically consistent while being robust to cryo-ET noise.

---

**Algorithm 2:** Workflow of NRCL

---

**Input:** Subtomogram $X$, clean version $X_{\text{clean}}$, noisy version $X_{\text{noisy}}$, temperature $\tau$, base
      encoder+projector+predictor $f_q$, momentum encoder+projector $f_k$, weight $\lambda_w$

**Output:** Contrastive loss $\mathcal{L}$ for $X, X_{\text{clean}}, X_{\text{noisy}}$

1 **for** *each batch in loader* **do**

2      Randomly sample transformation parameters $T$ and $T'$;

3      $X_1 \leftarrow TX$; $X_{1,\text{noisy}} \leftarrow TX_{\text{noisy}}$; $X_{1,\text{clean}} \leftarrow TX_{\text{clean}}$; $X_2 \leftarrow T'X$;

       // forward through encoders

4      $q_1, q_2 \leftarrow f_q(X_1, X), f_q(X_2, X)$;

5      $k_1, k_2 \leftarrow f_k(X_1, X), f_k(X_2, X)$;

6      $k_{\text{clean}}, k_{\text{noisy}} \leftarrow f_k(X_{1,\text{clean}}, X), f_k(X_{1,\text{noisy}}, X)$;

       // compute instance-level loss

7      $\mathcal{L}_{\text{instance}} \leftarrow L_{\text{sym}}(q_1, k_2, \tau) + \lambda_w L_{\text{wass}}(q_1, k_2) + L_{\text{sym}}(q_2, k_1, \tau) + \lambda_w L_{\text{wass}}(q_2, k_1)$;

       // compute noise-aware loss

8      $\mathcal{L}_{\text{noise}} \leftarrow L_{\text{InfoNCE}}(q_1, k_{\text{clean}}, k_{\text{noisy}}, \tau)$;

       // compute total loss

9      $\mathcal{L} \leftarrow \mathcal{L}_{\text{instance}} + \mathcal{L}_{\text{noise}}$;

---

# E CLASSIFICATION

## E.1 EXPERIMENT SETTINGS

**Datasets.**

- *CryoEngine.* As described in Sec. 2.1, CryoEngine is used to help with representation learning. The composition and statistics is stated in Appendix C.2.

- *Benchmark simulation dataset.* A realistically simulated cryo-ET benchmark dataset at five different SNR levels(100, 0.1, 0.05, 0.03, 0.01)(X & M, 2020). It contains five representative heterogeneous structures(spliceosome(5LQW), RNA polymeraserifampicin complex(1I6V), RNA polymerase II elongation complex(6A5L), ribosome(5T2C), and capped proteasome(5MPA)). Each structure consists of 1000 images at each SNR level, with subtomograms having a size of $32^3$.

**Training Implementation.**   We employ NRCL(see Sec. 2.3) to encourage the model to learn robust representations. For each sample, the noisy view is generated through a combination of 3D spatial augmentations (random rotations and translations), additive Gaussian noise, random solarization (with probability 0.2), and brightness adjustment (within $\pm 20\%$ of the mean intensity). These perturbations are designed to simulate noise characteristics commonly observed in real tomographic tilt series acquisition. Training is conducted for 200 epochs using a batch size of 2048. The learning rate follows the square-root scaling rule which is learning rate $= \sqrt{\text{batch size}/512} \times 10^{-5}$ . The Weight decay is fixed at $1 \times 10^{-4}$. For the contrastive learning framework, we adopt a temperature parameter of 0.1 and a momentum coefficient of 0.99 for the momentum encoder.

In the finetune phase, for each benchmark test dataset corresponding to a specific SNR level, we fine-tune the classification head for 40 epochs using a combination of all SNR-100 samples and 10% of the samples from the target low-SNR level. The target low data is split into training, validation, and test sets using a 1:1:8 ratio. The finetuning uses a learning rate of $1 \times 10^{-6}$ with batch size 16.

## E.2 INTRODUCTION OF BASELINES

Here, we provide an introduction to the baseline cryo-ET classification methods. We include both supervised methods and unsupervised methods using different learning strategies. For all the baseline models trained with a 2D setting, each subtomogram is treated as a stack of 2D slices, the 2D backbone acts as a per-slice feature extractor, and slice features are aggregated along the depth dimension(pooling + MLP) into a volume-level representation. They are state-of-the-art methods including:

- **ConvNeXt v1** (Liu et al., 2022b): A modified CNN-based model that integrates elements inspired by transformers. While preserving the efficiency and locality of traditional CNNs, it enhances the capacity for global feature modeling and improves training scalability.

- **ConvNeXt v2** (Woo et al., 2023): A variant of ConvNeXt v1, which introduces global response normalization (GRN), improved depthwise convolution scaling, and an advanced training recipe. These additions further boost the model's representational power and convergence speed

- **ViT** (Dosovitskiy et al., 2021): A basic transformer-based model for vision tasks.

- **PVT** (Wang et al., 2022): A hierarchical transformer with a pyramidal structure, improving its ability to model features at multiple scales.

- **SwinViT** (Liu et al., 2021): A variant of ViT which incorporates a shifted window mechanism to perform efficient self-attention within local regions, thereby achieving both high performance and computational efficiency.

- **MoCo v3**(Pham et al., 2023) A contrastive learning framework that leverages a dynamic dictionary and momentum encoder to learn useful representations without labels.

- **MAE**(He et al., 2022) A self-supervised learning paradigm which learns by reconstructing missing image patches. It promotes semantic understanding from partial inputs.

- **DINO v2**(Oquab et al., 2024) A powerful self-supervised vision model that leverages large-scale training of vision transformers to learn high-quality, general-purpose image representations.

### E.3  CLASSIFICATION RECALL ACROSS DIVERSE MACROMOLECULAR STRUCTURES

To provide a more detailed evaluation of model performance, we report the per-class recall for all five categories. As shown in Tables 7-11, due to the poor performance of some baseline models, the classifier completely failed to identify any correct instances in some categories. These cases are marked with a hyphen (-) in the table for clarity. Moreover, certain non-zero recall values may appear deceptively high due to the model has completely overfitted to specific classes, rather than reflecting meaningful discriminative ability.

Table 7: Classification recall (%) for PDB ID 5LQW across different SNR levels.

| Method | SNR 100 | SNR 0.1 | SNR 0.05 | SNR 0.03 | SNR 0.01 |
|---|---|---|---|---|---|
| ConvNeXt v1 | 100.00 | 69.08 | 53.96 | 50.23 | 41.15 |
| ConvNeXt v2 | 99.90 | 2.34 | - | - | 1.06 |
| ViT | 99.70 | 1.10 | 0.40 | 0.20 | 0.60 |
| PVT v2 | 99.60 | 23.26 | 7.96 | 7.05 | 5.13 |
| SwinViT-B | 99.40 | 60.40 | 50.50 | 40.40 | 28.00 |
| SwinViT-S | 99.30 | 71.10 | 39.40 | 42.40 | 14.90 |
| Moco v3 | 99.60 | 38.50 | 32.70 | 24.80 | 12.50 |
| MAE | 98.40 | 47.50 | 35.00 | 29.00 | 37.00 |
| DINO v2 | 99.20 | 5.90 | 2.90 | 1.80 | 2.20 |
| **Ours** | 97.90 | 53.25 | 42.25 | 63.38 | 39.13 |

Table 8: Classification recall (%) for PDB ID 1I6V across different SNR levels.

| Method | SNR 100 | SNR 0.1 | SNR 0.05 | SNR 0.03 | SNR 0.01 |
|---|---|---|---|---|---|
| ConvNeXt v1 | 99.70 | - | - | - | - |
| ConvNeXt v2 | 100.00 | - | - | - | - |
| ViT | 99.40 | 35.40 | 21.90 | 17.40 | 13.10 |
| PVT v2 | 98.40 | - | - | - | - |
| SwinViT-B | 99.10 | 41.90 | 21.50 | 18.10 | 7.10 |
| SwinViT-S | 99.30 | 43.50 | 22.80 | 16.00 | 25.10 |
| Moco v3 | 99.60 | 53.70 | 21.60 | 8.20 | 0.10 |
| MAE | 98.90 | 49.10 | 29.80 | 17.50 | 35.50 |
| DINO v2 | 34.37 | 52.75 | 27.57 | 21.18 | 15.28 |
| **Ours** | 99.50 | 69.50 | 52.50 | 33.63 | - |

Table 9: Classification recall (%) for PDB ID 5MPA across different SNR levels.

| Method | SNR 100 | SNR 0.1 | SNR 0.05 | SNR 0.03 | SNR 0.01 |
|---|---|---|---|---|---|
| ConvNeXt v1 | 99.70 | - | - | - | - |
| ConvNeXt v2 | 99.30 | - | - | - | - |
| ViT | 94.10 | - | - | - | - |
| PVT v2 | 98.00 | - | - | - | - |
| SwinViT-B | 98.40 | 42.40 | 14.00 | 4.10 | 8.30 |
| SwinViT-S | 98.50 | 46.90 | 20.80 | 8.10 | 17.60 |
| Moco v3 | 98.40 | 11.00 | 0.50 | 0.20 | - |
| MAE | 96.00 | 25.20 | 19.10 | 19.10 | 7.60 |
| DINO v2 | 91.00 | 1.10 | 0.3 | - | - |
| **Ours** | 95.20 | 64.38 | 40.13 | 31.13 | 29.75 |

Table 10: Classification recall (%) for PDB ID 5T2C across different SNR levels.

| Method | SNR 100 | SNR 0.1 | SNR 0.05 | SNR 0.03 | SNR 0.01 |
|---|---|---|---|---|---|
| ConvNeXt v1 | 100.00 | 93.92 | 85.04 | 79.77 | 68.85 |
| ConvNeXt v2 | 100.00 | 99.54 | 100.00 | 100.00 | 99.97 |
| ViT | 100.00 | 96.70 | 93.40 | 91.90 | 88.90 |
| PVT v2 | 99.90 | 98.74 | 97.03 | 97.90 | 95.87 |
| SwinViT-B | 99.50 | 71.60 | 62.40 | 39.30 | 15.10 |
| SwinViT-S | 100.00 | 70.40 | 55.90 | 44.60 | 33.50 |
| Moco v3 | 99.90 | 94.90 | 91.70 | 90.20 | 87.70 |
| MAE | 99.80 | 62.90 | 39.30 | 50.70 | 20.30 |
| DINO v2 | 46.70 | 97.00 | 95.20 | 93.70 | 88.60 |
| **Ours** | 100.00 | 90.00 | 73.38 | 69.25 | 42.00 |

Table 11: Classification recall (%) for PDB ID 6A5L across different SNR levels.

| Method | SNR 100 | SNR 0.1 | SNR 0.05 | SNR 0.03 | SNR 0.01 |
|---|---|---|---|---|---|
| ConvNeXt v1 | 99.50 | - | - | - | - |
| ConvNeXt v2 | 100.00 | - | - | - | - |
| ViT | 96.70 | - | - | - | - |
| PVT v2 | 99.800 | - | - | - | - |
| SwinViT-B | 97.60 | 39.80 | 27.50 | 38.00 | 46.10 |
| SwinViT-S | 97.60 | 26.90 | 17.70 | 25.00 | 18.90 |
| Moco v3 | 98.90 | 6.90 | 2.70 | 3.00 | 5.90 |
| MAE | 95.80 | 38.90 | 22.80 | 32.50 | 18.40 |
| DINO v2 | 80.90 | 5.30 | 0.4 | 0.3 | - |
| **Ours** | 91.60 | 53.50 | 45.75 | 24.25 | 26.63 |

### E.4 Impact of Boxsize and Resolution in subtomogram classification

To assess robustness to changes in voxel resolution and subtomogram size, we additionally evaluated our method (CryoEngine+NRCL+APT-ViT) with a larger field of view ($64^3$ at 10Å) and a blurrier regime ($32^3$ at 20Å). We used the same pretrained weights and followed the same finetune implementation as in Sec 3 by keeping these weights frozen. The results in Table 12 show that our pretrained model still has a clear advantage over the best baseline (CryoEngine+MAE+ViT-B) in Sec 3.2 and thus demonstrates stronger generalization under different box-size and resolution settings.

Table 12: Classification recall (%) across different SNR levels with different boxsize and resolution. $N^3 @ s$ Å indicates an $N \times N \times N$ subtomogram resampled to $s$ Å/voxel.

| Input | Method | SNR 0.1 | SNR 0.05 | SNR 0.03 | SNR 0.01 | Average |
|---|---|---|---|---|---|---|
| $64^3$ @ 10Å | CryoEngine + MAE + ViT-B | 28.88 | 21.95 | 21.37 | 19.47 | 22.92 |
| $64^3$ @ 10Å | CryoEngine + NRCL + APT-ViT | 56.47 | 43.92 | 34.00 | 22.62 | 39.25 |
| $32^3$ @ 20Å | CryoEngine + MAE + ViT-B | 38.92 | 26.93 | 23.38 | 19.82 | 27.26 |
| $32^3$ @ 20Å | CryoEngine + NRCL + APT-ViT | 39.75 | 31.60 | 27.15 | 22.43 | 30.23 |
| $32^3$ @ 10Å | CryoEngine + MAE + ViT-B | 56.93 | 41.14 | 30.65 | 22.40 | 37.78 |
| $32^3$ @ 10Å | CryoEngine + NRCL + APT-ViT | 67.42 | 53.13 | 40.10 | 27.50 | 47.04 |

## F    ALIGNMENT

### F.1    EXPERIMENT SETTINGS

**Datasets.**    A realistically simulated cryo-ET benchmark dataset (X & M, 2020) at four different SNR levels(0.1, 0.05, 0.03, 0.01). It contains five representative heterogeneous structures(spliceosome(PDB ID: 5LQW), RNA polymeraserifampicin complex(PDB ID: 1I6V), RNA polymerase II elongation complex(PDB ID: 6A5L), ribosome(PDB ID: 5T2C), and capped proteasome(PDB ID: 5MPA)). Each structure consists of 1000 images at each SNR level, with subtomograms having a size of $32^3$. Fig. 10 presents subtomograms of five complexes under different SNR levels.

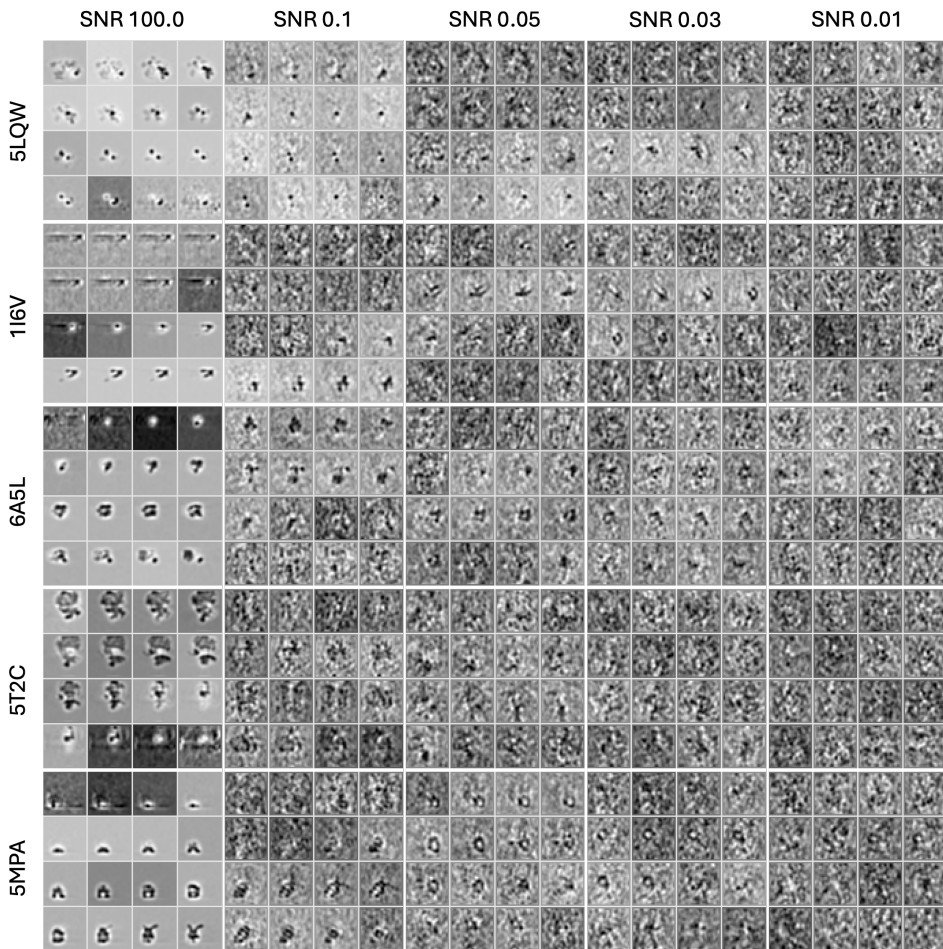

Figure 10: Subtomograms under different SNR levels (100.0, 0.1, 0.05, 0.03, 0.01).

**Training startegy.**    To enable the learning of effective features for alignment tasks, the model is trained on a broad distribution of alignment patterns. For each subtomogram, we generate 10 distinct alignment patterns, simulating possible spatial transformations and matching scenarios. These pairwise patterns serve as supervisory signals to guide the model in learning transformation-related representations.

During the finetuning phase, we adopt a more conservative set of alignment patterns to improve the model's sensitivity to small transformations. Specifically, we apply only rotations within $\pm 15°$ and translations of up to 20% of the subtomogram size. The finetuning consists of 10 training epochs.

**Alignment Error Metrics**    To quantitatively assess subtomogram alignment performance, we report both rotational and translational deviations between predicted and ground-truth transformations.

Given the estimated and ground-truth rotation matrices $\mathbf{R}_{\text{est}}$ and $\mathbf{R}_{\text{gt}}$, the rotation error $e_{\text{rot}}$ (in degrees) is computed as:

$$e_{\text{rot}} = \arccos\left(\frac{\text{tr}(\mathbf{R}_{\text{est}}^{\top}\mathbf{R}_{\text{gt}}) - 1}{2}\right) \cdot \frac{180}{\pi} \tag{39}$$

where $\mathbf{R}_{\text{est}}, \mathbf{R}_{\text{gt}} \in SO(3)$ are elements of the special orthogonal group, and $\text{tr}(\cdot)$ denotes the matrix trace. This expression yields rotational errors bounded within $[0°, 180°]$. The translation error $e_{\text{trans}}$ (in voxels) is defined as the Euclidean distance between the estimated and ground-truth translation vectors:

$$e_{\text{trans}} = \left\|\mathbf{t}_{\text{est}} - \mathbf{t}_{\text{gt}}\right\|_2 \tag{40}$$

### F.2 Introduction of Baselines

Here, we provide an introduction to the baseline cryo-ET alignment methods. For clarity, we categorize them into three groups: traditional algorithms, CNN-based deep learning models, and equivariant Transformer architectures.

**Traditional subtomogram alignment methods.**

- **H-T align** (Xu et al., 2012): A Fourier-based rotational alignment method specially designed for low SNR and high tilt angle conditions.
- **F&A align** (Chen et al., 2013): An efficient reference-free subtomogram alignment algorithm employing spherical harmonics and Wiener-filtered corrections.

**CNN-based deep learning methods for subtomogram alignment.**

- **Gum-Net** (X & M, 2020): An unsupervised CNN-based model designed for 3D geometric correspondences and specifically optimized to handle the noise inherent in cryo-ET data. It achieves significant improvements in both accuracy and computational efficiency. The Gum-Net framework includes three architectural variants: Gum-Net MP utilizes max pooling for robust feature extraction; Gum-Net AP applies average pooling to enhance feature aggregation; and Gum-Net SC streamlines the matching process by generating a single correlation map.
- **Jim-Net** (Zeng et al., 2021a): A multi-task unsupervised CNN-based model that simultaneously clusters and aligns subtomograms.

**Equivariant Transformer approaches.**

- **SE(3)-Transformer** (Fuchs et al., 2020a): A family of 3D roto-translation equivariant attention networks that leverage group theory to achieve strict SE(3) equivariance.
- **ConDor** (Sajnani et al., 2022): A self-supervised approach for canonicalizing the 3D pose of both full and partial shapes.
- **Equi-Pose** (Li et al., 2021): A self-supervised learning framework for estimating category-level 6D object poses directly from single 3D point clouds.
- **BOE-ViT** (Jiang et al., 2025): A vision transformer-based framework for 3D subtomogram alignment that incorporates equivariant design to boost orientation estimation. Unlike three point cloud-based methods above, BOE-ViT is specifically optimized for cryo-ET subtomograms, leveraging self-supervised learning and equivariance-aware attention mechanisms to enhance robustness.

## F.3 Performance Across Diverse Macromolecular Structures

We evaluated the alignment performance on five representative macromolecular complexes under varying SNR conditions. As shown in Tables 13-17, our model consistently outperforms existing methods in both accuracy and robustness, which includes H-T align, F&A align, the four variants of Gum-Net (Gum-Net MP, Gum-Net AP, and Gum-Net SC, Gum-Net) and Jimnet.

Our APT-ViT also demonstrates superior performance compared to equivariant neural networks including SE(3)-Transformer, ConDor, and Equi-Pose, and notably outperforms the current equivariant ViT like BOE-ViT across all SNR levels. This validates the effectiveness of our equivariant architectural improvements for subtomogram alignment tasks.

Table 13: Subtomogram alignment accuracy at different SNR levels. Each table entry shows the mean and standard deviation of both rotation and translation errors. Results are for PDB ID: 1I6V.

| Method | SNR 0.1 | | SNR 0.05 | | SNR 0.03 | | SNR 0.01 | |
| --- | --- | --- | --- | --- | --- | --- | --- | --- |
| | Rotation | Translation | Rotation | Translation | Rotation | Translation | Rotation | Translation |
| H-T align | 1.67±1.06 | 6.31±5.01 | 2.09±0.87 | 7.65±4.56 | 2.22±0.74 | 8.10±4.43 | 2.40±0.57 | 10.93±4.97 |
| F-A align | 1.71±1.08 | 6.63±4.96 | 2.06±0.90 | 7.76±4.67 | 2.23±0.74 | 8.48±4.62 | 2.37±0.56 | 10.94±4.98 |
| Gum-Net MP | 1.38±0.75 | 5.25±3.53 | 1.50±0.76 | 5.70±3.65 | 1.59±0.76 | 6.08±3.54 | 1.66±0.77 | 7.06±3.39 |
| Gum-Net AP | 1.25±0.76 | 4.75±3.37 | 1.39±0.76 | 5.35±3.49 | 1.53±0.75 | 5.81±3.46 | 1.65±0.77 | 7.02±3.35 |
| Gum-Net SC | 1.26±0.77 | 4.83±3.58 | 1.42±0.77 | 5.43±3.62 | 1.53±0.76 | 5.73±3.47 | 1.68±0.76 | 6.96±3.52 |
| Gum-Net | 0.75±0.77 | 2.99±3.17 | 0.87±0.76 | 3.49±3.31 | 1.05±0.71 | 3.96±2.77 | 1.42±0.78 | 5.66±3.53 |
| Jim-Net | 0.78±0.71 | 3.15±3.13 | 1.03±0.74 | 4.14±3.58 | 1.18±0.73 | 4.68±3.34 | 1.60±0.75 | 6.55±3.43 |
| SE(3)-Transformer | 1.74±0.11 | 4.17±0.63 | 1.45±0.33 | 5.81±0.88 | 1.16±0.14 | 3.43±0.64 | 1.64±0.59 | 4.92±0.34 |
| ConDor | 6.74±1.82 | 6.59±1.61 | 6.26±1.85 | 6.10±1.89 | 5.86±1.40 | 5.81±1.14 | 5.57±1.85 | 5.50±1.88 |
| Equi-Pose | 4.00±1.51 | 2.08±1.74 | 6.16±1.81 | 3.16±1.92 | 7.58±2.15 | 5.14±2.63 | 9.59±3.05 | 7.24±3.00 |
| BOE-ViT | 0.33±0.16 | 2.41±0.84 | 0.34±0.15 | 2.31±0.81 | 0.34±0.16 | 2.25±0.80 | 0.33±0.15 | 2.26±0.78 |
| **Ours** | **0.24±0.08** | **1.99±0.83** | **0.25±0.08** | **1.89±0.83** | **0.25±0.08** | **1.96±0.85** | **0.25±0.09** | **2.04±0.83** |

Table 14: Subtomogram alignment accuracy at different SNR levels. Each table entry shows the mean and standard deviation of both rotation and translation errors. Results are for PDB ID: 6A5L.

| Method | SNR 0.1 | | SNR 0.05 | | SNR 0.03 | | SNR 0.01 | |
| --- | --- | --- | --- | --- | --- | --- | --- | --- |
| | Rotation | Translation | Rotation | Translation | Rotation | Translation | Rotation | Translation |
| H-T align | 0.94±0.95 | 3.75±4.03 | 1.74±1.02 | 6.31±4.60 | 2.21±0.75 | 8.69±4.56 | 2.37±0.55 | 11.58±5.02 |
| F-A align | 1.06±1.06 | 4.31±4.41 | 1.85±0.99 | 6.99±4.85 | 2.18±0.79 | 8.69±4.55 | 2.39±0.58 | 11.31±4.83 |
| Gum-Net MP | 1.13±0.74 | 4.27±3.09 | 1.30±0.75 | 4.80±3.11 | 1.45±0.76 | 5.45±3.09 | 1.66±0.77 | 6.99±3.28 |
| Gum-Net AP | 0.98±0.67 | 3.72±2.74 | 1.20±0.72 | 4.45±2.85 | 1.40±0.74 | 5.29±3.02 | 1.64±0.77 | 6.97±3.33 |
| Gum-Net SC | 1.07±0.73 | 4.02±3.03 | 1.26±0.76 | 4.56±3.07 | 1.47±0.77 | 5.48±3.14 | 1.65±0.76 | 6.89±3.33 |
| Gum-Net | 0.46±0.54 | 1.80±1.90 | 0.71±0.63 | 2.55±2.12 | 1.12±0.73 | 3.93±2.45 | 1.45±0.76 | 5.94±3.32 |
| Jim-Net | 0.39±0.52 | **1.67±2.01** | 0.64±0.60 | 2.42±2.33 | 0.99±0.72 | 3.71±2.89 | 1.58±0.76 | 6.69±3.38 |
| SE(3)-Transformer | 1.49±0.57 | 3.87±0.85 | 1.21±0.27 | 3.42±0.93 | 1.20±0.32 | 3.44±0.45 | 1.44±0.21 | 3.93±0.86 |
| ConDor | 8.04±1.42 | 7.86±1.32 | 7.11±1.45 | 7.07±1.08 | 7.12±1.09 | 6.97±1.94 | 5.42±1.56 | 5.31±1.17 |
| Equi-Pose | 3.91±1.92 | 2.78±1.17 | 5.76±2.31 | 3.51±1.88 | 7.31±2.19 | 5.20±2.10 | 9.83±3.07 | 7.12±3.14 |
| BOE-ViT | 0.33±0.15 | 2.30±0.80 | 0.34±0.16 | 2.27±0.81 | 0.35±0.15 | 2.27±0.75 | 0.34±0.15 | 2.24±0.78 |
| **Ours** | **0.24±0.08** | 1.95±0.83 | **0.25±0.08** | **1.97±0.82** | **0.25±0.08** | **1.95±0.79** | **0.25±0.09** | **2.00±0.83** |

Table 15: Subtomogram alignment accuracy at different SNR levels. Each table entry shows the mean and standard deviation of both rotation and translation errors. Results are for PDB ID: 5LQW.

| Method | SNR 0.1 | | SNR 0.05 | | SNR 0.03 | | SNR 0.01 | |
|---|---|---|---|---|---|---|---|---|
| | Rotation | Translation | Rotation | Translation | Rotation | Translation | Rotation | Translation |
| H-T align | 0.61±0.87 | 2.64±3.55 | 1.62±1.14 | 6.08±4.92 | 2.15±0.88 | 8.49±4.72 | 2.38±0.56 | 11.36±5.13 |
| F-A align | 0.64±0.97 | 2.96±3.99 | 1.68±1.16 | 6.32±4.91 | 2.12±0.89 | 8.39±4.79 | 2.35±0.59 | 11.20±5.00 |
| Gum-Net MP | 1.02±0.70 | 4.07±3.16 | 1.25±0.78 | 4.89±3.30 | 1.38±0.75 | 5.41±3.31 | 1.65±0.78 | 6.79±3.08 |
| Gum-Net AP | 0.87±0.65 | 3.56±2.78 | 1.12±0.74 | 4.45±3.00 | 1.29±0.74 | 5.07±3.09 | 1.60±0.81 | 6.69±3.11 |
| Gum-Net SC | 0.96±0.71 | 3.83±3.13 | 1.22±0.79 | 4.76±3.28 | 1.38±0.76 | 5.28±3.33 | 1.65±0.78 | 6.82±3.20 |
| Gum-Net | 0.47±0.57 | 1.94±2.26 | 0.68±0.64 | 2.61±2.25 | 0.93±0.68 | 3.62±2.32 | 1.38±0.78 | 5.65±3.31 |
| Jim-Net | 0.30±0.47 | **1.42±2.01** | 0.51±0.58 | 2.30±2.36 | 0.74±0.62 | 3.13±2.63 | 1.50±0.76 | 6.30±3.13 |
| SE(3)-Transformer | 1.16±0.66 | 4.04±0.32 | 1.62±0.34 | 4.69±0.40 | 1.83±0.51 | 5.18±0.64 | 1.78±0.66 | 5.19±0.79 |
| ConDor | 7.21±1.44 | 7.07±1.07 | 6.95±1.90 | 6.93±1.31 | 6.49±1.57 | 6.30±1.08 | 6.25±1.41 | 6.16±1.87 |
| Equi-Pose | 4.34±1.78 | 2.40±1.28 | 5.88±1.96 | 3.52±2.07 | 7.53±2.37 | 4.73±2.59 | 10.25±3.03 | 7.07±2.83 |
| BOE-ViT | 0.33±0.15 | 2.30±0.83 | 0.34±0.16 | 2.27±0.79 | 0.34±0.15 | 2.24±0.77 | 0.34±0.16 | 2.21±0.77 |
| **Ours** | **0.24±0.08** | 1.96±0.80 | **0.25±0.08** | 1.98±0.82 | **0.25±0.08** | 1.98±0.84 | **0.25±0.08** | 2.05±0.80 |

Table 16: Subtomogram alignment accuracy at different SNR levels. Each table entry shows the mean and standard deviation of both rotation and translation errors. Results are for PDB ID: 5T2C.

| Method | SNR 0.1 | | SNR 0.05 | | SNR 0.03 | | SNR 0.01 | |
|---|---|---|---|---|---|---|---|---|
| | Rotation | Translation | Rotation | Translation | Rotation | Translation | Rotation | Translation |
| H-T align | 1.16±1.04 | 4.43±4.21 | 2.13±0.84 | 8.79±4.77 | 2.34±0.61 | 10.59±4.98 | 2.36±0.59 | 11.56±4.91 |
| F-A align | 1.54±1.12 | 6.39±5.19 | 2.17±0.80 | 9.39±5.09 | 2.35±0.58 | 10.81±4.93 | 2.40±0.55 | 11.81±4.89 |
| Gum-Net MP | 1.58±0.83 | 5.51±3.07 | 1.71±0.80 | 6.28±3.16 | 1.70±0.80 | 6.72±3.13 | 1.70±0.78 | 8.27±3.58 |
| Gum-Net AP | 1.30±0.79 | 4.71±2.76 | 1.58±0.80 | 5.94±3.05 | 1.63±0.81 | 6.70±3.20 | 1.68±0.78 | 8.14±3.51 |
| Gum-Net SC | 1.41±0.79 | 4.90±2.94 | 1.63±0.79 | 5.98±3.11 | 1.66±0.80 | 6.54±3.15 | 1.71±0.77 | 8.35±3.64 |
| Gum-Net | 0.73±0.81 | 2.70±2.87 | 1.19±0.84 | 4.23±3.01 | 1.43±0.79 | 5.67±2.96 | 1.76±0.75 | 10.46±5.10 |
| Jim-Net | 0.49±0.70 | 1.99±2.43 | 1.09±0.86 | 4.14±3.30 | 1.33±0.83 | 5.19±3.28 | 1.65±0.78 | 7.60±3.62 |
| SE(3)-Transformer | 2.26±0.70 | 4.31±0.64 | 1.69±0.01 | 5.53±0.45 | 1.97±0.78 | 5.30±0.47 | 1.91±0.08 | 6.35±0.43 |
| ConDor | 7.49±1.51 | 7.46±1.46 | 7.09±1.56 | 6.97±1.86 | 6.83±1.24 | 6.66±1.40 | 6.09±1.17 | 6.05±1.77 |
| Equi-Pose | 3.76±1.19 | 2.28±1.87 | 6.04±1.89 | 3.31±2.16 | 7.24±2.64 | 4.87±2.75 | 9.81±2.92 | 7.18±2.68 |
| BOE-ViT | 0.34±0.15 | 2.28±0.81 | 0.34±0.16 | 2.24±0.77 | 0.34±0.15 | 2.27±0.80 | 0.34±0.15 | 2.30±0.79 |
| **Ours** | **0.24±0.08** | **1.94±0.81** | **0.25±0.08** | **1.96±0.83** | **0.24±0.08** | **1.96±0.81** | **0.25±0.08** | **2.01±0.81** |

Table 17: Subtomogram alignment accuracy at different SNR levels. Each table entry shows the mean and standard deviation of both rotation and translation errors. Results are for PDB ID: 5MPA.

| Method | SNR 0.1 | | SNR 0.05 | | SNR 0.03 | | SNR 0.01 | |
|---|---|---|---|---|---|---|---|---|
| | Rotation | Translation | Rotation | Translation | Rotation | Translation | Rotation | Translation |
| H-T align | 1.72±0.99 | 6.65±4.55 | 2.08±0.88 | 7.47±4.46 | 2.16±0.81 | 8.42±4.47 | 2.38±0.58 | 11.22±5.03 |
| F-A align | 1.73±1.01 | 6.69±4.71 | 1.97±0.94 | 7.26±4.67 | 2.24±0.79 | 8.59±4.69 | 2.39±0.56 | 11.33±4.88 |
| Gum-Net MP | 1.40±0.80 | 5.52±3.60 | 1.43±0.78 | 5.63±3.44 | 1.53±0.76 | 6.12±3.45 | 1.68±0.77 | 7.30±3.33 |
| Gum-Net AP | 1.05±0.69 | 4.28±2.92 | 1.19±0.73 | 4.78±3.04 | 1.37±0.73 | 5.64±3.22 | 1.66±0.77 | 7.10±3.27 |
| Gum-Net SC | 1.12±0.76 | 4.47±3.30 | 1.24±0.78 | 4.92±3.40 | 1.38±0.77 | 5.71±3.43 | 1.66±0.78 | 7.16±3.35 |
| Gum-Net | 0.68±0.64 | 2.61±2.46 | 0.89±0.72 | 3.13±2.68 | 1.12±0.72 | 4.25±2.73 | 1.46±0.78 | 6.22±3.38 |
| Jim-Net | 0.57±0.56 | 2.37±2.20 | 0.72±0.64 | 3.10±2.71 | 0.88±0.66 | 3.90±2.94 | 1.55±0.78 | 6.75±3.47 |
| SE(3)-Transformer | 2.45±0.45 | 5.18±0.84 | 2.44±0.80 | 6.77±0.52 | 1.98±0.33 | 6.18±0.74 | 2.08±0.80 | 5.22±0.45 |
| ConDor | 8.05±1.03 | 7.88±1.46 | 7.32±1.35 | 7.13±1.85 | 7.13±1.87 | 7.05±1.34 | 5.81±1.20 | 5.78±1.63 |
| Equi-Pose | 4.19±1.65 | 2.47±1.71 | 5.58±1.99 | 3.50±2.12 | 7.07±2.51 | 4.94±2.69 | 12.07±2.76 | 7.12±3.28 |
| BOE-ViT | 0.33±0.15 | 2.24±0.82 | 0.33±0.15 | 2.24±0.80 | 0.33±0.15 | 2.20±0.80 | 0.33±0.16 | 2.21±0.77 |
| **Ours** | **0.25±0.08** | **2.00±0.80** | **0.25±0.08** | **1.95±0.85** | **0.25±0.08** | **2.00±0.80** | **0.25±0.08** | **2.02±0.82** |

## F.4 Impact of Box size on subtomogram alignment

To assess robustness to changes in subtomogram size, we additionally evaluated both BOE-ViT and our model on $64^3$ subtomograms by reusing the same pretrained weights from $32^3$ pretraining and applying identical fine-tuning procedures for the downstream alignment task. We used $32^3$ as the pretraining box size because 3D foundation model pretraining scales cubically in memory and compute, making $64^3$ pretraining prohibitively expensive for our 904k-subtomogram datasets.

As shown in Table 18, our method consistently outperforms BOE-ViT across all SNR levels and at both box sizes. Notably, our model maintains stable rotation accuracy at $64^3$ (0.26°) and continues to outperform BOE-ViT (0.50°), yet the larger box size does not provide additional performance gains over $32^3$ and even increases translation error due to amplified noise. These findings demonstrate that our method is robust across scales and that $32^3$ remains the most effective and computationally efficient choice for subtomogram alignment.

Table 18: Subtomogram alignment accuracy under different box sizes. Each entry shows mean ± std for rotation and translation errors. $N^3@s$ Å indicates an $N \times N \times N$ subtomogram resampled to $s$ Å/voxel.

| Input | Method | SNR 0.1 | | SNR 0.05 | | SNR 0.03 | | SNR 0.01 | |
|---|---|---|---|---|---|---|---|---|---|
| | | Rotation | Translation | Rotation | Translation | Rotation | Translation | Rotation | Translation |
| $64^3$ @ 10Å | BOE-ViT | 0.50±0.15 | 9.21±2.67 | 0.50±0.15 | 9.34±2.65 | 0.50±0.15 | 9.28±2.70 | 0.50±0.15 | 9.26±2.65 |
| $64^3$ @ 10Å | **Ours** | **0.26±0.08** | **5.71±1.66** | **0.26±0.08** | **5.79±1.64** | **0.26±0.08** | **5.76±1.67** | **0.26±0.08** | **5.74±1.66** |
| $32^3$ @ 10Å | BOE-ViT | 0.33±0.15 | 2.58±0.93 | 0.34±0.15 | 2.45±0.87 | 0.34±0.15 | 2.50±0.89 | 0.34±0.15 | 2.54±0.91 |
| $32^3$ @ 10Å | **Ours** | **0.25±0.08** | **2.00±0.80** | **0.25±0.08** | **1.95±0.85** | **0.25±0.08** | **2.00±0.80** | **0.25±0.09** | **2.02±0.82** |

## F.5 Generalization to filamentous and asymmetric homotrimeric structures

To evaluate the model's ability to transfer beyond the globular complexes used during pre-training, we constructed three additional subtomogram datasets representing structurally diverse biological assemblies. These datasets were generated from three Protein Data Bank entries that span distinct geometric families: 6BNO (thin, extended helical actin filaments), 6DPU (large hollow microtubule assembly resembling membrane-associated cylindrical architectures), and 9KJR (asymmetric homotrimeric PNPase). Each structure was converted into a density map and processed with CryoEngine to create $32^3$ subtomograms at 10 Å resolution.

To highlight generalization, we directly fine-tuned both BOE-ViT and our model using the same pre-trained weights obtained from the original $32^3$ corpus, without any additional large-scale pretraining on these new structural classes (filamentous and asymmetric homotrimeric assemblies), which are structurally far from the pretraining families. Alignment was performed under the most challenging noise setting (SNR = 0.01), enabling a stringent assessment of transfer to filamentous and asymmetric homotrimeric assemblies.

As shown in Table 19, our model achieves consistently lower rotation and translation errors across all three structures, despite the substantial geometric differences from the pre-training distribution. These results demonstrate that the learned representation transfers effectively to new structural families and maintains strong robustness even at extremely low SNR levels.

Table 19: Alignment accuracy on three biological assemblies at SNR 0.01. Each table entry shows the mean and standard deviation of rotation and translation errors (SNR = 0.01).

| Method | Actin Filament | | Microtubule | | PNPase | |
|---|---|---|---|---|---|---|
| | Rotation | Translation | Rotation | Translation | Rotation | Translation |
| BOE-ViT | 0.34±0.18 | 3.07±1.14 | 0.38±0.23 | 3.12±1.13 | 0.34±0.18 | 3.50±1.38 |
| **Ours** | **0.25±0.07** | **3.07±0.86** | **0.256±0.07** | **2.98±0.86** | **0.25±0.07** | **3.02±0.89** |

## G  AVERAGING

### G.1  DETAILS OF SUBTOMOGRAM AVERAGING

Subtomogram averaging is a key step in cryo-ET analysis that combines many noisy 3D sub-volumes of the same macromolecular complex into a single high-resolution map. By reinforcing structural features shared across particles, averaging improves the signal-to-noise ratio and enables more accurate visualization of macromolecular architecture.

In a reference-free, non-parametric setting, an initial consensus map is iteratively refined by aligning each subtomogram to the current consensus and recomputing the average (Briggs, 2013; Wan & Briggs, 2016). This strategy avoids bias from external templates and progressively improves structural resolution. In our framework, the pretrained model provides equivariant alignment features for robust registration across particle poses, while invariance properties support unbiased consensus construction.

### G.2  EXPERIMENT SETTINGS

**Real Dataset Descriptions.**

- *S. cerevisiae* **80S Ribosome:** This dataset provides 3,120 subtomograms of the large, asymmetric 80S ribosome from purified *S. cerevisiae* (Bharat & Scheres, 2016). The subtomograms are rescaled to $32^3$ voxels with a 1.365 nm voxel size and a 30° missing wedge.

- **Tobacco Mosaic Virus (TMV):** This dataset consists of 2,742 subtomograms of the helical Tobacco Mosaic Virus (Kunz et al., 2015). The subtomograms are binned to $32^3$ voxels with a 1.080 nm voxel size and a 30° missing wedge.

- **Aldolase:** This dataset contains 400 subtomograms of purified rabbit muscle aldolase, a small tetrameric enzyme (Noble et al., 2018a). The subtomograms are rescaled to $32^3$ voxels with a 0.750 nm voxel size and a 30° missing wedge.

- **Insulin Receptor:** This dataset includes 400 subtomograms of the purified, insulin-bound human insulin receptor, a flexible membrane protein (Noble et al., 2018b). The subtomograms are rescaled to $32^3$ voxels with a 0.876 nm voxel size and a 45° missing wedge.

**Introduction of Baselines.**  Unlike alignment, subtomogram averaging lacks standardized baselines. General-purpose vision models such as equivariant neural networks or vision transformers are difficult to apply directly, since they do not natively support reference-free consensus construction across many subtomograms. Therefore, we focus on five straightforward baselines for subtomogram averaging.

- **H-T align** (Xu et al., 2012): Originally proposed as a Fourier-based rotational alignment method for low SNR and high tilt conditions, here it is adapted to averaging by iteratively aligning particles to a consensus using its predicted transformations.

- **F&A align** (Chen et al., 2013): An efficient spherical-harmonics–based alignment algorithm with Wiener-filtered corrections, which can also be extended to averaging through iterative consensus refinement.

- **Gum-Net** (X & M, 2020): An unsupervised CNN-based alignment model that can be extended to averaging by aligning particles to a consensus with its predicted transformations. It improves efficiency but is not specifically optimized for large-scale consensus reconstruction.

- **Jim-Net** (Zeng et al., 2021a): A multi-task CNN framework that couples clustering with alignment, which can in principle support averaging. However, its main focus is handling heterogeneity rather than resolution-oriented consensus refinement.

- **BOE-ViT** (Jiang et al., 2025): A vision transformer with equivariant design, originally developed for subtomogram alignment. When adapted to averaging, it benefits from robust orientation estimation but still inherits the alignment-centric objective.

### G.3 VISUALIZATION

As shown in Fig. 11, representative 2D cross-sections highlight the noisy and heterogeneous appearance of subtomograms across the four benchmark datasets. Despite this low SNR, iterative alignment-based averaging is able to progressively recover structural features, as visualized in Fig. 12, where the consensus maps become increasingly refined and converge toward the true underlying structures.

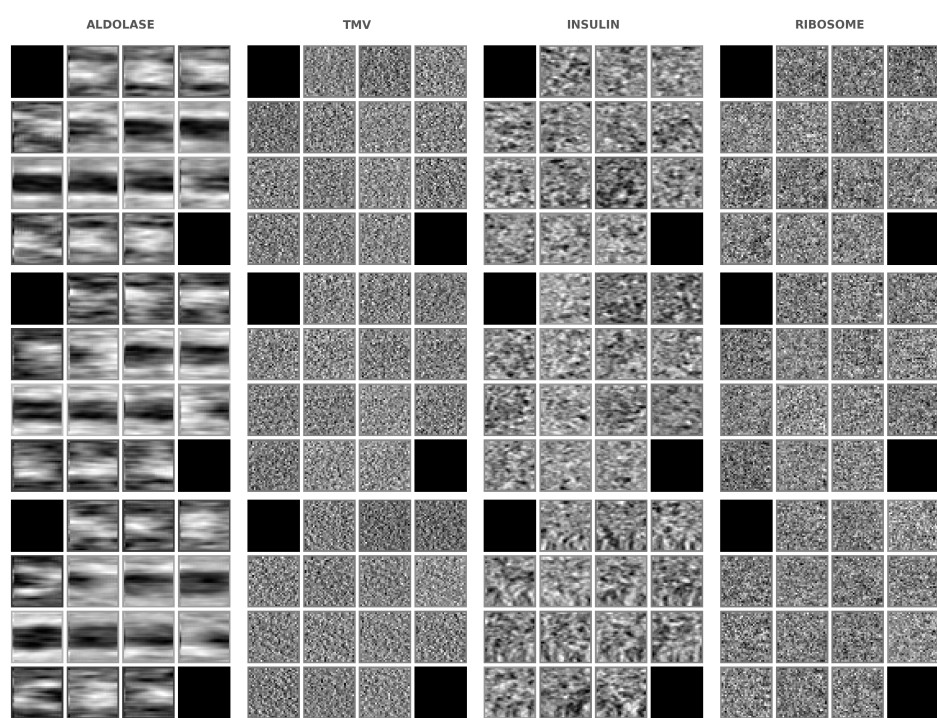

Figure 11: Representative 2D cross-sections of subtomograms from the four benchmark datasets.

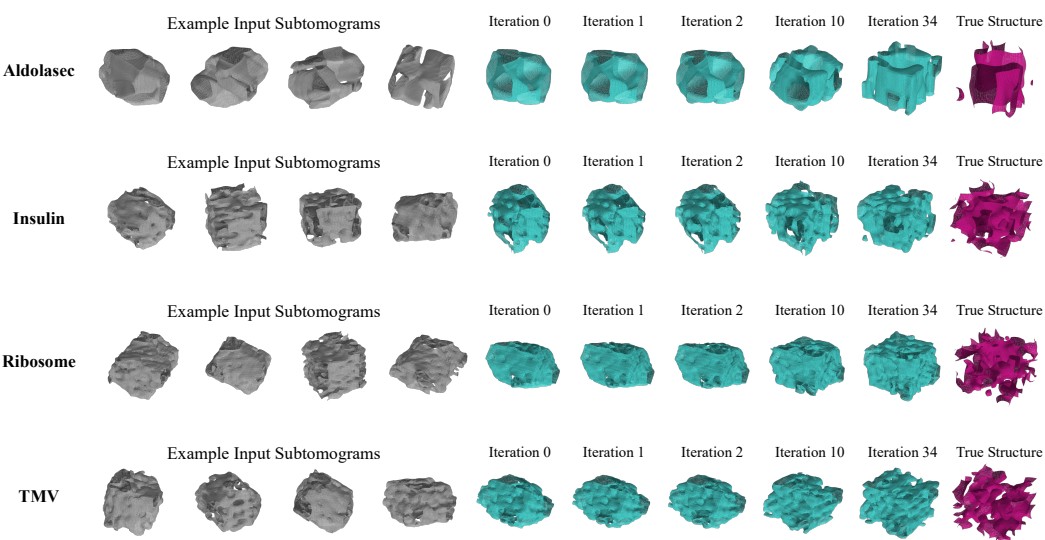

Figure 12: Visualization of iterative subtomogram averaging across four real datasets, showing progressive structural refinement from noisy inputs to averaged structure.

# H   Mechanism Analysis and Ablation Study

## H.1   Impact of Rotation Representation.

To better understand the impact of different geometric representations on alignment performance, we investigated three common representations within the SO(3) group: Euler angles, $\mathbb{R}^6$ with Gram-Schmidt orthonormalization ($\mathbb{R}^6 + GSO$) (Zhou et al., 2019), and $\mathbb{R}^9$ with singular value decomposition ($\mathbb{R}^9 + SVD$) (Levinson et al., 2020) . As shown in Table 20, Euler angles consistently outperform the other two representations across all SNR levels, achieving the lowest rotation and translation errors.

These results suggest that, despite its simplicity, the Euler angle representation provides more stable and accurate rotation alignment in the context of noisy subtomogram data. This advantage likely stems from the fact that Euler angles directly parameterize $SO(3)$ without requiring projection operations. In contrast, both the $\mathbb{R}^6 + GSO$ and $\mathbb{R}^9 + SVD$ approaches involve intermediate representations that must be mapped back to the rotation group via Gram-Schmidt or SVD, which can introduce numerical instability—especially under low SNR conditions. Such instability may degrade alignment accuracy when the input is corrupted by high levels of noise.

Table 20: Impact of rotation representation. Each cell reports the mean and standard deviation of the rotation error and translation error.

| Rot Representation | SNR 0.1 | | SNR 0.05 | | SNR 0.03 | | SNR 0.01 | |
|---|---|---|---|---|---|---|---|---|
| | Rotation | Translation | Rotation | Translation | Rotation | Translation | Rotation | Translation |
| Euler (Ours) | **0.25±0.08** | **1.97±0.81** | **0.25±0.08** | **1.95±0.83** | **0.25±0.08** | **1.97±0.81** | **0.25±0.08** | **2.03±0.82** |
| $\mathbb{R}^6$ + GSO | 2.01±1.21 | 2.84±1.09 | 1.98±1.17 | 2.83±1.11 | 1.99±1.19 | 2.89±1.18 | 2.00±1.22 | 2.95±1.22 |
| $\mathbb{R}^9$ + SVD | 1.92±1.18 | 2.75±1.04 | 1.90±1.14 | 2.73±1.06 | 1.90±1.13 | 2.79±1.14 | 1.94±1.13 | 2.85±1.16 |

## H.2   Evaluation of Representations in latent space.

To assess the effectiveness of our NRCL strategy in producing semantically meaningful features, we evaluate the learned representations using three common downstream evaluation protocols: K-nearest neighbor (KNN) classification, linear probing, and fine-tuning with a supervised classification head. As shown in Table 21, we can say that out model's better performance on classification task doesn't come from the classification head. Meanwhile, the performance drop observed in the KNN and linear probing settings under low SNR levels highlights the challenge of extracting discriminative features from noisy subtomograms.

Table 21: Accuracy↑ (%) by KNN, linear probing, and classification head under varying SNR levels over all datasets. The best result is highlighted in bold, the second best is underlined.

| Method | Evaluation Protocal | SNR 0.1 | SNR 0.05 | SNR 0.03 | SNR 0.01 |
|---|---|---|---|---|---|
| PVT | | 27.62 | 24.06 | 22.62 | 21.06 |
| SwinViT-S | KNN | 35.03 | 28.26 | 23.88 | 21.22 |
| Moco V3 | | 25.42 | 21.51 | 21.07 | 20.00 |
| **Ours** | KNN | 42.96 | 37.16 | 32.50 | 23.72 |
| | Linear Probing | 48.28 | 36.90 | 30.08 | 22.45 |
| | Classification Head | **67.42** | **53.13** | **40.10** | **27.50** |

## H.3   Impact of Noise-resilient Contrastive Learning.

To prove the effectiveness of the proposed noise-robust contrastive learning strategy, we compare the performance of our model with alternative self-supervised training strategy and full-finetune based transfer learning approaches. The results in Table 22 reveal that NRCL delivers the most consistent and substantial improvements across all SNR levels, regardless of the backbone architecture. For instance, under ViT-B, NRCL achieves noticeably higher performance than MAE (62.72% vs. 56.9%3 at SNR 0.1 and 47.14% vs. 41.13 %at SNR 0.05), and the margin becomes even more pronounced

under the most challenging noise conditions. When paired with APT-ViT, NRCL further pushes performance to the highest levels in the table (e.g., 67.42% at SNR 0.1 and 53.13% at SNR 0.05), clearly outperforming the BYOL counterpart. The relatively worse performance of MAE and BYOL demonstrates that existing self-supervised methods fall short in adapting models to the high-noise conditions typical of cryo-ET datasets. This proofs the effectiveness of our proposed NRCL strategy, which enables the model to learn meaningful and discriminative representations in latent space even under severe noise conditions.

Table 22: Impact of the NRCL strategy. Each cell reports the average recall across 5 different classes.

| Method | SNR 0.1 | SNR 0.05 | SNR 0.03 | SNR 0.01 |
|---|---|---|---|---|
| CryoEngine + MAE + ViT-B | 56.93 | 41.13 | 30.65 | 22.40 |
| CryoEngine + NRCL + ViT-B | 62.72 | 47.14 | 27.26 | 23.65 |
| CryoEngine + BYOL + APT-ViT | 44.65 | 34.45 | 30.03 | 22.33 |
| CryoEngine + NRCL + APT-ViT(Ours) | 67.42 | 53.13 | 40.10 | 27.50 |

## H.4 Parameter Exploration

**Impact of Attention Heads.** As shown in Table 23, the 8-head attention configuration consistently achieves the lowest rotation and translation errors across all SNR levels, with a mean rotation error of $0.25$–$0.26°$ and translation error around 2.00 voxels. Both reducing the number of heads (e.g., to 2 or 4) and increasing it to 10 lead to notable performance degradation, particularly in translation accuracy. These results suggest that 8 heads provide a favorable balance between expressiveness and stability, effectively modeling the spatial dependencies in 3D structures without overfragmenting attention or introducing redundant complexity.

**Impact of Hidden Dimension.** Table 24 reports performance under varying hidden dimensions. A moderate hidden size of 120 yields the best overall accuracy, while increasing it to 240 or 480 significantly deteriorates performance. Notably, the 480-dimension configuration incurs the largest rotation and translation errors, likely due to overfitting and increased optimization difficulty under noisy input conditions. These findings indicate that compact representations are not only more computationally efficient but also better suited for robust alignment in high-noise cryo-ET scenarios.

Table 23: Impact of attention heads on subtomogram alignment accuracy at different SNR levels. Each table entry shows the mean and standard deviation of both rotation and translation errors.

| Head Number | SNR 0.1 | | SNR 0.05 | | SNR 0.03 | | SNR 0.01 | |
|---|---|---|---|---|---|---|---|---|
| | Rotation | Translation | Rotation | Translation | Rotation | Translation | Rotation | Translation |
| 2 | 0.32±0.14 | 4.29±1.93 | 0.32±0.15 | 4.45±2.06 | 0.32±0.15 | 4.55±2.06 | 0.33±0.16 | 4.75±2.13 |
| 4 | 0.30±0.14 | 3.66±1.72 | 0.30±0.14 | 3.66±1.72 | 0.30±0.14 | 3.73±1.76 | 0.31±0.14 | 3.89±1.84 |
| 8 (ours) | **0.25±0.08** | **2.00±0.80** | **0.25±0.08** | **1.95±0.85** | **0.25±0.08** | **2.00±0.80** | **0.25±0.09** | **2.02±0.82** |
| 10 | 0.32±0.14 | 4.42±2.02 | 0.31±0.15 | 4.45±2.04 | 0.31±0.14 | 4.53±2.10 | 0.32±0.15 | 4.74±2.16 |

Table 24: Impact of hidden dimension on subtomogram alignment accuracy at different SNR levels. Each table entry shows the mean and standard deviation of both rotation and translation errors.

| Hidden dim | SNR 0.1 | | SNR 0.05 | | SNR 0.03 | | SNR 0.01 | |
|---|---|---|---|---|---|---|---|---|
| | Rotation | Translation | Rotation | Translation | Rotation | Translation | Rotation | Translation |
| 120 (ours) | **0.25±0.08** | **2.00±0.80** | **0.25±0.08** | **1.95±0.85** | **0.25±0.08** | **2.00±0.80** | **0.25±0.09** | **2.02±0.82** |
| 240 | 0.22±0.11 | 2.69±1.23 | 0.22±0.10 | 2.73±1.25 | 0.22±0.10 | 2.81±1.30 | 0.22±0.10 | 2.93±1.40 |
| 480 | 0.31±0.15 | 4.03±1.85 | 0.32±0.15 | 4.08±1.87 | 0.32±0.15 | 4.17±1.93 | 0.33±0.15 | 4.35±1.97 |

## H.5 Impact of APT-ViT on Classification

As in Table 25, the two pairs of experiments provide a clean architectural ablation to evaluate the contribution of APT-ViT. The results show that replacing ViT-B with APT-ViT consistently improves performance across nearly all SNR levels, regardless of the pretrain strategy. It demonstrates that the

Table 25: Impact of APT-ViT architecture on subtomogram classification. Each cell reports the average recall across 5 different classes.

| Method | SNR 0.1 | SNR 0.05 | SNR 0.03 | SNR 0.01 |
|---|---|---|---|---|
| CryoEngine + NRCL + ViT-B | 62.72 | 47.14 | 27.26 | 23.65 |
| CryoEngine + NRCL + APT-ViT(Ours) | 67.42 | 53.13 | 40.10 | 27.50 |
| CryoEngine + BYOL + ViT-B | 36.68 | 24.68 | 25.05 | 22.43 |
| CryoEngine + BYOL + APT-ViT | 44.65 | 34.45 | 30.03 | 22.33 |

improvement is attributable to the architectural design of APT-ViT, which provides direct evidence that APT-ViT offers more robust and better feature representation than traditional ViT, validating its effectiveness as a superior backbone for low-SNR subtomogram analysis.

# I  INTERPRETABILITY ANALYSIS

To better understand the model's decision-making process and enhance interpretability, we employ Grad-CAM (Selvaraju et al., 2017) to visualize spatial attention within the learned 3D representations. As shown in Fig. 13, the highlighted regions correspond to areas where the model concentrates when localizing the target particle within the input subtomogram. The visualizations are generated from slices extracted around the central structure of each particle. These results show that the model consistently attends to meaningful structural regions, indicating its ability to leverage informative 3D features for accurate alignment and classification.

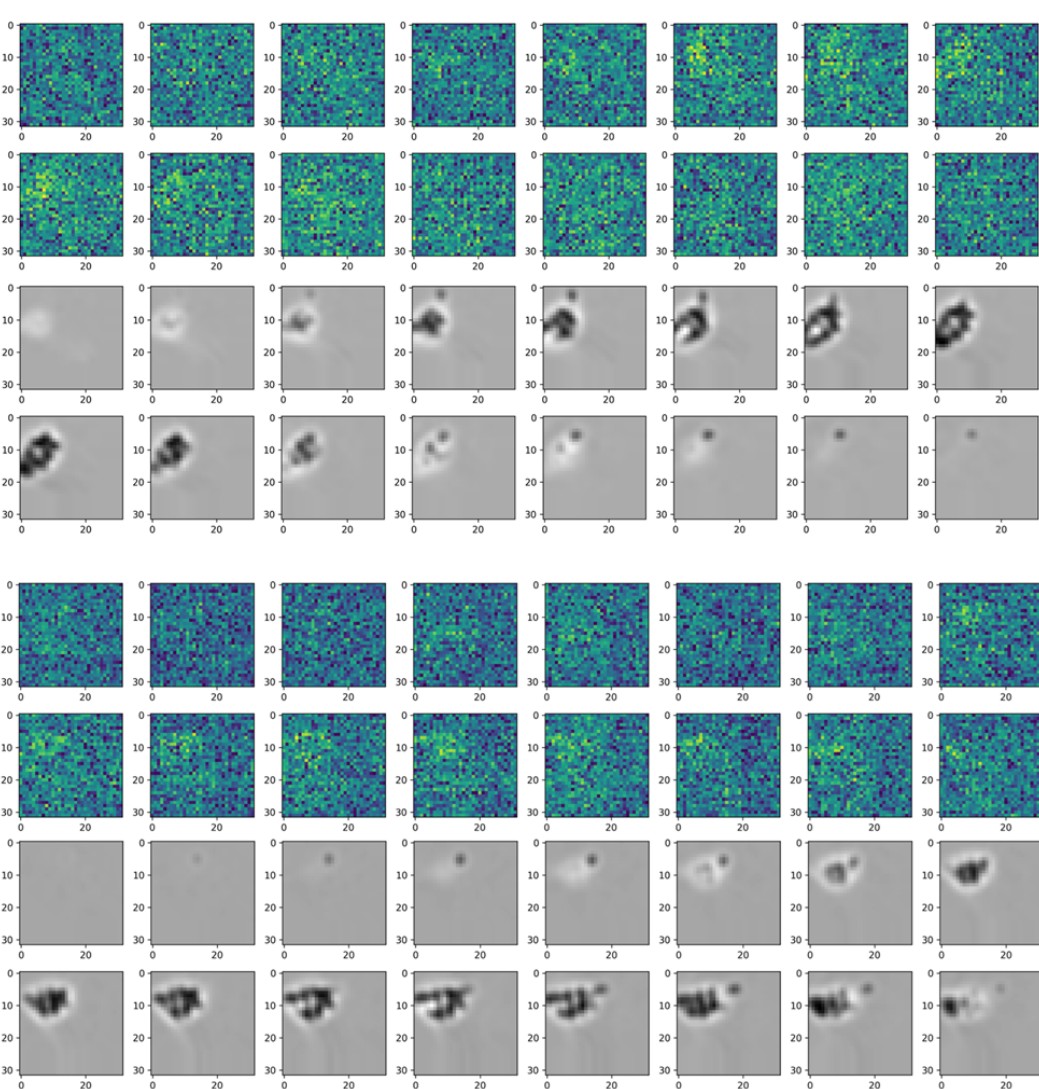

Figure 13: Visualization of gradient and its corresponding subtomogram slice. The highlighted region, which receives greater attention from the model, corresponds to the location of the particle.

## J BACKGROUND OF CRYO-ELECTRON TOMOGRAPHY

### J.1 INTRODUCTION OF CRYO-ET

Cryo-electron tomography (cryo-ET) is a sophisticated imaging method capable of producing detailed three-dimensional reconstructions of biological specimens at nanometer resolution (Doerr, 2017; Wagner J, 2017; Kühlbrandt, 2014). In cryo-ET, samples are first rapidly vitrified at temperatures below –150°C, which preserves their native cellular structures without the distortions introduced by chemical fixation or dehydration (Pfeffer & Mahamid, 2018). During imaging, the vitrified specimen is incrementally tilted under an electron beam through a defined angular range (typically ±60° with 1–3° increments), creating a tilt series of two-dimensional projection images (Dokland, 2009). These projection images are computationally reconstructed into a three-dimensional tomogram, representing a comprehensive spatial view of the cellular environment (Han et al., 2017; Mastronarde & Held, 2017). Subsequently, macromolecular complexes are identified and localized as subtomograms, which are extracted and classified into structurally homogeneous groups through subtomogram averaging, significantly enhancing structural details and interpretability (Fig. 14). This workflow makes cryo-ET exceptionally valuable for visualizing and understanding macromolecular architecture in its authentic cellular context, underpinning many advancements in in situ structural biology (Chen et al., 2019; Böhning & Bharat, 2021).

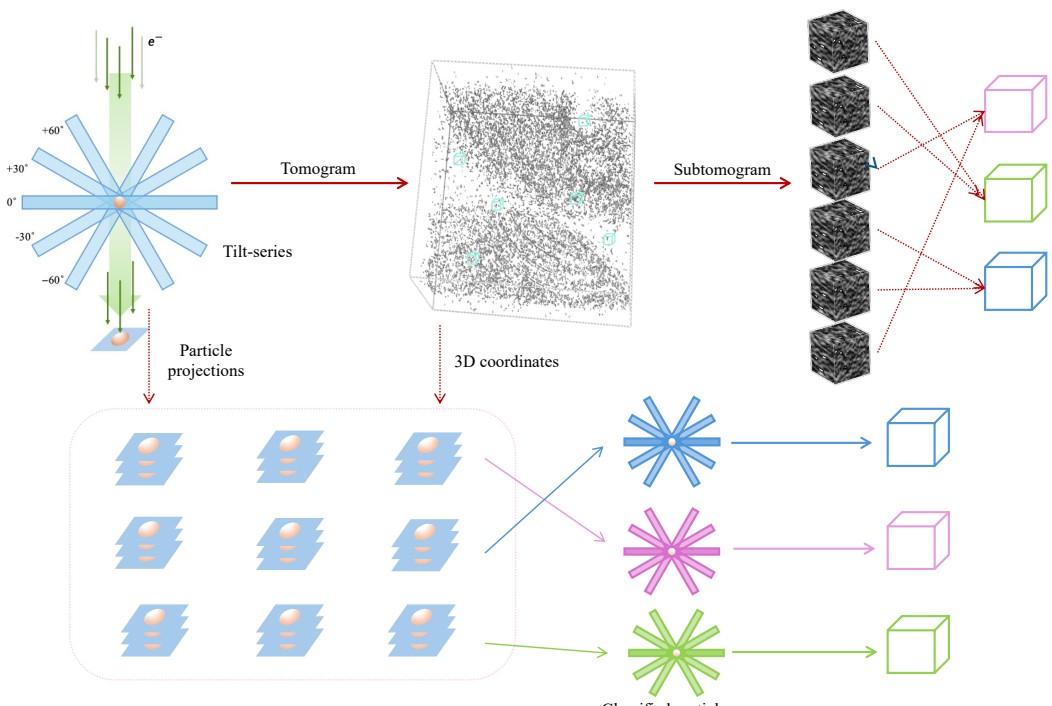

Figure 14: Overview of the cryo-ET workflow: tilt series acquisition, tomogram reconstruction, subtomogram extraction and classification.

### J.2 INTRODUCTION OF CRYO-ET SUBTOMOGRAM ALIGNMENT AND AVERAGING

Subtomogram alignment is an essential computational step in cryo-electron tomography (cryo-ET) that enables detailed structural analysis of macromolecules within their native cellular environments (Castaño-Díez & Zanetti, 2019; Pfeffer & Mahamid, 2018; Dokland, 2009). In this process, numerous structurally similar particles, termed subtomograms, are extracted from three-dimensional tomograms and aligned to generate averaged structures with significantly enhanced resolution and signal-to-noise ratio (Kim et al., 2023). However, subtomogram alignment presents substantial computational complexity arising from several unique factors inherent to cryo-ET data: firstly, the alignment

necessitates precise resolution of both translational and rotational parameters in three-dimensional space (Kovacs & Wriggers, 2002); secondly, cryo-ET datasets exhibit inherently lower signal-to-noise ratios compared to conventional single-particle cryo-EM, complicating accurate alignment (Danev et al., 2010); and finally, the analysis involves handling large volumetric datasets, demanding significant computational resources (Turk & Baumeister, 2020). To address these challenges, alignment algorithms typically utilize advanced cross-correlation-based strategies, including exhaustive angular searches or optimized rotational matching algorithms (X & M, 2020; Xu et al., 2012). Alignment proceeds iteratively, refining particle orientations and translations with each cycle until an optimal averaged structure is obtained (X & M, 2020; Zeng et al., 2021a). High-quality subtomogram alignment is therefore fundamental for extracting biologically meaningful structural insights and elucidating macromolecular assemblies directly within their physiological contexts (Chen et al., 2019).

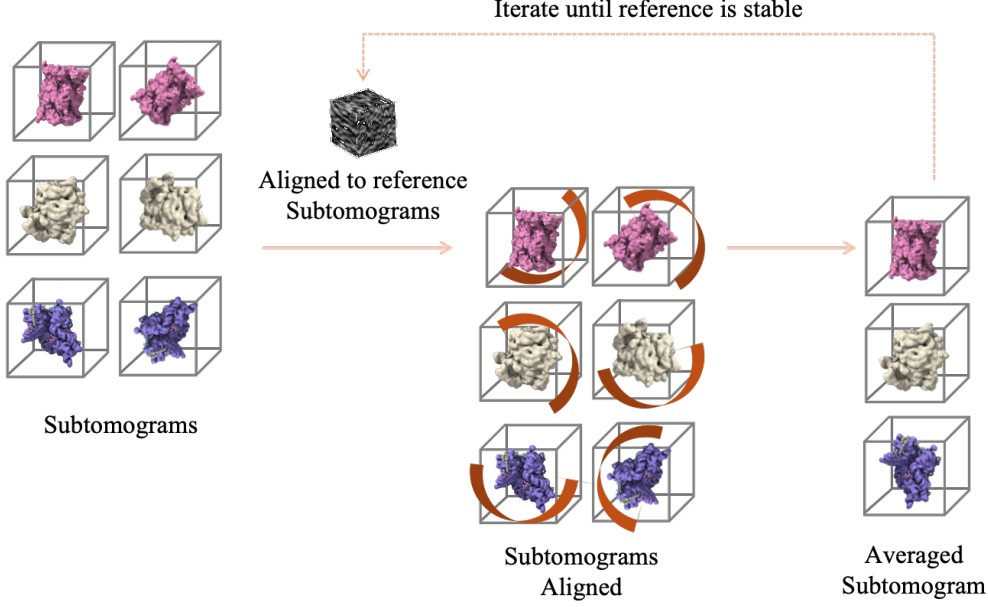

Figure 15: Workflow of subtomogram alignment and averaging.

### J.3 SUBTOMOGRAM PROCESSING PIPELINE

All subtomograms used for pretraining are generated by CryoEngine and then fed to APT-ViT without CTF correction or denoising. The synthetic subtomograms generation starting from experimentally validated PDB models, we first convert each structure into a 10 Å density map by placing element-specific isotropic Gaussian kernels at atomic centers and do low-pass filtering, pack multiple particles into a $500 \times 500 \times 200$ virtual volume using 3D Poisson-disk sampling with a safety margin and uniformly sampled SO(3) orientations, simulate a tilt series with realistic microscope geometry ($2°$ steps, default $-60°$ to $+60°$, extended to $-90°$ to $+90°$ for the pretraining set), reconstruct the tomogram via weighted back-projection, and finally extract $32^3$ subtomograms around each particle with offsets.

## K  USE OF LARGE LANGUAGE MODELS (LLMS)

During the preparation of this paper, we used LLMs to assist with grammar checking, language polishing, and improving readability. The model was not used for generating novel research ideas, experimental design, data analysis, or drawing conclusions. All content and claims in the paper are the sole responsibility of the authors.

