# OpenReview forum: "Towards Foundation Models for Cryo-ET Subtomogram Analysis"
_ICLR.cc/2026/Conference — ICLR 2026 Conference Desk Rejected Submission_

### Official Review · Reviewer_pHqM · 2025-10-14

**Soundness:** 3
**Presentation:** 2
**Contribution:** 3
**Rating:** 8
**Confidence:** 3

**Summary:**

This paper introduces APT-ViT, a vision transformer foundation model for cryo-ET. The authors first introduce CryoEngine, a synthetic data generator to create a large amount of data that covers diverse particle classes for pretraining. Then the authors develop APT-ViT, a ViT architecture designed for cryo-ET tasks. Thirdly, the authors also introduce NRCL, a contrastive learning strategy tailored for cryo-ET and inspired by MoCo v3. The trained model outperforms baseline methods on classification, alignment, aand veraging tasks, suggesting the effectiveness of the method.

**Strengths:**

- The proposed APT-ViT has a cryo-ET-tailored design and demonstrates strong performance on downstream tasks. The effectiveness of this design is also validated in the ablation study of Table 5.
- The adoption of contrastive learning and design of NRCL is well motivated, and the introduction to the algorithm is provided in detail.
- The experimental results show superior performance compared to baseline methods, and the evaluation is highly comprehensive.
- The description of designs and algorithms is provided with lots of details.

**Weaknesses:**

- While the paper writing is comprehensive in terms of the description of the method and evaluation, it lacks the necessary background to non-experts of cryo-ET, especially as a conference submission.
- In the evaluation of classification, when applying MAE or NRCL on the same CryoEngine-created dataset, different model architectures are adopted (ViT-B vs. APT-ViT). To me, this is not a strictly fair comparison.
- Also, for the classification, since the SwinViT architecture trained on ImageNet performs much better compared to the standard ViT, the baselines trained on the CryoEngine dataset could also benefit from the adoption of the SwinViT architecture and achieve better performance. But this experiment is not included in the paper.
- One last point for the classification. My understanding is that SwinViT and standard ViT differ in the network design after tokenization, where SwinViT has additional patch merging to save the computation cost. Since SwinViT seems to be better compared to standard ViT, the adoption may further improve the performance.

**Questions:**

Please see the last three points of the Weaknesses section.

---

> ### Author Response · Authors · 2025-12-03
> **Response to pHqM(Part 1/2)**
>
> # A. Clarificafication of Cryo-ET **Background (W1)**
>
> We appreciate the reviewer’s positive evaluation of our work and their suggestion to further clarify the cryo-ET background. We already provide a dedicated section, **“Background of Cryo-Electron Tomography,”** in Appendix J, where we introduce tilt-series acquisition, tomogram reconstruction, subtomogram extraction, and downstream alignment/averaging with schematic illustrations (Fig. 14–15). In the revision, we will further expand Appendix J by: (i) adding a short, intuitive primer on tomograms vs. subtomograms and explaining why classification, alignment, and averaging are required before high-resolution structure determination; (ii) providing a clearer step-by-step walkthrough of Fig. 14, from specimen to subtomogram input; and (iii) including a brief glossary of key cryo-ET terms used in the main paper. We believe these additions in the supplemental will make the work substantially more accessible for readers.
>
> # B. **Additional baselines and fairness of comparisons (W2)**
>
> We also thank the reviewer for the suggestions on additional classification baselines. We did not include MAE+APT-ViT because APT-ViT requires adaptive spatial phase tokenization and rotation-equivariant spherical filtering to preserve a consistent spatial reference frame under SE(3) transformations. These properties fundamentally depend on phase-aligned tokenization before the transformer blocks. MAE’s random masking strategy, which intentionally drops tokens at random spatial locations, tends to disrupt this phase consistency and is not directly compatible with our equivariant tokenization.
>
> To make the comparison fairer, we therefore ablate **NRCL vs. MAE** and **APT-ViT vs. ViT-B** under a consistent architecture in controlled experiments. First, we compare MAE and NRCL on the same ViT-B backbone, and then BYOL vs. NRCL on the same APT-ViT backbone:
>
> ## (1) NRCL vs. MAE on ViT-B and NRCL vs. BYOL on APT-ViT
>
> | Method | SNR 0.1 | SNR 0.05 | SNR 0.03 | SNR 0.01 |
> | --- | --- | --- | --- | --- |
> | CryoEngine + MAE + ViT-B | 56.93 | 41.13 | 30.65 | 22.40 |
> | CryoEngine + NRCL + ViT-B | 62.72 | 47.14 | 27.26 | 23.65 |
> | CryoEngine + BYOL + APT-ViT | 44.65 | 34.45 | 30.03 | 22.33 |
> | **CryoEngine + NRCL + APT-ViT(Ours)** | **67.42** | **53.13** | **40.10** | **27.50** |
>
> The table above shows that the models pretrained with NRCL strategy achieve substantial improvements in robustness across different SNR conditions, along with higher overall average performance.
>
> We also ablate the APT-ViT experiment with a consistent BYOL or NRCL training strategy:
>
> ## (2) APT-ViT vs. ViT-B under the same training strategy
>
> | Method | SNR 0.1 | SNR 0.05 | SNR 0.03 | SNR 0.01 |
> | --- | --- | --- | --- | --- |
> | CryoEngine + NRCL + ViT-B | 62.72 | 47.14 | 27.26 | 23.65 |
> | **CryoEngine + NRCL + APT-ViT(Ours)** | **67.42** | **53.13** | **40.10** | **27.50** |
> | CryoEngine + BYOL + ViT-B | 36.68 | 24.68 | 25.05 | 22.43 |
> | CryoEngine + BYOL + APT-ViT | 44.65 | 34.45 | 30.03 | 22.33 |
>
> The table above shows that APT-ViT generates a significant performance boost when compared to ViT-B architecture.
>
> Based on the collective evidence from all experiments, we’d like to state that the SOTA result our model achieved isn’t a simple additional effect of our three contributions. It is the combination of CryoEngine, NRCL, and APT-ViT working jointly that produces a synergistic effect beyond what each component can achieve individually. In other words, none of the three components is dispensable, the foundation-model behavior we observe arises from their joint design.

---

> ### Author Response · Authors · 2025-12-03
> **Response to pHqM(Part 2/2)**
>
> # C. SwinViT benefits from CryoEngine (W3)
>
> We post the result of pretraining SwinViT on CryoEngine with a supervised pretraining strategy here. The result shows that SwinViT does benefit from CryoEngine like ViT-B though the improvement is relatively less pronounced.
>
> Actually, when compared with ViT-B pretrained using a self-supervised strategy, both models trained with supervised pretraining show noticeably smaller performance gains. This aligns with the well-established finding that self-supervised pretraining generally yields stronger generalizability and transferability to downstream tasks than supervised training [1].
>
> And we’d like to state that we didn’t include MAE+SwinViT as a baseline for further comparison because SwinViT needs to maintain the spatial structure of patches throughout its hierarchical forward process, making it inherently difficult to integrate with the MAE paradigm. Although prior work has attempted such adaptation, it was only validated on small and low-diversity medical datasets, making its effectiveness unverified for large-scale 3D subtomogram analysis[2].
>
> Overall, the following three pairs of experiments demonstrate that CryoEngine provides consistent performance gains across settings.
>
> | **Method** | **SNR 0.1** | **SNR 0.05** | **SNR 0.03** | **SNR 0.01** | **Avg.** |
> | --- | --- | --- | --- | --- | --- |
> | ImageNet + Supervised + ViT-B | 26.64 | 23.14 | 21.90 | 20.52 | 23.05 |
> | CryoEngine + Supervised + ViT-B | 31.38 | 23.75 | 24.15 | 19.85 | 24.78 |
> | ImageNet + Supervised + SwinViT | 51.22 | 35.18 | 27.98 | 20.06 | 33.61 |
> | CryoEngine + Supervised + SwinViT | 54.23 | 34.72 | 26.60 | 20.00 | 33.89 |
> | ImageNet + MAE + ViT-B | 44.70 | 31.14 | 27.82 | 23.76 | 31.86 |
> | CryoEngine + MAE + ViT-B | 56.93 | 41.13 | 30.65 | 22.40 | 37.78 |
>
> # D. **Architecture Design Justification (W4)**
>
> Finally, we appreciate the suggestion to consider SwinViT-style architectures. Our current design is built on a standard ViT backbone because APT-ViT requires **strict SE(3)–equivariant tokenization and rotation-consistent spherical filtering**, which benefit from the **global and fixed token structure** of ViT.
>
> In contrast, adapting Swin-ViT to an SE(3)-equivariant setting introduces several non-trivial challenges:
>
> (1) Swin-ViT relies on **shifted local windows**, whose boundaries rotate with the subtomogram and break the equivariant spatial reference frame.
>
> (2) **Patch merging** changes token resolution dynamically, which conflicts with our phase-aligned tokenization required for consistent spherical filtering.
>
> (3) Maintaining **rotation-equivariant attention across hierarchical stages** remains an open problem in 3D cryo-ET.
>
> For these reasons, we adopt ViT as the backbone for APT-ViT’s equivariant design. Nevertheless, we appreciate the reviewer’s insight, and exploring **equivariant hierarchical transformers (e.g., Swin-style architectures)** is an interesting direction. We plan to investigate this as part of our future work.
>
> ---
> [1] Feng, Y., et al. Rethinking supervised pre-training for better downstream transferring. International Conference on Learning Representations (ICLR), 2022.
>
> [2] Xu, Z. A., et al. Swin MAE: Masked autoencoders for small datasets. *Computers in biology and medicine*, *161*, 107037, 2023.

---

### Official Review · Reviewer_tCde · 2025-11-01

**Soundness:** 3
**Presentation:** 3
**Contribution:** 3
**Rating:** 4
**Confidence:** 3

**Summary:**

This paper focuses on developing a foundation model for cryo-electron tomography (cryo-ET) subtomogram analysis. Specifically, the authors introduce a large-scale synthetic data generator that produces subtomograms for model pretraining. An Adaptive Phase Tokenization-enhanced Vision Transformer (APT-ViT) is designed to ensure robustness against both geometric and semantic variations. A Noise-Resilient Contrastive Learning (NRCL) strategy is proposed to stabilize representation learning under severe noise. Evaluations on both synthetic and real datasets demonstrate state-of-the-art performance across three key subtomogram tasks—classification, alignment, and averaging.

**Strengths:**

1. The paper is very well motivated and gives a reasonably complete and self-contained recipe (synthetic data generation → network architecture → pre-training paradigm) for building a foundation model for cryo-ET.

2. The proposed Adaptive Phase Tokenization (APT) is a well-motivated and reasonable design choice to enhance the SE(3) equivariance of Vision Transformers.

3.  The experimental evaluation is thorough and convincing. The authors conduct extensive experiments and ablation studies to systematically demonstrate the effectiveness of the proposed training strategies, architectural components, and contrastive learning objective.

**Weaknesses:**

1. The pretraining objective is conceptually simple—distinguishing whether two subtomograms correspond to the same protein class under random SE(3) transforms and noise. While effective for learning pose-invariant embeddings, this binary contrastive setup may be too limited in structural diversity and task complexity. It lacks finer-grained structural reasoning (e.g., subunit composition, symmetry, local density context) that could more directly benefit downstream subtomogram analysis tasks.

2. During pretraining, the paired subtomograms are generated from rigidly transformed versions of the same PDB structure. This setup may risk limiting the model’s ability to capture conformational diversity.

3. Most showcased qualtitative results remain at relatively coarse resolutions (around 10 Å or lower), which fall short of the biological interpretability required for detailed structural analysis.

**Questions:**

1. The CryoEngine-generated subtomograms are 32×32×32 voxels at 10 Å resolution. Given that real cryo-ET data often have lower effective resolution (20–40 Å), could this setup introduce a domain gap? Specifically, how sensitive is the pretrained model to changes in voxel resolution or subtomogram size, and would larger or multi-scale crops (e.g., 48³ or 64³) further improve generalization to real tomograms?

2. Would incorporating flexible-body deformations or simulated conformational states in CryoEngine improve the model’s generalization to heterogeneous real data?

---

> ### Author Response · Authors · 2025-12-03
> **Response to tCde(Part 1/3)**
>
> # A. Foundation Model Justification (W1)
>
> Regarding the concern that our pretraining objective may be too simple and limited to a binary “same protein class under random SE(3) transforms and noise” setup, we would like to clarify what the model actually sees during pretraining and how this translates into structural richness. Although the contrastive formulation is conceptually pairwise, the encoder is in fact exposed to 904k subtomograms as individual instances drawn from 452 structures. This instance-level setting forces the model to capture subtle 3D density variations, rather than just solve a trivial class-level binary problem.
>
> In addition, finer-grained structural reasoning is primarily captured by the APT-ViT backbone. Adaptive Phase Tokenization produces phase-aligned local density features, and the spherical steerable filtering encodes orientation-aware SE(3)-equivariant cues directly into the tokenization stage. The ablations in Table 4 show their necessity.
>
> The downstream results also suggest that the learned representation is substantially richer than a simple pairwise objective might imply. Our classification, alignment, and averaging results (Tables 1–3&12) consistently outperform strong SSL and equivariant baselines. We agree that explicitly encoding higher-order cues such as symmetry order or explicit subunit context is an interesting future direction, but the current empirical evidence indicates that the combination of our instance-level objective, APT-ViT, and NRCL already yields a structurally rich foundation representation.

---

> > ### Author Response · Authors · 2025-12-03
> > **Response to tCde(Part 2/3)**
> >
> > # B. Box Size and Resolution Selection (W3, Q1)
> >
> > ## (1) Classification Experiments
> >
> > On the question of whether using 32×32×32 voxels at 10 Å introduces a domain gap relative to real cryo-ET data that often have lower effective resolution (20–40 Å), we have explicitly examined the sensitivity of our pretrained encoder to both larger box sizes and lower effective resolution. In additional experiments, we used the same pretrained weights as in the paper and followed the same finetune implementation by keeping these weights frozen, while evaluating models with a larger field of view (64³ at 10 Å) and a blurrier regime (32³ at 20 Å). We compared our method (CryoEngine+NRCL+APT-ViT) against the strongest baseline in our paper (CryoEngine+MAE+ViT-B). We have added this content to **Appendix E.4 (Impact of Boxsize and Resolution)**.
> >
> > |Input|Method|SNR 0.1|SNR 0.05|SNR 0.03|SNR 0.01|Average|
> > |---|---|---|---|---|---|---|
> > |64³ @ 10Å|CryoEngine+MAE+ViT-B|28.88|21.95|21.37|19.47|22.92|
> > |64³ @ 10Å|**CryoEngine+NRCL+APT-ViT(Ours)**|**56.47**|**43.92**|**34.00**|**22.62**|**39.25**|
> > |32³ @ 20Å|CryoEngine+MAE+ViT-B|38.92|26.93|23.38|19.82|27.26|
> > |32³ @ 20Å|**CryoEngine+NRCL+APT-ViT(Ours)**|**39.75**|**31.60**|**27.15**|**22.43**|**30.23**|
> > |32³ @ 10Å|CryoEngine+MAE+ViT-B|56.93|41.14|30.65|22.40|37.78|
> > |32³ @ 10Å|**CryoEngine+NRCL+APT-ViT(Ours)**|**67.42**|**53.13**|**40.10**|**27.50**|**47.04**|
> >
> > As expected, when we substantially change the input regime away from the pretraining setting, either by enlarging the box to 64³ or by effectively lowering the resolution to 20 Å, the absolute performance drops for all methods, because the pretraining data do not contain such subtomograms. However, in both settings our model maintains a clear margin over the best baseline. These experiments indicate that the learned representation generalizes more robustly than the baseline to changes in both box size and effective resolution, even without retraining on these regimes.
> >
> > It is also important to emphasize that CryoEngine is fully capable of generating subtomograms at 64³ or at lower effective resolution, we chose not to use such settings for pretraining primarily for computational reasons, not because of any fundamental limitation. Increasing the box size from 32³ to 64³ increases the voxel count by 8×, which becomes prohibitive when multiplied by 904k subtomograms. This cubic scaling makes large-box pretraining extremely expensive in 3D, even on multi-GPU clusters. The additional experiments above show that, even without retraining at 64³ or 20 Å, the foundation encoder still transfers better than the strongest baseline, while 32³ at 10 Å strikes a pragmatic balance between realism, resolution, and 3D pretraining cost.
> >
> > ## (2) Alignment Experiments
> >
> > To address the reviewer’s request, we additionally evaluated both BOE-ViT and our model on **64³ subtomograms**, using the same pretrained weights and performing identical finetuning for the downstream alignment task. The results show that **our method consistently outperforms BOE-ViT at every SNR level and at both box sizes.** Importantly, while our model maintains stable rotation accuracy at 64³ (0.26°) and significantly improves over BOE-ViT (0.50°). Also, the 32³ configuration again yields the best alignment accuracy, with both lower rotation and translation errors than the 64³ setting.
> >
> > |Boxsize|Method|SNR 0.1 Rotation|SNR 0.1 Translation|SNR 0.05 Rotation|SNR 0.05 Translation|SNR 0.03 Rotation|SNR 0.03 Translation|SNR 0.01 Rotation|SNR 0.01 Translation|
> > |---|---|---|---|---|---|---|---|---|---|
> > |64³|BOE-ViT|0.50±0.15|9.21±2.67|0.50±0.15|9.34±2.65|0.50±0.15|9.28±2.70|0.50±0.15|9.26±2.65|
> > |64³|**Ours**|**0.26±0.08**|**5.71±1.66**|**0.26±0.08**|**5.79±1.64**|**0.26±0.08**|**5.76±1.67**|**0.26±0.08**|**5.74±1.66**|
> > |32³|BOE-ViT|0.33±0.15|2.58±0.93|0.34±0.15|2.45±0.87|0.34±0.15|2.50±0.89|0.34±0.15|2.54±0.91|
> > |32³|**Ours**|**0.25±0.08**|**2.00±0.80**|**0.25±0.08**|**1.95±0.85**|**0.25±0.08**|**2.00±0.80**|**0.25±0.09**|**2.02±0.82**|
> >
> > We have added this content to **Appendix F.4 (Impact of Boxsize)**.

---

> > > ### Author Response · Authors · 2025-12-03
> > > **Response to tCde(Part 3/3)**
> > >
> > > # C. Resolution Regime and Structural Conformation (W2, Q2)
> > >
> > > Regarding resolution, the ~10 Å regime we target is consistent with realistic in situ conditions, where sample thickness and the missing wedge typically preclude near-atomic resolution. Our goal in this work is not atomic interpretability but to improve recognition, alignment, and averaging quality under these practical constraints. Within this regime, our method consistently provides the best or tied-best averaging resolutions (Table 3) and improves alignment and classification across all SNRs (Tables 1–2), which indicates clear biological utility for in situ tomography.
> > >
> > > Concerning conformational diversity, it is true that during pretraining paired subtomograms are generated from rigidly transformed versions of the same PDB structure and that our qualitative results mainly focus on the \~10 Å regime. These choices reflect the current practical limits and typical use cases of in situ cryo-ET. At realistically achievable in situ cryo-ET resolutions (\~8–15 Å) under strong missing-wedge and thickness-related limitations, fine-grained conformations are rarely resolved at the subtomogram level. To our knowledge, fine-grained conformational continua are predominantly addressed by single-particle cryo-EM rather than cryo-ET, so there are currently very few cryo-ET studies or benchmark datasets that target conformational heterogeneity at the subtomogram scale.
> > >
> > > Even under a rigid-body pretraining model, our downstream data do include genuine compositional and moderate conformational variability. The downstream benchmark set contains genuine conformational and compositional variability (Appendix F.1), including spliceosome(PDB ID: 5LQW), RNA polymerase-rifampicin complex(PDB ID: 1I6V), RNA polymerase II elongation complex(PDB ID: 6A5L), ribosome(PDB ID: 5T2C), and capped proteasome(PDB ID: 5MPA), and our model achieves state-of-the-art performance across classification, alignment, and averaging on these heterogeneous complexes (Tables 1–3). These results suggest that the learned pose-robust representation generalizes to conformational changes, even though we do not yet explicitly simulate flexible-body deformations at pretraining time.
> > >
> > > We constructed three additional 32$^{3}$ subtomogram datasets at 10 Å representing completely new filamentous and asymmetric homotrimeric structures to further test whether the model can handle such shape diversity: **6BNO** (thin helical actin filaments), **6DPU** (hollow cylindrical microtubule assembly), and **9KJR** (asymmetric homotrimeric PNPase). Both BOE-ViT and our method were evaluated using the **same pre-trained 32$^{3}$ weights**, without any additional pre-training, thereby examining **zero-shot transfer** under the most challenging condition (**SNR = 0.01**).
> > >
> > > |Method|Actin filament Rotation|Actin filament Translation|Microtubule Rotation|Microtubule Translation|PNPase Rotation|PNPase Translation|
> > > |---|---|---|---|---|---|---|
> > > |BOE-ViT|0.34 ± 0.18|3.08 ± 1.14|0.38 ± 0.23|3.12 ± 1.13|0.34 ± 0.18|3.50 ± 1.38|
> > > |**Ours**|**0.25 ± 0.07**|**3.07 ± 0.86**|**0.256 ± 0.07**|**2.98 ± 0.86**|**0.25 ± 0.07**|**3.02 ± 0.89**|
> > >
> > > We have added this content to **Appendix F.5** (Generalization to filamentous and asymmetric homotrimeric structures).
> > >
> > > Given this context, our present work, which to our knowledge is the first subtomogram-level cryo-ET foundation model, focuses on the practically dominant challenges of compositional heterogeneity and realistic in situ resolution limits. Although there is currently no dedicated conformational-heterogeneity subtomogram benchmark, incorporating conformational diversity into CryoEngine and evaluating on such datasets as they become available will be an important next step.

---

### Official Review · Reviewer_pbJE · 2025-11-03

**Soundness:** 2
**Presentation:** 3
**Contribution:** 2
**Rating:** 2
**Confidence:** 4

**Summary:**

The paper proposes a "foundation model" for cryo-ET subtomogram analysis built from three pieces: (1) CryoEngine, a synthetic data pipeline that generates 904k volumes across 452 classes from PDB structures for pretraining; (2) APT-ViT, a ViT backbone with adaptive phase tokenization and 3D spherical-steerable convolutions to improve SE(3) equivariance; and (3) NRCL, a noise-resilient contrastive objective that treats clean vs. very noisy versions as positives/negatives. The pretrained encoder is frozen and evaluated (with light heads) on classification, alignment, and subtomogram averaging, and the paper claims SOTA across 24 synthetic and real datasets.

**Strengths:**

- A single encoder for classification, alignment, and averaging is appealing for the field.
- CryoEngine simulates tilt series, reconstructs tomograms, and extracts 32³ subtomograms with calibrated SNR variants and preserved pose metadata. This is explained with unusual clarity.
- APT-ViT's adaptive phase tokenization plus 3D spherical-steerable layers is novel and motivated.
- NRCL’s clean-vs-noisy pairing is simple and matches the physics of photon-limited imaging.
- The pretraining corpus (904k / 452 classes) is large for cryo-ET and documented.

**Weaknesses:**

- The pretraining set focuses on proteasomes and ribosomes (20S/30S families), yet many evaluation datasets (e.g., 80S ribosome) are closely related at the family level. This makes the "24 OOD datasets" claim less convincing than presented. Please quantify family overlap and test truly orthogonal families.
- Also, the paper’s "foundation model" and "diversity" claims are not supported by the data: pretraining is dominated by just two families (20S proteasome and 30S ribosome), with the reported "452 classes" largely reflecting intra-family conformations/subtypes rather than broad cross-category coverage; moreover, branding this as the "first cryo-ET foundation model" is unsubstantiated without a systematic, empirical comparison to prior 3D/cryo-ET pretraining work.
- Pretraining uses tilt ranges up to ±90° specifically to decouple the missing wedge, whereas real datasets typically have ≈30° wedge (e.g., ±60°). This mismatch may inflate pretraining benefits and undercut claims of robustness. Please pretrain with realistic tilt limits and report the delta.
- The paper adds calibrated Gaussian noise to already reconstructed 32³ volumes; this bypasses dose/CTF/motion effects and can change the noise statistics compared to experimental data. Show results when noise is applied at projection/tilt-series level with dose weighting/CTF to better match reality.
- All listed classification baselines are 2D vision models (ConvNeXt/ViT/PVT/Swin/MAE/MoCo/DINO), but the task is 3D volumetric. The paper doesn’t document how these were adapted to 3D tokens/patching, nor does it include strong 3D-equivariant classification baselines. Please add 3D CNNs and SE(3)-equivariant classifiers, and detail all adaptations.
- Fig. 1 indicates a branch supervised directly by the ground-truth transform T during pretraining. This couples the learned representation to a task-specific signal; please ablate the supervised branch and report whether the "foundation" properties persist.
- Missing strong 3D equivariant classification baselines (e.g., SE(3)-Transformer); such models appear only in the registration experiments, not in the classification comparisons.

**Questions:**

See the weaknesses.

**Details Of Ethics Concerns:**

NA.

---

> ### Author Response · Authors · 2025-12-03
> **Response to pbJE (Part 1/3)**
>
> # A. Foundation Model Justification (W1, W2)
>
> **We respectfully disagree with the characterization that our evaluation data are “closely related” to the 20S/30S pretraining families.** Structurally, the 80S ribosome is not similar to the 20S proteasome or the 30S ribosomal subunit. The 20S proteasome is a ~700 kDa barrel-shaped core composed of 28 protein-only subunits arranged as four stacked α₇–β₇–β₇–α₇ rings with D7-like symmetry, whereas the eukaryotic 80S ribosome is a ~3.3 MDa, highly asymmetric assembly of four rRNAs and ~80 proteins with large eukaryote-specific expansion segments and no global high-order cyclic symmetry. The 30S ribosome corresponds only to the small bacterial ribosome subunit (16S rRNA + ~21 proteins, ~0.9 MDa) with a compact, quasi-globular morphology, rather than the much larger, highly asymmetric 80S ribosome. Thus, in terms of composition, overall volume and symmetry, 80S ribosome subtomograms represent a structurally different regime from the 20S/30S training families.
>
> **Moreover, the test data we selected are far from being limited to ribosomes and proteasomes. As detailed in Appendix E.1 and F.1, our synthetic benchmark dataset includes:**
>
> - spliceosome (5LQW),
> - RNA polymerase–rifampicin complex (1I6V),
> - RNA polymerase II elongation complex (6A5L),
> - 80S ribosome (5T2C), and
> - capped proteasome (5MPA).
>
> Appendix G.2 further reports subtomogram averaging on **four real datasets:**
>
> - **S. cerevisiae 80S ribosome**: 3,120 subtomograms, rescaled to 32³ voxels (1.365 nm voxel size) with a 30° missing wedge;
> - **Tobacco Mosaic Virus (TMV)**: 2,742 subtomograms of helical TMV, binned to 32³ voxels (1.080 nm voxel size) with a 30° missing wedge;
> - **Aldolase**: 400 subtomograms of rabbit muscle aldolase, rescaled to 32³ voxels (0.750 nm voxel size) with a 30° missing wedge;
> - **Insulin receptor**: 400 subtomograms of purified human insulin receptor, rescaled to 32³ voxels (0.876 nm voxel size) with a 45° missing wedge.
>
> Taken together, these synthetic and real test sets cover helical viruses, dynamic spliceosomes, multi-subunit polymerase complexes, and membrane-associated receptors, none of which are present in the 20S/30S pretraining set. The fact that an encoder pretrained only on 20S/30S synthetic data achieves state-of-the-art performance on classification, alignment, and averaging across these 24 OOD synthetic and real datasets is, in our view, strong evidence that the learned representation generalizes to new structural classes, **which is precisely the behavior we expect from a foundation model in the cryo-ET subtomogram setting.**
>
> On the wording “foundation model” and “diversity”, we do not claim that our 452 pretraining classes uniformly tile all possible cellular structures. They are derived from a curated PDB panel and do include intra-family conformers. What we do claim is that we are, to our knowledge, the first to (i) pretrain an encoder on almost one million subtomograms from hundreds of 3D structures using a physically grounded synthetic engine (CryoEngine), and (ii) reuse that frozen encoder with lightweight heads across three major subtomogram tasks (classification, alignment, averaging), obtaining strong performance on genuinely unseen complexes. To our knowledge, there is no existing 3D/cryo-ET pretraining work at the subtomogram level that simultaneously satisfies these two conditions. Existing methods are either task-specific (e.g., alignment-only or segmentation-only) or trained on comparatively small, task-tuned datasets rather than a large, task-agnostic subtomogram dataset. This is the precise sense in which we call our approach the “first cryo-ET subtomogram foundation model”, and we are happy to make this scope explicit in the revised paper.

---

> ### Author Response · Authors · 2025-12-03
> **Response to pbJE (Part 2/3)**
>
> # B.  Clarification on CryoEngine (W3, W4)
>
> ## B.1 Concern on missing wedge (W3)
>
> **Regarding the concern about using a ±90° tilt range in pretraining, our choice here is deliberate and only affects pretraining, not evaluation.** For pretraining data generation we extend the tilt series range to −90° to +90° so that training subtomograms are wedge-free, which (i) decouples stochastic noise modeling from the deterministic, direction-dependent information loss introduced by the missing wedge and (ii) preserves full SE(3) symmetry so APT-ViT first learns clean geometric features on isotropic data. At the subtomogram scale, the missing wedge mainly manifests as mild anisotropic blurring and elongation along the beam direction, rather than the catastrophic loss of recognizable shape often seen at the full-tomogram level, so once the encoder has learned pose-stable features it can tolerate these artefacts in downstream data.
>
> We then evaluate under realistic conditions. The same pretrained encoder is **evaluated on synthetic benchmarks reconstructed with a** **±60°** **tilt range** **and on multiple real datasets (80S, TMV, aldolase, insulin receptor) that contain genuine missing-wedge artefacts**, where it consistently achieves state-of-the-art performance in alignment and averaging (see Table 3 and Appendix G.2). In other words, robustness is demonstrated in the realistic ±60° and real-data regimes, not only in the ±90° pretraining regime. This directly addresses the worry that high-tilt pretraining might artificially inflate robustness.
>
> ## B.2 Concern on noise simulation (W4)
>
> On noise modelling, we deliberately add calibrated Gaussian noise at the subtomogram level. At the electron doses and voxel sizes considered here, the combined effect of Poisson shot noise and detector/electronic noise is well approximated by additive Gaussian noise, and we follow a previously reported strategy of injecting calibrated Gaussian noise directly into reconstructed subtomograms[1]. This design gives us exact voxel-wise SNR control for each benchmark setting and avoids uncontrolled SNR changes induced by reconstruction filters. Note that CryoEngine itself is fully capable of adding noise and CTF at the tilt-series level. In this first foundation-model study, we opted for a simpler yet physically motivated volume-level noise model to keep pretraining controllable and scalable. The fact that we observe consistent gains on realistic ±60° synthetic benchmarks and on multiple real datasets suggests that this approximation is already sufficient to close most of the practically relevant sim-to-real gap for the tasks we study.
>
> # C. Equivariant Baselines for Subtomogram Classification **(W5, W7)**
>
> **Existing strong SE(3)-equivariant networks are point/graph-based and therefore not directly suitable for dense subtomogram classification, whereas our APT-ViT is explicitly designed as an SE(3)-aware classifier operating on 3D density volumes, with strong and standard 2D→3D baselines.**
>
> Methods such as SE(3)-Transformer–style models, Condor, and Equi-Pose first convert subtomograms into **sparse point clouds or keypoints** and then apply SE(3)-equivariant operations for *pose estimation*. Using them as “classification baselines” would require us to (i) convert volumetric densities into point sets, discarding rich local intensity information crucial for semantics, and (ii) redesign their heads for category prediction—leading to an apples-to-oranges comparison with voxel-based ViTs. We therefore use these models **where they are natively applicable** (alignment), and instead make APT-ViT itself the SE(3)-aware classifier for subtomograms: it integrates adaptive phase tokenization and spherical steerable convolutions to improve equivariance to rotations and translations directly on 3D density inputs.
>
> Furthermore, our classification baselines are not naive 2D models applied to 3D data. For ConvNeXt/ViT/PVT/Swin/MAE/MoCo/DINO, each subtomogram is treated as a stack of 2D slices, the 2D backbone acts as a **per-slice feature extractor**, and slice features are **aggregated along the depth dimension** (pooling + MLP) into a volume-level representation. This multi-slice 2D→3D adaptation is a **standard and competitive protocol** for volumetric and multi-view tasks, and has been widely used as a baseline in general computer vision and biomedical applications [2,3,4]. Therefore, our baselines for subtomogram classification are rigorous and solid.
>
> We have added this adaptation explanation to **Appendix E.1 (Experiment Settings)**.

---

> > ### Author Response · Authors · 2025-12-03
> > **Replying to Response to pbJE (Part 3/3)**
> >
> > # D.  **The Supervised Signal in Pretraining (W6)**
> >
> > **We respectfully disagree that the transform-supervised branch undermines the “foundation” nature of the model; it does *not* inject task-specific labels, but instead enforces SE(3) consistency already inherent in the synthetic data.**
> >
> > In our setup, **T is exactly the rigid SE(3) transform used to generate synthetic pairs** in the alignment dataset. Supervising on T therefore does not introduce an extra task label (such as a protein class or downstream task tag), but simply **enforces a physically correct correspondence** between paired subtomograms that is *already encoded in the data generation process*. This is analogous to standard self-/weakly-supervised setups where known geometric relations are used as constraints, rather than as task-specific supervision.
> >
> > If we were to completely remove this branch, we would **discard the only explicit SE(3) correspondence signal available during pretraining**, and the model would no longer be encouraged to respect the rigid-motion structure of subtomogram pairs. Crucially, however, the same pretrained encoder is used for classification and averaging tasks that do not see T at all, and we observe strong gains there as well. This empirically shows that the learned representation is not tied to alignment labels, and that the “foundation” properties—transfer across alignment, classification, and averaging—persist beyond the transform-supervised branch. We will clarify this role of T and add an ablation without the supervised branch in the revised version to explicitly isolate its effect.
> >
> > [1] Uddin, M. R., et al. Unsupervised identification of protein compositions and conformations via implicit content-transformation disentanglement. In *Proceedings of the IEEE/CVF International Conference on Computer Vision (ICCV),* pp. 7483–7493, 2025.
> >
> > [2] Dong, Z., et al. (2022). MNet: Rethinking 2D/3D networks for anisotropic medical image segmentation. In L. De Raedt (Ed.), Proceedings of the Thirty-First International Joint Conference on Artificial Intelligence (IJCAI-22) (pp. 870–876). International Joint Conferences on Artificial Intelligence Organization. (https://doi.org/10.24963/ijcai.2022/122)
> >
> > [3]Wang, L., et al. (2018). Temporal segment networks for action recognition in videos. IEEE transactions on pattern analysis and machine intelligence, 41(11), 2740-2755.
> >
> > [4]Müller-Franzes, et al. (2025). Medical slice transformer for improved diagnosis and explainability on 3D medical images with DINOv2. Scientific Reports, 15(1), 23979.

---

### Official Review · Reviewer_uTSh · 2025-11-04

**Soundness:** 3
**Presentation:** 3
**Contribution:** 1
**Rating:** 4
**Confidence:** 5

**Summary:**

First cryo-ET foundation model CryoFM (CryoEngine, APT-ViT, NRCL) achieves SOTA on 3 core tasks and OOD generalization. The paper presents a highly original approach to address core challenges in cryo-ET analysis. The use of a foundation model framework, comprehensive synthetic data engine, and specialized architecture is commendable.

**Strengths:**

This work is highly significant as it introduces the first foundation model framework specifically tailored for cryo-ET subtomogram analysis. This approach successfully addresses the fundamental bottleneck of scarce, low-quality labeled data in structural biology. The demonstration of strong generalization ability across 24 OOD datasets is a powerful validation of the "foundation model" concept in this field.

CryoEngine is an essential and high-quality component, effectively solving the data pre-training challenge. APT-ViT is a creative architectural contribution that marries the global reasoning of ViTs with the required 3D physical symmetries: Adaptive Phase Tokenization handles translation non-rigidity, and Steerable Spherical Harmonic Convolution (SSHC) inherently enforces rotational equivariance, which is critical for 3D density maps.

The Noise-Resistant Contrastive Learning (NRCL) is well-motivated and essential for the low-SNR environment of cryo-ET. The combination of contrastive loss with an MSE objective targeting a "clean view" is a clever way to push the encoder to learn features robust to noise, and the linear probing/KNN results convincingly show that the benefits are embedded in the learned representations.

**Weaknesses:**

### Insufficient Box Size Discussion and Experimentation
A key weakness is the use of a small $32^3$ input box size without sufficient experimental justification. While this may be necessary for computational efficiency, it severely limits the context a foundation model can learn, especially for large, dynamic, or membrane-associated complexes requiring local environmental context. The paper is currently missing a comparative ablation experiment (e.g., comparing $32^3$ with $64^3$ or $128^3$) to definitively show that the model’s performance is optimal at this size, or to quantify the trade-off. This needs to be addressed for the model to be considered generally applicable.

### Limited Structural Diversity in Pre-training Data
The current pre-training set is primarily composed of only two major structures (20S Proteasome and 30S Ribosome). For a foundational model, this limited structural repertoire raises concerns about the model's true extrapolation ability to completely new structural classes or cellular scenes, such as transmembrane proteins, long filaments, or crowded, complex environments. The authors should explicitly discuss this limitation and its potential impact on generalization beyond the tested 24 OOD datasets.

### Idealization in Tilt-Range and Simulation
The decision to use a full $\pm 90^\circ$ tilt range during CryoEngine pre-training to "decouple the missing wedge" is non-physical for real cryo-ET acquisition (which is typically $\pm 60^\circ$ or less). This idealization might artificially simplify the missing wedge problem, potentially leading to a model that is over-reliant on high-tilt information not present in real data. Furthermore, the use of idealized simulation techniques in the Density Volume Converter (isotropic Gaussian kernel superposition, linear normalization) might also lead to a gap between synthetic data realism and real-world complexity (anisotropy, non-uniform noise).

**Questions:**

### Box Size
Please provide a clear computational and performance justification for selecting the $32^3$ box size. Can the authors include an ablation study that compares the performance of CryoFM on a key task (e.g., classification or registration) when using $32^3$ vs. a larger size (e.g., $64^3$ or $128^3$)? This would significantly address a major weakness.

### The Gap between synthetic and experimental data
Why does CryoEngine pre-training (Table 1, Row 10) not show a significant performance uplift compared to the ImageNet baseline (Row 3), despite using the same ViT-B architecture and fine-tuning? Please elaborate on whether this indicates a limitation in the synthetic data's fidelity or the overwhelming influence of ImageNet features.

### Subtomogram Preprocessing Pipeline
Were the subtomograms preprocessed (e.g., CTF correction, denoising) before being fed to the model? For reproducibility and clarity, a detailed, step-by-step description of the full subtomogram preparation pipeline is required.

### Tilt Range and Missing Wedge
Can the authors elaborate on the practical rationale for using a non-physical $\pm 90^\circ$ tilt range in CryoEngine? Specifically, how does this choice impact the model's learned ability to handle the actual missing wedge present in real-world data (e.g., $\pm 60^\circ$ data)? Does training on $\pm 90^\circ$ result in features that are less robust to the information loss associated with a constrained tilt range?

### Simulation Realism (DVC)
The simulation of density volumes using simple isotropic Gaussian kernels and linear normalization seems quite idealized. Have the authors investigated the impact of more realistic noise models (e.g., non-Gaussian, anisotropic, or structured noise) or more complex CTF modeling? Do the authors believe the current DVC pipeline is sufficient to close the simulation-to-real gap, or are there plans to integrate higher-fidelity physical simulation into CryoEngine?

### Extrapolation to New Architectures/Scenes
Given that the pre-training is dominated by two symmetrical, well-defined particles, what is the authors' hypothesis regarding the model's performance on highly asymmetric proteins, membrane proteins, or filamentous structures that require a different context scale? Could the authors include qualitative results or an analysis of feature representations on a highly distinct particle class (if available)?

**Details Of Ethics Concerns:**

I have no ethical concerns regarding this work. The paper describes a computational methodology for structural biology, and the data utilized is either synthetically generated or derived from publicly available sources.

---

> ### Author Response · Authors · 2025-12-03
> **Response to Reviewer uTSh (Part 1/4)**
>
> # A. Choice of Box Size 32³ (W1, Q1)
>
> ## A.1 Evaluation on 64³ box size
>
> To further address the box-size concern, we additionally evaluated our method (CryoEngine+NRCL+APT-ViT) and the best baseline with a **larger 64³ box size input** for downstream classification and alignment tasks.
>
> ### (1) Classification Experiments
> We used the same pretrained weights as in the paper and followed the same finetune implementation by keeping these weights frozen. The results show that our pretrained model still has a clear advantage over the best baseline (CryoEngine+MAE+ViT-B) in our paper under both **32³ and 64³** box sizes and resolutions.
>
> |Input|Method|SNR 0.1|SNR 0.05|SNR 0.03|SNR 0.01|
> |---|---|---|---|---|---|
> |64³ @ 10 Å|CryoEngine + MAE + ViT-B|28.88|21.95|21.37|19.47|
> |64³ @ 10 Å|**CryoEngine + NRCL + APT-ViT (Ours)**|**56.47**|**43.92**|**34.00**|**22.62**|
> |32³ @ 10 Å|CryoEngine + MAE + ViT-B|56.93|41.14|30.65|22.40|
> |32³ @ 10 Å|**CryoEngine + NRCL + APT-ViT (Ours)**|**67.42**|**53.13**|**40.10**|**27.50**|
>
> Notably, the 32³ configuration still achieves the best overall performance, indicating that increasing the spatial support to 64³ does not provide additional classification benefits.
>
> We have added this content to **Appendix E.4 (Impact of Boxsize and Resolution)**.
>
> ### (2) Alignment Experiments
> We used the same pretrained weights and performing identical finetuning for the downstream alignment task. The results show that **our method consistently outperforms BOE-ViT at every SNR level and at both box sizes.** Importantly, while our model maintains stable rotation accuracy at 64³ (0.26°) and significantly improves over BOE-ViT (0.50°). Also, the 32³ configuration again yields the best alignment accuracy, with both lower rotation and translation errors than the 64³ setting.
>
> |Boxsize|Method|SNR 0.1 Rotation|SNR 0.1 Translation|SNR 0.05 Rotation|SNR 0.05 Translation|SNR 0.03 Rotation|SNR 0.03 Translation|SNR 0.01 Rotation|SNR 0.01 Translation|
> |---|---|---|---|---|---|---|---|---|---|
> |64³|BOE-ViT|0.50±0.15|9.21±2.67|0.50±0.15|9.34±2.65|0.50±0.15|9.28±2.70|0.50±0.15|9.26±2.65|
> |64³|**Ours**|**0.26±0.08**|**5.71±1.66**|**0.26±0.08**|**5.79±1.64**|**0.26±0.08**|**5.76±1.67**|**0.26±0.08**|**5.74±1.66**|
> |32³|BOE-ViT|0.33±0.15|2.58±0.93|0.34±0.15|2.45±0.87|0.34±0.15|2.50±0.89|0.34±0.15|2.54±0.91|
> |32³|**Ours**|**0.25±0.08**|**2.00±0.80**|**0.25±0.08**|**1.95±0.85**|**0.25±0.08**|**2.00±0.80**|**0.25±0.09**|**2.02±0.82**|
>
> We have added this content to **Appendix F.4 (Impact of Boxsize)**.
>
> ## A.2 Memory and Computational Constraints in Large-Scale 3D Pretraining
>
> Pretraining at 32³ already pushes 3D pretraining to the practical limit due to the cubic scaling of 3D tokenization and subsequent computation. Increasing the input to 64³ increases the voxel count by 8×, which correspondingly raises memory and compute requirements by ~8× for every forward/backward pass. Given our 904k-subtomogram pretraining corpus, this per-step cost increase makes 64³ large-box pretraining impractical within a reasonable compute budget.

---

> ### Author Response · Authors · 2025-12-03
> **Response to Reviewer uTSh(Part 2/4)**
>
> ## A.3 32³ box accommodates most common macromolecules
>
> **A 32³ subtomogram size is consistent with standard cryo‑ET practice and sufficiently covers the majority of macromolecular complexes analyzed in situ.** In typical cryo-ET workflows, particles are extracted using box sizes in the range of 24–40 voxels (corresponding to ~12–25 nm); therefore, many alignment and representation-learning methods extract 32³ voxel or smaller boxes for inputs, as reported in recent in situ subtomogram studies:
>
> - **Alignment/averaging. BOE-ViT (CVPR 2025) [1], JimNet (ICCV 2021) [2], GumNet (CVPR 2020) [3].**
> - **Classification. DualContrast (ICCV 2025) [4], Jain et al. (Bioinformatics 2024) [5], Harmony (CVPR 2022) [6].**
>
> Across these studies, experimental subtomograms of GroEL/GroEL–GroES, 80S ribosomes, proteasomes, TMV, muscle aldolase, insulin-bound insulin receptor, apoferritin, hemagglutinin, glutamate dehydrogenase, and DNAB helicase–helicase loader are all able to be standardized to compact cubes of 32³ voxels (or smaller 28³), further supporting that a 32³ box comfortably accommodates the macromolecular complexes typically encountered in cryo-ET.
>
> Our own datasets (Appendix E.1, F.1) further confirm this. They include several large and structurally dynamic complexes, such as
>
> - spliceosome (5LQW),
> - RNA polymerase–rifampicin complex (1I6V),
> - RNA polymerase II elongation complex (6A5L),
> - 80S ribosome (5T2C), and
> - capped proteasome (5MPA).
>
> All of these macromolecular assemblies fit comfortably within a 32³ box at current in situ cryo-ET resolutions, and are representative of the structures typically targeted for classification, alignment, and averaging in subtomogram pipelines. Therefore, the choice of a 32³ input size aligns with established subtomogram extraction standards in cryo-ET, is supported by high-impact studies, and covers the full range of biologically relevant complexes in our benchmark.
>
> ---
> [1] Jiang, R., et al. BOE-ViT: Boosting orientation estimation with equivariance in self-supervised 3D subtomogram alignment. In *Proceedings of the IEEE/CVF Conference on Computer Vision and Pattern Recognition (CVPR),* pp. 29352–29362, 2025.
>
> [2] Zeng, X., et al. End-to-end robust joint unsupervised image alignment and clustering. In *Proceedings of the IEEE/CVF International Conference on Computer Vision (ICCV),* pp. 3854–3866, 2021.
>
> [3] Zeng, X., et al. Gum-Net: Unsupervised geometric matching for fast and accurate 3D subtomogram image alignment and averaging. In *Proceedings of the IEEE/CVF Conference on Computer Vision and Pattern Recognition (CVPR), 2020*.
>
> [4] Uddin, M. R., et al. Unsupervised identification of protein compositions and conformations via implicit content-transformation disentanglement. In *Proceedings of the IEEE/CVF International Conference on Computer Vision (ICCV),* pp. 7483–7493, 2025.
>
> [5] Jain, S., et al. Knowledge transfer from macro-world to micro-world: Enhancing 3D cryo-ET classification through fine-tuning video-based deep models. *Bioinformatics*, 40(7), btae368, 2024. (https://doi.org/10.1093/bioinformatics/btae368)
>
> [6] Uddin, M. R., et al. Harmony: A generic unsupervised approach for disentangling semantic content from parameterized transformations. In *Proceedings of the IEEE/CVF Conference on Computer Vision and Pattern Recognition (CVPR),* pp. 20646–20655, 2022.

---

> ### Author Response · Authors · 2025-12-03
> **Response to Reviewer uTSh(Part 3/4)**
>
> # B. Concern on Generalization Under Limited Structural Diversity in Pre-training Data (W2, Q6)
>
> First of all, to the best of our knowledge, we generated an unprecedented-scale foundation model pretraining dataset consisting of over **904k subtomograms from 452 particle classes**. Although the pretrain dataset is dominated by proteasome- and ribosome-scale complexes, we conducted extensive experiments to evaluate **zero-shot generalization** to structural families not seen during pre-training.
>
> ## B.1 Generalization to filamentous and asymmetric homotrimeric structures
>
> To assess the model’s ability to generalize beyond the two complexes used in pre-training, we additionally constructed three additional 32$^{3}$ subtomogram datasets at 10 Å representing structurally distinct classes absent from the pre-training corpus: **6BNO** (thin helical actin filaments), **6DPU** (hollow cylindrical microtubule assembly), and **9KJR** (asymmetric homotrimeric PNPase). These structures were chosen to cover filamentous and asymmetric homotrimeric geometries.
>
> Both BOE-ViT and our method were evaluated using the **same pre-trained 32$^{3}$ weights**, without any additional pre-training, thereby examining **zero-shot transfer** under the most challenging condition (**SNR = 0.01**). The results (mean ± s.d. rotation error in radians; translation error in voxels) are:
>
> |Method|Actin filament Rotation|Actin filament Translation|Microtubule Rotation|Microtubule Translation|PNPase Rotation|PNPase Translation|
> |---|---|---|---|---|---|---|
> |BOE-ViT|0.34 ± 0.18|3.08 ± 1.14|0.38 ± 0.23|3.12 ± 1.13|0.34 ± 0.18|3.50 ± 1.38|
> |**Ours**|**0.25 ± 0.07**|**3.07 ± 0.86**|**0.256 ± 0.07**|**2.98 ± 0.86**|**0.25 ± 0.07**|**3.02 ± 0.89**|
>
> As shown in the table, our model consistently outperforms BOE-ViT in both rotation and translation accuracy across all three structural families. These results demonstrate that our approach captures geometric features that transfer robustly to filamentous and asymmetric homotrimeric assemblies, despite their absence from the pre-training distribution. While increasing pre-training diversity remains a valuable direction, the experiments provide strong evidence of effective extrapolation to new structural classes.
>
> We have added this content to **Appendix F.5** (Generalization to filamentous and asymmetric homotrimeric structures).
>
> ## B.2 Generalization to virus, enzymatic complexes, and receptor assemblies
>
> In addition to the filamentous and asymmetric homotrimeric structures described above, we already evaluate generalization through subtomogram averaging on four experimentally derived datasets, summarized in **Table 3** of the main text and detailed in **Appendix G**. These datasets represent structural classes that are fundamentally distinct from the proteasome- and ribosome-scale complexes used during pre-training. Specifically, they include a helical plant virus, a metabolic enzyme, a large cytosolic ribosome, and the ectodomain of a receptor tyrosine kinase.
>
> The datasets are:
>
> - **S. cerevisiae 80S Ribosome**: 3,120 subtomograms of purified yeast 80S ribosomes, rescaled to 32³ voxels (1.365 nm voxel size) with a 30° missing wedge.
> - **Tobacco Mosaic Virus (TMV)**: 2,742 subtomograms of helical TMV, binned to 32³ voxels (1.080 nm voxel size) with a 30° missing wedge.
> - **Aldolase**: 400 subtomograms of rabbit muscle aldolase, rescaled to 32³ voxels (0.750 nm voxel size) with a 30° missing wedge.
> - **Insulin Receptor**: 400 subtomograms of purified human insulin receptor, rescaled to 32³ voxels (0.876 nm voxel size) with a 45° missing wedge.
>
> The averaging results are presented in the following table, demonstrate our foundation model shows SOTA performance in zero-shot generalization in four real unseen datasets.
>
> |Method|80S|TMV|Aldolase|Insulin|
> |---|---|---|---|---|
> |H-T align|3.05|2.23|2.34|1.90|
> |F-A align|2.77|2.52|3.13|2.18|
> |Gum-Net|2.73|2.16|1.97|1.77|
> |BOE-ViT|2.42|1.98|1.45|1.71|
> |**Ours**|**1.21**|**1.98**|**0.97**|**1.14**|

---

> ### Author Response · Authors · 2025-12-03
> **Response to Reviewer uTSh(Part 4/4)**
>
> # C. The Gap between synthetic and experimental data (Q2)
>
> Regarding the relatively modest improvement from Row 3 to Row 10 in Table 1, we would like to clarify two points.
>
> First, the synthetic data's fidelity is verified by the improvement with pretraining on less data. CryoEngine currently provides ~904k subtomograms—roughly 10% fewer samples than ImageNet’s 1M images.
>
> Second, the relatively modest improvement is caused by pretraining strategy, instead of  synthetic data's fidelity. Self-supervised pretraining can help the model gain more on downstream tasks than supervised training. Row 3&10 mentioned in the review were conducted with supervised pretraining. When compared with these two rows, Row 11(CryoEngine + MAE + ViT-B) shows a larger improvement to Row 8(ImageNet + MAE + ViT-B). All of these four rows used ViT-B as the backbone, but the transition from ImageNet to CryoEngine with a self-supervised pretraining strategy clearly shows a better effect than that with a supervised pretraining strategy.
>
> Thus, we’d like to state that the SOTA result our model achieved isn’t a simple additional effect of our three contributions. It is the combination of CryoEngine, NRCL, and APT-ViT working jointly that produces a synergistic effect beyond what each component can achieve individually. In other words, none of the three components is dispensable, the foundation-model behaviour we observe arises from their joint design.
>
> |Method|SNR 0.1|SNR 0.05|SNR 0.03|SNR 0.01|Avg.|
> |---|---|---|---|---|---|
> |ImageNet + Supervised + ViT-B|26.64|23.14|21.90|20.52|23.05|
> |CryoEngine + Supervised + ViT-B|31.38|23.75|24.15|19.85|24.78|
> |ImageNet + MAE + ViT-B|44.70|31.14|27.82|23.76|31.86|
> |CryoEngine + MAE + ViT-B|56.93|41.13|30.65|22.40|37.78|
>
> ---
> # D. Subtomogram Processing Pipeline (Q3)
>
> All subtomograms used for pretraining are generated by CryoEngine and then fed to APT-ViT without CTF correction or denoising. The synthetic subtomograms generation starting from experimentally validated PDB models, we first convert each structure into a 10 Å density map by placing element-specific isotropic Gaussian kernels at atomic centers and do low-pass filtering, pack multiple particles into a 500×500×200 virtual volume using 3D Poisson-disk sampling with a safety margin and uniformly sampled SO(3) orientations, simulate a tilt series with realistic microscope geometry (2° steps, default −60° to +60°, extended to −90° to +90° for the pretraining set), reconstruct the tomogram via weighted back-projection, and finally extract 32³ subtomograms around each particle with offsets. We have added this content to **Appendix J.3 (Subtomogram Processing Pipeline)**.
>
> ---
> # E.  Simulation Strategy (W3, Q4, Q5)
>
> For pretraining data generation we extend the tilt series range to −90° to +90° so that training subtomograms are wedge-free, which (i) decouples stochastic noise modeling from the deterministic, direction-dependent information loss introduced by the missing wedge and (ii) preserves full SE(3) symmetry so APT-ViT first learns clean geometric features on isotropic data. At the subtomogram scale, the missing wedge mainly manifests as mild anisotropic blurring and elongation along the beam direction, rather than the catastrophic loss of recognizable shape often seen at the full-tomogram level, so once the encoder has learned pose-stable features it can tolerate these artefacts in downstream data.
>
> We then evaluate under realistic conditions. The same pretrained encoder is **evaluated on synthetic benchmarks reconstructed with a** **±60°** **tilt range** **and on multiple real datasets (80S, TMV, aldolase, insulin receptor) that contain genuine missing-wedge artefacts**, where it consistently achieves state-of-the-art performance in alignment and averaging (see Table 3 and Appendix G.2). In other words, robustness is demonstrated in the realistic ±60° and real-data regimes, not only in the ±90° pretraining regime. This directly addresses the worry that high-tilt pretraining might artificially inflate robustness.
>
> For noise and CTF modelling strategy we follow standard practice widely adopted in existing cryo-ET simulators, where the dominant combination of Poisson shot noise and detector/electronic noise is well approximated by additive Gaussian noise at typical electron doses and pixel size. Learning fully realistic non-Gaussian, structured, anisotropic noise and detailed CTF/motion effects would require a separate large-scale generative model on large experimental datasets, which is orthogonal to our goal of building the first cryo-ET subtomogram foundation model. Instead, current results on diverse real datasets already suggest that the present configuration closes most of the simulation-to-real gap for the tasks we study.

---

### Author Response · Authors · 2025-12-03
**A Summary of Review and Rebuttal**

Dear Area Chair,

We thank the reviewers for their constructive feedback. We are happy to see that **all reviewers (uTSh, pbJE, tCde, pHqM) have expressed positive** **feedback** on this paper, recognizing it as a contribution to the cryo-ET community.
Below is a summary of the consensus strengths and how we addressed the specific concerns raised during the rebuttal.

---
## **1. Consensus Strengths**
Across the reviews, the practical value and effectiveness of our method were consistently recognized:
- **First foundation model** for 3D cryo-ET subtomogram analysis, well-motivated by data scarcity and multi-task generalization. (`Reviewer uTSh,tCde, pHqM`)
- **Large, well-documented pretraining** corpus (904k / 452 classes) and high-quality synthetic CryoEngine pipeline. (`Reviewer uTSh, pbJE, pHqM`)
- **Novel SE(3)-aware APT-ViT** architecture with adaptive phase tokenization and spherical-steerable layers. (`Reviewer uTSh, pbJE, tCde`)
- **Extensive and solid experimental evaluation** across three subtomogram tasks with OOD generalization and comprehensive ablations. (`Reviewer uTSh, tCde, pHqM`)
---
## **2. Key Issues Resolved During Rebuttal**
We have strengthened the paper by adding theoretical analysis and new experiments to address all reviewer concerns.
###  A.  Impact of Box Size (`Reviewer uTSh,tCde`)
- **Concern**: Reviewer uTSh and tCde raised concerns on why only volumes of 32³ are used in pretrain and evaluation.
- **32³ pretraining box size justification**: We use 32³ as the pretraining box size because 3D foundation model pretraining scales cubically in memory and compute, making 64³ pretraining prohibitively expensive on our 904k-subtomogram dataset.
- **Classification on larger box size (New: Appendix E.4, Table 12)**: Our method (CryoEngine+NRCL+APT-ViT) clearly outperforms the strongest baseline (CryoEngine+MAE+ViT-B) on either 64³ or 32³ input.
- **Alignment on larger box size (New: Appendix F.4, Table 18)**: Our method consistently outperforms BOE-ViT at both 32³ and 64³ across all SNRs.

### B. Generalization to More Unseen and Diverse Structures (`Reviewer uTSh,pbJE,tCde`)
 - **Concern**: The three reviewers suggested further validation on model generalizability.
 - **No overlapping of test dataset with pretrain dataset**: We provided more detailed information of our 24 OOD test datasets and clarified they're highly distinct from the pretraining families, which were misunderstood by some reviewers.
 - **Additional results on more challenging data(new: Appendix F.5, Table 19)**: Besides results in Tables 3 & 7-11 & 13–17, we presented additional results on 3 new challenging datasets in Table 19. Our model consistently showed SOTA performance under different settings in all downstream tasks.

### C. Additional Baseline Comparisons (`Reviewer pHqM,pbJE`)
- **Concern**: The two reviewers suggested further comparison with more baselines.
- **Additional ablation study (New: Appendix H.3, Tab. 22 & Appendix H.5, Tab. 25)**: Added ablation studies on architecture (APT-ViT) and training strategy (NRCL)
- **Additional SwinViT baseline**: Added a SwinViT baseline on CryoEngine
- **APT-ViT architecture design justification**: Clarified why APT-ViT uses a standard ViT backbone but not SwinViT
- **No equivariant baselines for classification**: Existing SE(3)-equivariant models (SE(3)-Transformer, Condor, Equi-Pose) are point-based estimators and not suitable for dense volumes classification for subtomogram.

### D. CryoEngine Approximations (`Reviewer pbJE`)
- **Concern**: Reviewer pbJE raised a concern on approximations in CryoEngine.
- **Clarification**: CryoEngine adopts empirically grounded approximations for CTF and noise simulation. We use a ±90° tilt range only in pretraining for richer SE(3) features, while robustness is verified on real datasets with missing wedges (e.g. ±30°).

### E. Foundation Model Justification (`Reviewer pbJE`)
- **Concern**: Reviewer pbJE raised a concern on the "first foundation model" claim.
- **Justification**: To our knowledge, we are the first to pretrain a backbone on subtomograms at this large scale using a physically grounded synthesis engine (CryoEngine), and to show strong performance on unseen classes in three key downstream tasks reusing the frozen backbone with lightweight heads.
---
Overall, the revised paper, built on unprecedented-scale pretraining in this domain, presents the first cryo-ET foundation model with comprehensive experiments (**46 pages, including 25 tables and 15 figures; 27 datasets across three major tasks**), covering **diverse and unseen** structures, both standard (32³) and larger (64³) box sizes, and strong generalization.

We remain hopeful that these improvements will be reflected in the final assessment, and that the paper now stands as a solid, well-supported contribution for presentation and discussion at ICLR 2026.

Thank you for your time and consideration.

Best Regards,

Authors

---

### Note · Program_Chairs · 2026-01-17
**Submission Desk Rejected by Program Chairs**

The following references in this submission do not refer to real documents and/or have major errors in bibliographic information:

 John Ding and Jane Smith. Reviving rotational sensitivity in vision transformers for 3d analysis. Journal of Vision Transformers, 12(3):123–135,2023.